# Chemical capture of diazo metabolites reveals biosynthetic hydrazone oxidation

Katarina Pfeifer[1,3], Devon  Van Cura[1,3], Kelvin J. Y. Wu[1] & Emily P. Balskus[1,2✉]

Chemically reactive microbial natural products have enabled therapeutic development[1] via their well-established anticancer, antibiotic and antioxidant activities. However, discovery of reactive metabolites is particularly challenging because they may not tolerate traditional bioactivity-guided isolation workflows[2]. Diazo-containing natural products are a subset of highly reactive microbial metabolites that display potent bioactivity[3] and enable powerful biosynthetic transformations[4,5]; however, instability of the diazo group to light[6], heat[7], mild acid[8] and mechanical shock[9] has precluded their efficient discovery and application. Here we develop a reactivity-based screening approach to capture diazo-containing metabolites and facilitate their discovery by mass spectrometry. This workflow revealed two novel diazo-containing natural products, 4-diazo-3-oxobutanoic acid (**1**) and diazoacetone (**2**), from the human lung pathogen *Nocardia ninae*. Biosynthetic investigations revealed a distinct enzymatic logic for diazo formation involving hydrazone oxidation catalysed by the metalloenzyme Dob3, and its biochemical characterization suggests promising future applications in biocatalysis. Overall, our work highlights the power of reactivity-guided strategies for identifying reactive metabolites and facilitating the discovery of unique enzymatic transformations.

Microorganisms produce various natural products that contain the diazo functional group ($R^1R^2C=N^+=N^-$), including diazobenzofluorenes, diazobenzoquinones, α-diazoketones and α-diazoesters[3] (Fig. 1a). The diazo group is highly reactive owing to the strong thermodynamic driving force of $N_2$ release and can impart cytotoxicity to natural products through covalent and radical-induced damage of cellular targets. This has led to the exploration of diazo-containing metabolites as antibiotics and chemotherapeutic agents[3]. Diazo compounds are also used extensively as reagents in synthetic chemistry, as they enable powerful chemical transformations[10], and have emerged as important substrates for engineered haem-dependent enzymes that perform non-biological reactions in the context of biocatalysis[11] and synthetic biology[5].

The synthetic utility and biological activities of diazo-containing natural products have motivated efforts to understand their biosynthesis. So far, two distinct biosynthetic strategies to generate this functional group have been demonstrated biochemically. In the first strategy, diazotization of primary amines is accomplished with enzymatically produced nitrite, typically catalysed by ATP-dependent ligases[6,12–18]. The second strategy involves condensation of a hydrazide intermediate with a ketone followed by non-enzymatic oxidation to form the diazo group[19]. A third unconfirmed strategy proposed in azaserine biosynthesis involves generation of a hydrazine intermediate, oxidation to an α-hydrazonoacetyl intermediate, and conversion to an α-diazoester by two-electron *N*-oxidation[20–22] (Fig. 1b and Supplementary Fig. 1). However, a dedicated hydrazone

oxidoreductase has not been identified in this or any other biosynthetic pathway.

Although fewer than 30 diazo-containing natural products have been reported, bioinformatic investigations suggest a larger potential for microbial production of these metabolites. We performed a literature search to identify bioinformatically predicted and experimentally verified biosynthetic gene clusters involved in diazo formation. By combining these results with our own bioinformatic search (Methods), we identified 469 biosynthetic gene clusters that are potentially responsible for producing diazo-containing metabolites in diverse bacteria, including human-, plant- and animal-associated strains (Source Data for Fig. 1). However, only 20 of these gene clusters (4.3%) have been linked to 11 distinct natural products that contain a diazo group or are produced via a diazo-containing intermediate (Fig. 1c), suggesting as-yet untapped chemical diversity of diazo-containing natural products. Possible reasons for the relatively small number of characterized diazo-containing natural products include poor expression, low production titres and/or homologous gene clusters producing redundant metabolites. Furthermore, recent work has revealed biosynthetic pathways that utilize cryptic diazo-containing intermediates to produce metabolites that lack a diazo group[15–18]. Given the reactivity of the diazo group, we hypothesized that instability may also contribute to the relative scarcity of known diazo-containing metabolites. Challenges identifying and isolating reactive diazo metabolites are likely to impede the discovery of natural products with notable biological activities and biosynthetic pathways.

[1]Department of Chemistry and Chemical Biology, Harvard University, Cambridge, MA, USA. [2]Howard Hughes Medical Institute, Harvard University, Cambridge, MA, USA. [3]These authors contributed equally: Katarina Pfeifer, Devon Van Cura. ✉e-mail: balskus@chemistry.harvard.edu

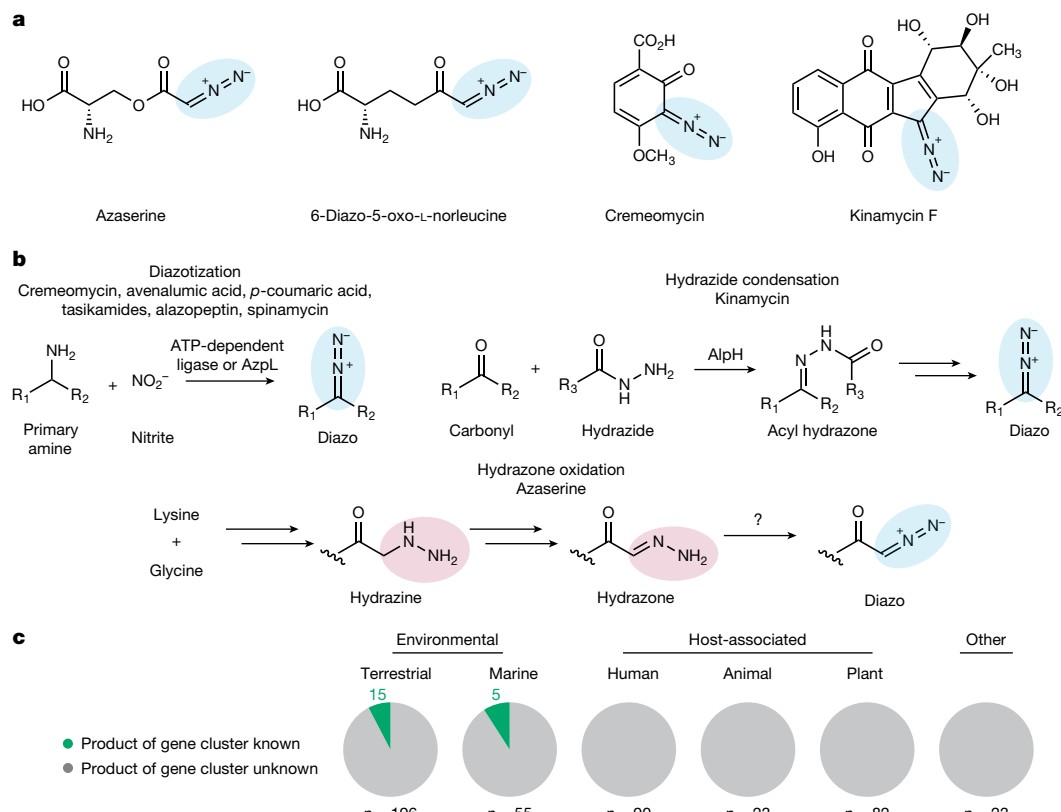

**Fig. 1 | Discovery of new diazo-containing natural products may reveal chemical and biosynthetic diversity. a**, Select diazo-containing natural products. **b**, Known diazo-forming biosynthetic logic includes diazotization of amines with nitrite, hydrazide condensation followed by non-enzymatic oxidation, and hydrazone oxidation. In azaserine biosynthesis, the hydrazone oxidoreductase has not yet been identified. **c**, Pie charts representing predicted bacterial biosynthetic gene clusters that encode diazo natural products.

Our genome mining, combined with genome mining from previous studies, revealed 469 biosynthetic gene clusters from diverse sources that are potentially involved in the biosynthesis of diazo-containing natural products. 'Host-associated' refers to any microorganism isolated from a plant or animal source and 'other' refers to organisms for which isolation information was not readily available.

## Reactivity-based screening workflow development

Traditional natural product discovery workflows, such as bioactivity-guided fractionation, do not address the key challenge of diazo group instability. We envisioned utilizing an alternative discovery strategy by leveraging the reactive diazo functional group as a handle for chemical derivatization. Reactivity-based screening uses chemical probes that are designed to react with a specific natural product functional group with minimal off-target reactivity, labelling metabolites of interest and facilitating their detection using liquid chromatography–mass spectrometry (LC–MS)-based metabolomics[2]. This approach has been utilized to identify and/or isolate natural products with various functional groups[2,23–28] (Supplementary Fig. 2). Furthermore, chemical probes may be designed to enhance the stability and detection of labelled metabolites.

To identify a chemical probe for our proposed discovery workflow, we examined diazo-alkyne cycloaddition reactions owing to their prior applications in complex biological matrices[29,30]. We also expected the resulting pyrazole products[31] to have increased ionization efficiency and lipophilicity relative to the corresponding diazo precursors, facilitating their detection by reversed-phase LC–MS. We began by investigating reactions between the model diazo-containing natural product azaserine and several alkynes. Of the alkynes tested, only strained cyclooctynes exhibited robust formation of the corresponding regioisomeric pyrazole products by LC–MS/MS (Extended Data Fig. 1a–c). Dibenzocyclooctyne C-6 acid (DBCO-acid) was selected for further optimization owing to its superior yields.

The model reaction between DBCO-acid and azaserine was subsequently optimized (Extended Data Fig. 1d–f). Aqueous solvents balanced high yields and facile sample handling, and a 16 h overnight incubation balanced product yield and experimental efficiency. Investigation of additional diazo compounds revealed good reactivity of DBCO-acid toward monosubstituted α-diazocarbonyls (Extended Data Fig. 2). During these analyses, we observed two peaks in the extracted ion chromatograms (EICs) of the DBCO-acid adducts, corresponding to the two pyrazole regioisomers produced during the 1,3-dipolar cycloaddition reaction. Products were not observed with disubstituted diazo-containing compounds, in contrast to previous reports[31,32].

To assess the ability of DBCO-acid to label diazo-containing natural products in complex biological matrices, we performed the reaction using spent culture medium from the azaserine producing bacterium *Glycomyces harbinensis*[33]. Targeted LC–MS/MS analysis of spent culture medium revealed a mass feature matching an authentic standard of azaserine-DBCO (**3**), validating the applicability of the workflow for detecting diazo-containing natural products in complex metabolite mixtures (Extended Data Fig. 3). We next used *G. harbinensis* spent medium to develop an untargeted LC–MS-based comparative metabolomics workflow (Extended Data Fig. 4). Comparative metabolomics of DBCO-acid treated spent medium at reaction initiation ($t_0$) versus a 16 h incubation ($t_{16}$) revealed seven metabolites that were significantly increased ($P < 0.05$, $\log_2(\text{fold change}) \geq 2$), including the expected regioisomers of **3**. These data validated our reactivity-based screening approach for diazo-containing natural product discovery.

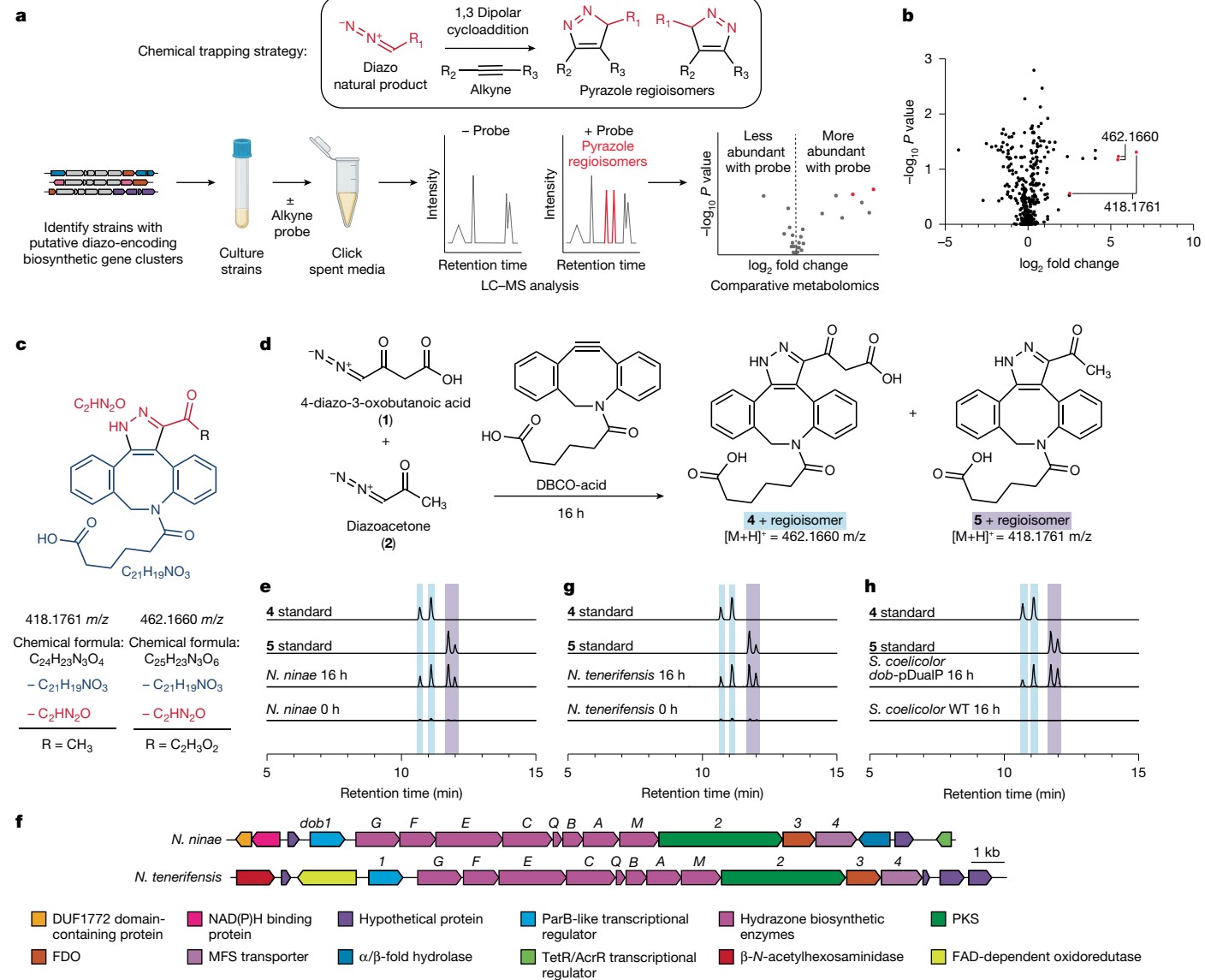

**Fig. 2 | A reactivity-guided comparative metabolomics workflow uncovers 1 and 2 from the human pathogen *N. ninae*. a**, Reactivity-based screening workflow. Adapted from a figure created in BioRender. Pfeifer, K. (2025) https://BioRender.com/fu16iy7. **b**, Volcano plot for comparative metabolomics of *N. ninae* spent medium. *P* values are calculated by a one-way ANOVA model with Tukey's post hoc test. **c**, Analysis of predicted chemical formulas suggests that **1** and **2** are novel diazo-containing natural products. **d**, Reaction of **1** and **2** with DBCO-acid yields **4** and **5**. **e**, EICs of $m/z$ = 418.1761 ± 5 ppm and $m/z$ = 462.1660 ± 5 ppm in DBCO-treated *N. ninae* spent media at $t$ = 0 h and $t$ = 16 h, compared to synthetic standards of **4** and **5**. EICs for 462.1660 and 418.1761 are

superimposed. $t_0$ and $t_{16}$ data were normalized to the $t_{16}$ maximum separately for **4** and **5**. **f**, The *dob* biosynthetic gene cluster in *N. ninae* and *Nocardia tenerifensis*. NRPS, non-ribosomal peptide synthetase. **g**,**h**, EICs of $m/z$ = 418.1761 ± 5 ppm and $m/z$ = 462.1660 ± 5 ppm in DBCO-treated *N. tenerifensis* spent medium (**g**) and *S. coelicolor dob*-pDualP spent medium (**h**) at $t$ = 0 h and $t$ = 16 h, compared to synthetic standards of **4** and **5**. All experiments were performed in biological triplicates and representative results are shown. EICs for 462.1660 and 418.1761 are superimposed. $t_0$ and $t_{16}$ data were normalized to the $t_{16}$ maximum separately for **4** and **5**. WT, wild type.

## Discovery of diazo-containing metabolites

We then sought to apply our workflow to discover previously unidentified diazo-containing microbial natural products (Fig. 2a). To prioritize organisms for screening, we mined bacterial genomes for biosynthetic gene clusters potentially involved in diazo production. We chose to target diazo-containing metabolites that were likely to be produced by hydrazone *N*-oxidation to facilitate the study of this enigmatic transformation previously proposed in azaserine biosynthesis[20–22]. We identified biosynthetic gene clusters of interest by searching for homologues of the hydrazone biosynthetic enzymes previously described in the s56-p1, triacsin and azaserine pathways[20–22,34–38] (Supplementary Fig. 3) encoded alongside at least one additional

predicted oxidoreductase. This analysis identified 129 biosynthetic gene clusters spanning a diverse range of Actinobacteria and Proteobacteria. Analysis of these gene clusters using prettyClusters[39] identified five conserved biosynthetic gene cluster architectures in addition to the known azaserine, triacsin and s56-p1 gene clusters (Extended Data Fig. 5). These gene clusters encode many additional biosynthetic enzymes, including potential diazo-forming oxidoreductases, indicating the potential for production of diverse natural product structures.

We selected ten commercially available strains, spanning three of the five gene cluster architectures identified in our bioinformatic search, for cultivation and comparative metabolomics analysis. Cultures were grown in up to six different growth media for five to seven days. Spent culture media were incubated with DBCO-acid, and

LC−MS-based comparative metabolomics were performed to identify potential diazo-containing natural products. Hits were observed from *N. ninae* and *Streptomyces yunnanensis* (Fig. 2b and Supplementary Fig. 4). Only two pairs of mass features obtained from spent medium of *N. ninae* were present in $t_{16}$ but not in $t_0$ samples and featured the doubled EIC peaks characteristic of DBCO-acid adduct regioisomers (418.1761 *m/z* and 462.1660 *m/z*) (Fig. 2b). Furthermore, their calculated molecular formulas ($C_{24}H_{23}N_3O_4$ ([M + H]$^+$ = 418.1761) and $C_{25}H_{23}N_3O_6$ ([M + H]$^+$ = 462.1660)), contained sufficient nitrogen atoms to support the presence of a pyrazole functional group. The molecular formula of each putative diazo-containing natural product was determined by subtracting the molecular formula of DBCO-acid ($C_{21}H_{19}NO_3$) to give $C_4H_4N_2O_3$ and $C_3H_4N_2O$ (Fig. 2c). On the basis of the reactivity profile of DBCO-acid and the hydrazone-forming biosynthetic enzymes present in this strain, we hypothesized that both natural products contained a hydrazone-derived α-diazoacetyl group. Subtracting the formula of the diazoacetyl group ($C_2HN_2O$) from the molecular formulas of both putative diazo-containing natural products yielded $C_2H_3O_2$ and $CH_3$ as the remaining atoms in each molecule, suggesting the original diazo natural products were 4-diazo-3-oxobutanoic acid (**1**) and diazoacetone (**2**), respectively (Fig. 2c,d).

To verify the structures of these natural products, we attempted to synthesize authentic standards of the probe adducts 4-diazo-3-oxobutanoic acid-DBCO (**4**) and diazoacetone-DBCO (**5**). Whereas **5** was readily accessible, **4** rapidly decarboxylated, precluding its isolation and purification. Accordingly, we synthesized the methyl ester **4-OMe**, which was stable and could be readily deprotected enzymatically in situ to provide **4** under mild conditions. LC−MS/MS analysis of **5** and deprotected **4-OMe** standards revealed features identical to the 462.1660 and 418.1761 masses observed from *N. ninae*, confirming the derivatized structures as **4** and **5**, respectively (Fig. 2e and Supplementary Fig. 5). Notably, we were unable to observe underivatized **1** and **2** in spent culture medium using LC−MS. Underivatized **2** yielded low peak areas at 10 µM and was undetectable at 1 µM, while derivatization of **2** with DBCO-acid enabled detection at 100 nM (Supplementary Fig. 6). The concentration of **2** in *N. ninae* spent culture medium is approximately 3 µM (Supplementary Fig. 7). This demonstrates that these metabolites would not have been readily identified using standard analytical techniques and highlights the utility of our reactivity-guided discovery workflow. **1** and **2** have not previously been identified in living systems, demonstrating the ability of this workflow to identify novel natural products.

## Gene cluster discovery and validation

We next set out to identify the biosynthetic gene cluster responsible for producing **1** and **2** to facilitate studies of diazo biosynthesis. Our genome mining analysis highlighted two potential diazo-forming biosynthetic gene clusters in *N. ninae*, *dob* and *nin* (Supplementary Fig. 8), which both encode hydrazone-forming enzymatic machinery and an additional oxidoreductase. To determine the gene cluster responsible for producing **1** and **2**, we formulated a biosynthetic hypothesis. Related hydrazone-forming pathways use a conserved set of enzymes to generate a carrier protein-bound α-hydrazonoacetyl thioester that is transferred to a nucleophilic intermediate[20–22,37]. The structures of **1** and **2** require the formation of a C−C bond between the α-hydrazonoacetyl thioester intermediate and a carbon-based nucleophile. Polyketide synthases (PKSs) catalyse C−C bond-forming Claisen-type condensation reactions. On the basis of the structure of **2**, we hypothesized that a PKS might catalyse C−C bond formation between malonyl-CoA and the α-hydrazonoacetyl thioester intermediate to yield 4-hydrazono-3-oxobutanoic acid (**6**), which could feasibly undergo a subsequent 2-electron *N*-oxidation to produce the diazo functional group of **1**. We hypothesized that **2** could be produced by spontaneous non-enzymatic decarboxylation of **1**.

Only the *dob* gene cluster encoded a predicted PKS consistent with a role in the biosynthesis of **1** and **2** (Fig. 2f). BLAST searches and genomic neighbourhood analysis using the Enzyme Function Initiative-Genome Neighborhood Tool (EFI-GNT) indicated that 14 additional *Nocardia* strains contain the *dob* gene cluster[40,41] (Supplementary Table 6). To test the link between this gene cluster and the biosynthesis of **1** and **2**, a selection of these strains was cultivated, spent medium was derivatized with DBCO-acid, and targeted LC−MS/MS analysis was performed to assess production of **4** and **5**. In spent medium from one of the strains, *N. tenerifensis*, we detected mass features matching standards of **4** and **5**, strengthening the relationship between the *dob* gene cluster and production of **1** and **2** (Fig. 2g and Supplementary Fig. 9). Derivatization of *N. tenerifensis* spent medium with DBCO-acid yielded 1 µM **5** (Supplementary Fig. 10), further highlighting the utility of our approach for the detection of diazo-containing natural products produced in low titres.

To confirm the role of the *dob* gene cluster in the biosynthesis of **1** and **2**, we expressed it in a heterologous host. The *dob* biosynthetic gene cluster was cloned into a dual-inducible vector (*dob*-pDualP; Supplementary Fig. 11) and conjugated into *Streptomyces coelicolor* M1152. *S. coelicolor dob*-pDualP was cultured in a variety of growth media supplemented with ε-caprolactam and oxytetracycline inducers. Spent culture medium was derivatized with DBCO-acid and assessed for production of **4** and **5**. Only the strain carrying *dob*-pDualP, and not the wild-type strain, produced **4** and **5** (approximately 2 µM), definitively linking the *dob* gene cluster to biosynthesis of **1** and **2** (Fig. 2h and Supplementary Figs. 12 and 13).

## Discovery of a diazo-forming metalloenzyme

With the role of the *dob* gene cluster in production of **1** and **2** validated, we next sought to assign biosynthetic roles to each of the encoded enzymes. On the basis of the homology of DobG, E, F, B, C, Q, M and A to hydrazone-forming enzymes previously reported in the triacsin[37] and azaserine[20–22] pathways, we anticipated these genes would similarly produce a carrier protein-bound α-hydrazonoacetyl thioester (**7**) intermediate from L-lysine and glycine (Supplementary Fig. 3). Analysis of the proteins encoded by the remaining *dob* genes suggested the type I PKS Dob2 as a likely candidate to catalyse C−C bond formation and the ferritin-like diiron oxidase or oxygenase (FDO) Dob3 as a potential diazo-forming oxidoreductase. The remaining biosynthetic genes are predicted to encode a transcriptional regulator (Dob1) and a major facilitator superfamily transporter (Dob4).

To investigate the biosynthesis of **1** and **2**, we first sought to heterologously express and reconstitute the activities of the putative hydrazone biosynthetic enzymes[20–22,37] (Supplementary Fig. 14; for raw gel image see Supplementary Fig. 46). As expected, the *N*-oxygenase DobG catalysed the NAD(P)H- and FAD-dependent oxidation of L-lysine to *N*-6-hydroxylysine (**8**) (Extended Data Fig. 6). Attempts to reconstitute activity of the MetRS/cupin fusion enzyme DobE in vitro were unsuccessful; however, biotransformations using *Escherichia coli* expressing DobE incubated with **8**, glycine and ATP yielded *N*-6-carboxylmethylaminolysine (**9**) (Extended Data Fig. 7). Incubation of hydrazinoacetic acid (**10**) with succinyl-CoA and the GCN5-*N*-acetyltransferase DobB produced succinyl-hydrazinoacetic acid (**11**) (Extended Data Fig. 8).

We reconstituted the activity of the remaining hydrazone biosynthetic enzymes in a single cascade reaction (Extended Data Fig. 9). Compound **10** was incubated with succinyl-CoA, DobB, ATP, DobC, *holo*-DobQ, FAD, DobM and DobA. Loading of **11** onto the carrier protein DobQ by adenylase DobC yielded succinyl-hydrazinoacetic acid-DobQ (**12**). In contrast to previous reports, we observed accumulation of a mass feature consistent with succinyl-hydrazonoacetic acid-DobQ (**13**) in assays lacking the C45 peptidase DobA. Thus, we hypothesize that **12** is oxidized to **13** by the FAD-dependent oxidoreductase DobM and **13** is hydrolysed by DobA to release the previously observed

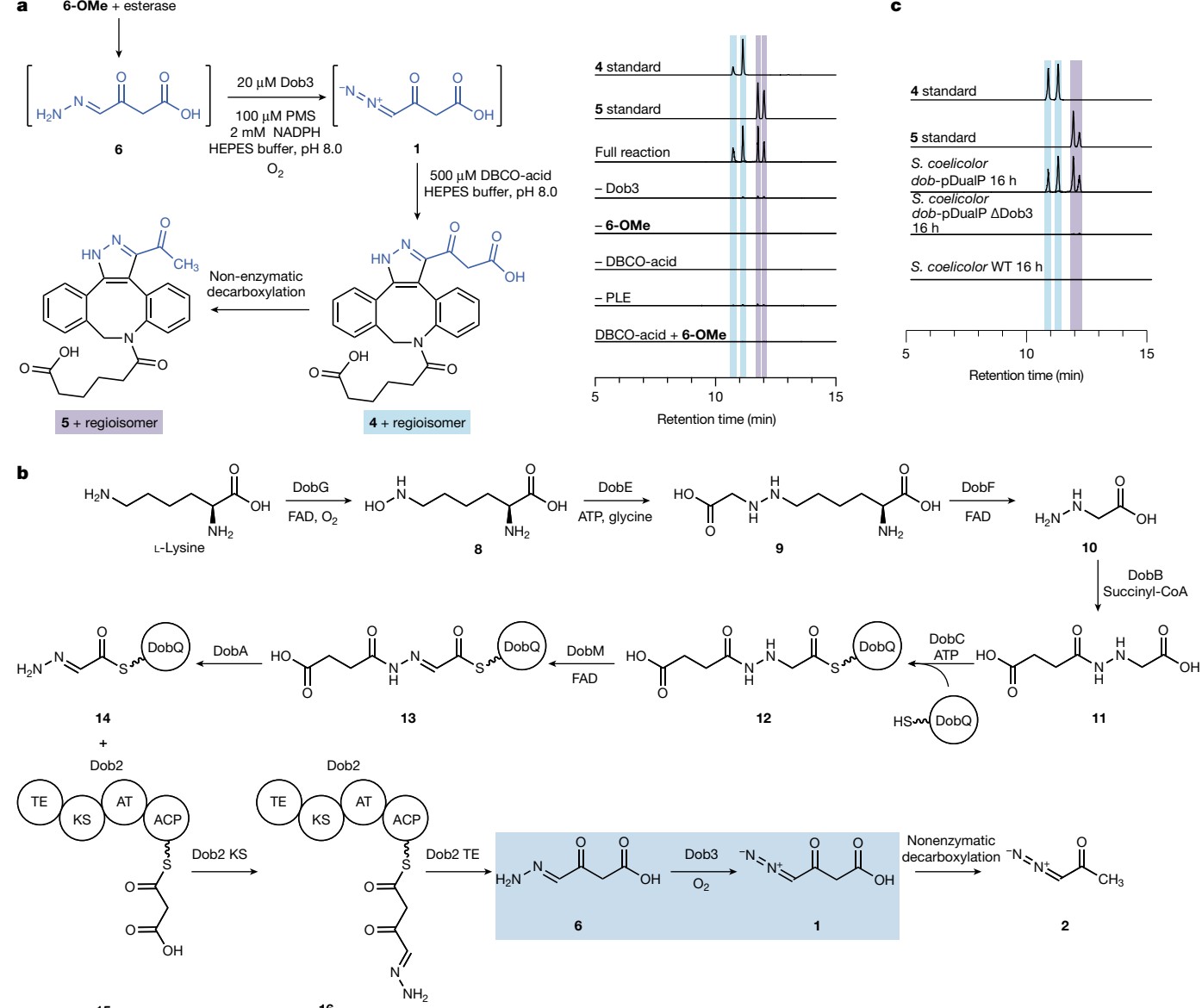

**Fig. 3 | Reconstitution of Dob3 activity confirms hydrazone oxidation as a novel diazo-forming biosynthetic strategy. a**, Overnight incubation of Dob3 with **6**-OMe, PMS, PLE, NADPH and DBCO-acid at pH 8.0 yields **1** and **2**, demonstrating Dob3 catalyses *N*-oxidation of free **6** to produce **1**. EICs of $m/z = 418.1761 \pm 5$ ppm and $m/z = 462.1660 \pm 5$ ppm. EICs for 462.1660 and 418.1761, respectively, are superimposed. The standard of **4** was prepared by incubation of **4**-OMe with PLE. **b**, Proposed biosynthetic pathway of **1** based on in vitro biochemical experiments. TE, thioesterase; KS, ketosynthase; AT, acyltransferase; ACP, acyl carrier protein. **c**, EICs of $m/z = 418.1761 \pm 5$ ppm and $m/z = 462.1660 \pm 5$ ppm in DBCO-treated *S. coelicolor dob*-pDualP $\Delta dob3$ spent medium at $t = 16$ h compared to probe-treated *S. coelicolor dob*-pDualP. $t_0$ and $t_{16}$ data were normalized to the $t_{16}$ maximum separately for **4** and **5**. EICs for 462.1660 and 418.1761 are superimposed. All experiments were performed in biological triplicates. Representative results are shown.

intermediate hydrazonoacetic acid-DobQ (**14**). Our observation of hydrolysis after oxidation differs from previous reports which suggested that hydrolysis either precedes[22] or is concurrent with[20,37] oxidation.

After confirming that the hydrazone-forming enzymes produce the expected α-hydrazonoacetyl intermediate **14**, we sought to investigate the unique enzymatic transformations leading to production of **1** and **2**. Type I PKS Dob2 is predicted to contain thioesterase, ketosynthase, acyltransferase and acyl carrier protein domains. The acyltransferase domain of Dob2 was predicted to load methylmalonyl-CoA by antiSMASH[42], but based on the structures of **1** and **2**, we hypothesized that this domain could use malonyl-CoA to form malonyl-Dob2 (**15**). On the basis of canonical PKS logic, we hypothesized that **14** may be translocated from DobQ to a conserved Cys residue in the ketosynthase domain of Dob2 prior to ketosynthase domain-catalysed C–C bond-forming decarboxylative Claisen condensation with a malonyl extender unit to form 4-hydrazono-3-oxobutanoic acid-Dob2 (**16**). Finally, the thioesterase domain of Dob2 might catalyse translocation to a conserved Ser residue prior to hydrolytic release of the mature PKS product **6** (Supplementary Fig. 15).

The putative diazo-forming enzyme Dob3 was the sole unassigned oxidoreductase encoded in the *dob* gene cluster and therefore the most likely candidate for hydrazone oxidation. A structure homology search of the Dali Webserver[43] using an AlphaFold[44] predicted structure of Dob3 revealed strong structural homology to the characterized ferritin-like diiron oxygenases (FDOs) AurF ($Z = 40.1$, 31% sequence identity) and CmlI ($Z = 40.9$, 47% sequence identity), which catalyse six-electron *N*-oxidation of aryl amines to aryl nitro groups during the biosynthesis of aureothin and chloramphenicol, respectively[45,46]

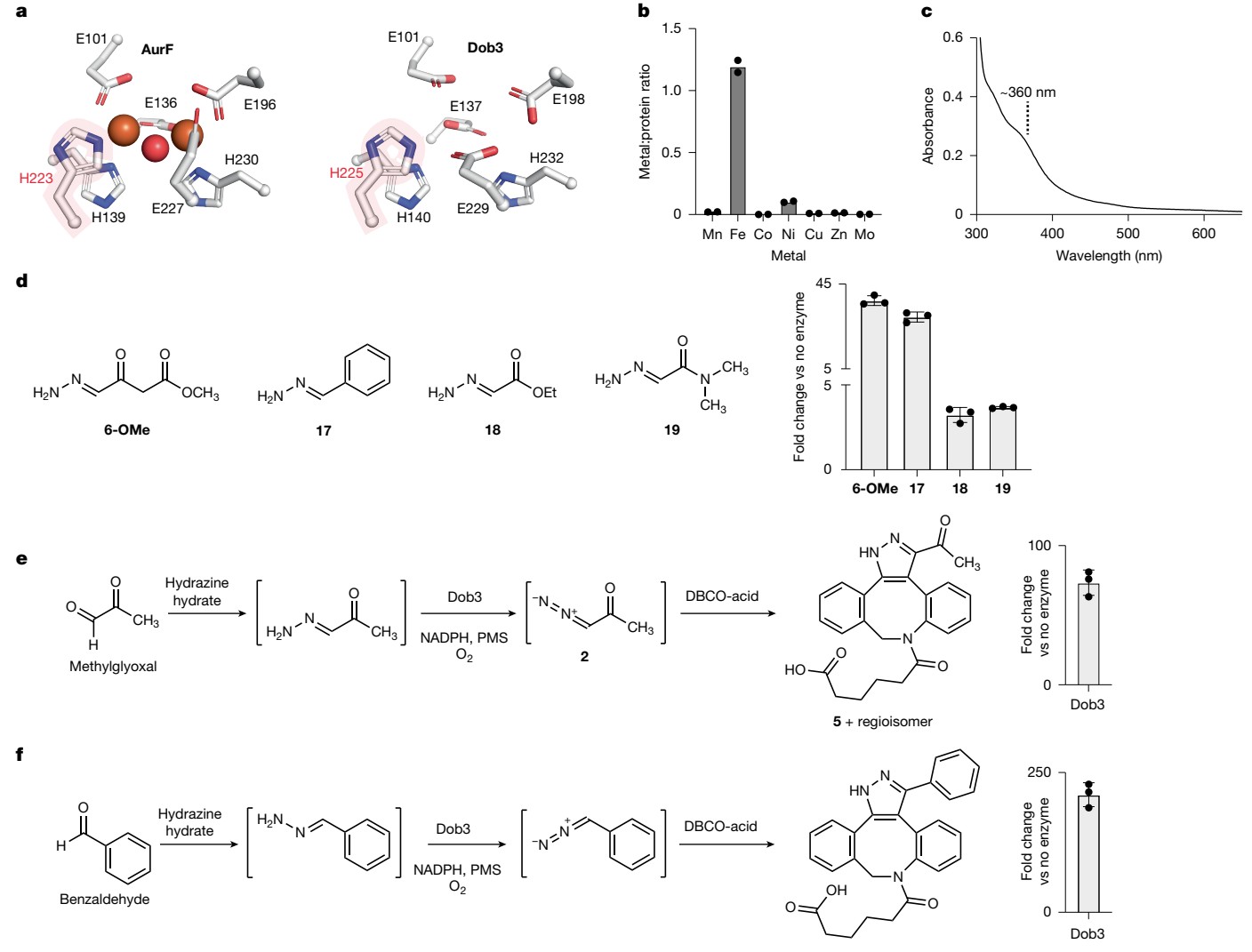

**Fig. 4 | Dob3 catalyses hydrazone oxidation. a**, Comparison of active site residues in the AurF crystal structure and Dob3 AlphaFold structure. **b**, ICP-MS reveals that Fe is the most abundant metal in Dob3 and is present at 1.2 equivalent Fe atoms per monomer of protein. Experiment was performed in biological duplicate, and mean values are shown. **c**, UV-Vis analysis of Dob3 is consistent with a μ-oxo-diferric cofactor in as-isolated protein. **d**, Dob3 oxidizes varied hydrazone substrates. **e**, One-pot incubation of methylglyoxal,

hydrazine hydrate, Dob3 and DBCO-acid produces **5**. **f**, One-pot incubation of benzaldehyde, hydrazine hydrate, Dob3 and DBCO-acid produces **20**. All experiments were performed in biological triplicates unless otherwise indicated. All peak areas were calculated as the sum of regioisomer peak areas. Error bars indicate mean ± s.d. Fold change was calculated by dividing the peak areas by the average peak area of no enzyme controls.

(Supplementary Fig. 16). On the basis of this analysis, we hypothesized that Dob3 could catalyse *N*-oxidation of a hydrazone to a diazo group during the biosynthesis of **1**. However, the order of reactions catalysed by Dob2 and Dob3 in the pathway remained unclear.

We next set out to characterize these enzymes by first attempting to reconstitute the activity of Dob2 in vitro. However, incubation of *holo*-Dob2 and malonyl-CoA with enzymatically produced **14** did not produce a mass feature consistent with the predicted hydrazone product **6**. We suspected this could be due to a combination of low yields and poor ionization efficiency rather than incorrect assignment of the biochemical transformation. Accordingly, we attempted to use **6** as a substrate to reconstitute the activity of the putative diazo-forming enzyme Dob3. As the susceptibility of **6** to decarboxylation precluded its direct synthesis, we synthesized the *O*-methyl ester **6-OMe** and used enzymatic deprotection to generate **6** in situ. Incubation of **6-OMe** with porcine liver esterase (PLE), Dob3, phenazine methosulfate (PMS), NADPH and DBCO-acid revealed production of **1** and **2**, demonstrating the ability of Dob3 to catalyse hydrazone *N*-oxidation to form the

diazo group of **1** (Fig. 3a). Investigation of alternative redox systems demonstrated that PMS and NADPH provided the greatest product conversions (Supplementary Fig. 17). The *dob* gene cluster does not encode a putative redox partner protein, suggesting the native redox partner is likely to be encoded elsewhere in the genome. Given the facile detection of **4** and **5** by LC–MS, we then revisited the activity of Dob2 in a coupled enzyme assay with Dob3 followed by DBCO-acid derivatization. Enzymatically generated **14** was incubated with Dob2, malonyl-CoA, Dob3, PMS, NADPH and DBCO-acid and resulted in the production of **4** and **5**, confirming the role of Dob2 in this pathway (Extended Data Fig. 10).

To evaluate the importance of Dob3 in the biosynthesis of **1** and **2** in vivo, we designed the heterologous expression vector *dob*-pDualP Δ*dob3* in which Dob3 is replaced with an ampicillin resistance gene (Supplementary Fig. 18). The *dob*-pDualP Δ*dob3* vector was transferred into *S. coelicolor* M1152 via conjugation, and the *S. coelicolor dob*-pDualP Δ*dob3* strain was cultured. Spent medium was derivatized with DBCO-acid, and LC–MS analysis demonstrated loss of **4** and

5 (Fig. 3b), supporting the necessity of Dob3 for the biosynthesis of 1 and 2 in vivo.

To probe the timing of diazo formation in the biosynthesis of 1 and 2, we investigated whether Dob3 could oxidize 14 prior to C–C bond formation by Dob2. Dob3 and PMS were incubated with the hydrazone biosynthetic enzymes and DBCO-acid in the absence of Dob2. However, no pyrazole products were detected. This suggests that C–C bond formation by Dob2 precedes diazo formation by Dob3 (Fig. 3c). We next attempted to probe the ability of Dob3 to oxidize Dob2-tethered 6 (16) using Ppant ejection and whole protein LC–MS assays. However, we did not detect the expected Dob2-tethered diazo product, potentially due to its instability. Attempts to trap labile intermediates using DBCO-acid were also unsuccessful. As the ability of Dob3 to oxidize 16 cannot be ruled out, the order of thioester hydrolysis versus diazo formation during the biosynthesis of 1 and 2 remains uncertain.

## Dob3 is a unique FDO enzyme

We next sought to explore the biochemical basis for the biosynthetically unprecedented hydrazone oxidation reaction catalysed by the FDO Dob3. Members of the FDO family have been demonstrated to catalyse a variety of C–H oxygenation, C–H desaturation and N–H oxygenation reactions. However, no FDO N-oxygenase has previously been described to catalyse two-electron oxidation of the hydrazone functional group.

Structural predictions with AlphaFold suggest that Dob3 is likely to be dimeric (Supplementary Fig. 19), consistent with size-exclusion chromatography of purified Dob3 (Supplementary Fig. 20). Comparison of the predicted structure of Dob3 with known FDO structures revealed the six conserved metal-binding residues present in all FDOs (E101, E137, H140, E198, E229 and H232), as well as a seventh conserved histidine residue characteristic of FDO N-oxygenases (H225)[47] (Fig. 4a and Supplementary Fig. 21).

FDOs typically employ diiron or mixed Mn/Fe dimetal cofactors. Inductively coupled plasma mass spectrometry (ICP-MS) of the as-isolated protein expressed in LB medium revealed that iron was the most abundant metal, and approximately 1.2 equivalents of Fe per Dob3 monomer were present (Fig. 4b). Ultraviolet–visible (UV-Vis) spectroscopy of as-isolated Dob3 revealed a broad feature at around 360 nm, consistent with a μ-oxo-diferric cofactor and analogous to the UV-Vis spectra of as-isolated AurF and VlmB[48–50] (Fig. 4c). To probe the possibility that a metal other than Fe is important for Dob3 activity, we attempted to obtain apo-Dob3 for supplementation with various metals. However, expression of Dob3 in minimal medium lacking metal supplementation resulted in insoluble protein. We also attempted expression in M9 medium supplemented with only Mn[51], but the protein was insoluble. Similarly, expression of variants in which one of the putative metal-binding residues (either E101, E137, H140, E198, H225, E229 or H232) was mutated to alanine also resulted in insoluble protein, suggesting the importance of metal binding for Dob3 stability. Indeed, expression in M9 medium supplemented with only Fe or both Fe and Mn yielded soluble protein, and ICP-MS analysis indicated incorporation of roughly 2.4 and 3.1 equivalents of Fe per Dob3 monomer, respectively (Supplementary Fig. 22). Together, these data strongly suggest that Dob3 employs a diiron cofactor. We hypothesize that the diiron cofactor reacts with molecular $O_2$ to produce an intermediate which either hydroxylates the terminal nitrogen atom of a hydrazone intermediate followed by dehydration or promotes direct dehydrogenation to produce the diazo functional group (Supplementary Fig. 23).

The synthetic utility of diazo compounds inspired us to investigate the substrate scope of Dob3 to assess its early potential as a diazo-forming biocatalyst. We tested the activity of Dob3 toward structurally varied hydrazones, including 6-OMe, benzylidenehydrazine (17), ethyl hydrazonoacetate (18) and hydrazonodimethylacetamide (19), using derivatization with DBCO-acid to facilitate product detection.

Conversion to the corresponding pyrazole product was observed for all substrates to varying degrees, demonstrating the potential of Dob3 to oxidize a variety of hydrazone substrates (Fig. 4d). We also showed that Dob3 could oxidize hydrazones generated in situ from condensation of hydrazine and commercially available aldehydes to yield 5 and diazomethylbenzene-DBCO (20) (Fig. 4e,f). Low levels of non-enzymatic hydrazone oxidation were observed for some substrates (ranging from 3.4% to 31% of enzymatic levels) (Supplementary Figs. 24 and 25). Overall, the activity of Dob3 towards additional substrates, including precursors to important chemical reagents such as ethyl diazoacetate, highlights its promise as a starting point for engineering diazo-producing biocatalysts.

## Discussion

Microorganisms produce a variety of natural products that contain chemically reactive functional groups, and such molecules often have potent biological activity and unusual structural features that may be assembled via enzymatic chemistry. However, traditional workflows for natural product discovery may not be suitable for identifying and characterizing reactive microbial metabolites, and it is likely that many remain undetected. Developing improved methods to enable the discovery of these natural products could reveal novel bioactive metabolites and biosynthetic enzymes.

Previously, reactivity-based screening approaches have been used to detect metabolites containing labile functional groups. We demonstrate here that this strategy can also be applied to discover reactive diazo-containing natural products. Our approach addresses long-standing challenges of instability and low titres that have hindered the study of these metabolites. Notably, our workflow identified the novel diazo-containing natural products 1 and 2 from the human pathogen N. ninae and enabled elucidation of their biosynthesis.

Previously characterized diazo-containing metabolites have potent biological activity and may play important biological roles for the microorganisms that produce them. The importance of these natural products is further emphasized by the phylogenetic and ecological diversity of strains containing gene clusters putatively involved in biosynthesizing diazo-containing natural products, including microorganisms found in soil and aquatic environments as well plant, insect and mammalian pathogens. To our knowledge, 1 and 2 are the first diazo-containing natural products identified from a human pathogen, and their discovery raises interesting questions about the biological roles of these metabolites in virulence and human disease.

The discovery of diazo-containing natural products also has broader relevance to synthetic chemistry. Carbene transfer reactions using engineered haem proteins are a powerful emerging class of stereoselective biocatalytic transformations[11]. Studies on the integration of unnatural carbene transfer reactions into microbial biosynthesis have demonstrated the production of chiral cyclopropanes from glucose, presenting an economically and environmentally attractive approach for pharmaceutical bioproduction[5]. However, metabolic engineering using unnatural carbene transfer pathways is currently limited by the small number of diazo-containing natural products that can act as carbene donors[3,5]. Compound 2 is a powerful and extensively utilized synthetic reagent that has previously enabled carbene transferase-catalysed reactions[52,53]. The enzymes involved in biosynthesis of 2 may be deployed in engineered biosynthetic carbene transfer pathways, with potential impacts on renewable biomanufacturing.

Characterizing the biosynthesis of 1 and 2 led to the identification of Dob3, a member of the FDO enzyme family and the first diazo-forming biosynthetic enzyme that is known to catalyse 2-electron N-oxidation of hydrazones. Elucidation of this pathway provides biochemical confirmation that hydrazone oxidation is utilized to produce

diazo-containing natural products, as previously proposed for azaserine biosynthesis[20–22]. However, the *aza* biosynthetic gene cluster lacks an encoded FDO, suggesting that a different enzyme family catalyses hydrazone oxidation in that pathway.

Wild-type Dob3 provides a promising starting point for developing engineered biocatalysts to produce synthetically useful diazo compounds from readily accessible hydrazone intermediates. This enzymatic activity remained unknown until recent work describing the use of vanadium-dependent haloperoxidase enzymes to oxidize non-native hydrazone substrates[54]. These enzymes are hypothesized to activate the terminal hydrazone nitrogen via $H_2O_2$-dependent halogenation, followed by elimination to form the diazo compound. We hypothesize that diazo formation catalysed by Dob3 follows a distinct chemical logic whereby Dob3 uses a diiron cofactor and molecular $O_2$ to oxidize the hydrazone, potentially through *N*-oxygenation followed by dehydration or direct dehydrogenation. The discovery of Dob3 expands the scope of transformations performed by FDO enzymes and illustrates how biosynthetic pathways that construct reactive natural products are rich sources of unappreciated enzymatic reactions with the potential to enable chemical bioproduction.

Finally, our work suggests that many chemically reactive microbial natural products may have remained undiscovered and uncharacterized owing to their instability. Natural product biosynthetic gene clusters predicted to generate reactive functional groups are found in many microbial strains, including human pathogens, suggesting that reactive microbial metabolites have important yet underappreciated biological roles. We anticipate that additional reactivity-guided approaches may facilitate further exploration of this microbial 'dark matter' for natural product and biosynthetic enzyme discovery.

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

## Methods

### General experimental procedures

Primer synthesis and DNA sequencing were performed by Genewiz. Purification of recombinant plasmids was performed using an EZNA Plasmid DNA Mini Kit from Omega Bio-Tek. Restriction enzymes were purchased from New England Biolabs and digestions were performed according to the manufacturer's protocols. Gibson Master Mix was purchased from New England Biolabs and Gibson Assembly was performed according to the manufacturer's protocol. Nickel nitriloacetic acid agarose (Ni-NTA) resin was purchased from Qiagen and Thermo Fisher Scientific. Benzylidenehydrazine was purchased from AstaTech. NADPH was purchased from Fisher Scientific. DBCO-acid was purchased from Conju-Probe. Diazoacetone was purchased from Enamine. Novex Tris-Glycine SDS–PAGE gels were purchased from Thermo Fisher Scientific. Protein concentrations were determined by measuring UV-absorption at 280 nm using a Thermo Scientific NanoDrop 2000. ExPASy ProtParam was used to calculate extinction coefficients. Optical densities of *E. coli* cultures were measured at 600 nm using a Beckman Coulter DU730 Life Sciences UV/Vis spectrophotometer. All water used was purified using a MilliQ water purification system. The remaining materials were purchased from Sigma Aldrich unless otherwise indicated. All experiments were performed in biological triplicates unless indicated and either individual data points with errors bars (± 1 s.d.) or representative results are shown. All starting materials used for chemical synthesis were either purchased from Sigma Aldrich or accessed synthetically as described below.

### Sample preparation method 1

Ten microlitres of reaction mixture were diluted with 150 μl of LC–MS grade $H_2O$, 20 μl of LC–MS grade acetonitrile (ACN), and 20 μl of LC–MS grade methanol and samples were filtered (0.2 μm, VWR, nylon).

### Sample preparation method 2

Reaction mixtures were diluted with 70 μl of exchange buffer (20 mM HEPES pH 8.0, 10 mM $MgCl_2$, 50 mM NaCl, 10% glycerol), 15 μl of LC–MS grade methanol, and 15 μl of LC–MS grade ACN and filtered (3 kDa, Pall, Omega membrane)

### Agilent 6530 qTOF spectrometer with Dual AJS source

Unless otherwise indicated, samples were analysed by LC–MS using an Agilent 1200 series LC system coupled to an Agilent 6530 qTOF spectrometer with Dual AJS source. Drying gas temperature was 300 °C, drying gas flow was 11 l min⁻¹, nebulizer pressure was 45 psi, sheath gas temperature was 275 °C, sheath gas flow was 11 l min⁻¹, capillary voltage was 3,500 V, nozzle voltage was 500 V, fragmentor voltage was 125 V, and skimmer voltage was 65 V. Mass spectra were acquired in positive ion mode. A mass window of 10 ppm was used for EICs.

### Thermo Orbitrap IQ-X Tribrid

Unless otherwise indicated, samples were analysed using a Horizon Vanquish UHPLC coupled to a Thermo Orbitrap IQ-X Tribrid mass spectrometer. The LC column was a Kinetex C18 column (1.7 μm, 100 Å, 150 × 2.1 mm, Kinetex). Two microlitres of sample were injected. The flow rate was 0.4 ml min⁻¹ using mobile phase A (0.1% formic acid in water) and B (0.1% formic acid in ACN). The LC conditions were 0–2 min 10% B isocratic, 2–29 min 10–90% B, 29–32 min 90% B isocratic, 32–32 min 90–10% B, 32–35 min 10% B isocratic. The MS settings were: mass range 400–600 $m/z$, 120,000 resolution, the RF lens was 35%, the standard AGC target was used, and the auto maximum injection time mode was used. MS/MS spectra were acquired with a 1 $m/z$ isolation window, 15,000 resolution, standard AGC target, auto maximum injection time mode and higher-energy collision-induced dissociation (HCD) fragmentation with 30% collision energy. Spectra were acquired in positive mode. A mass window of 5 ppm was used for EICs.

### Comparative metabolomics using AcquireX and Compound Discoverer

Data-dependent MS/MS acquisition was performed via AcquireX (deep scan mode) from 4 replicate injections with media + DBCO-acid providing an exclusion list. Data-dependent MS/MS spectra were acquired with $2.0 × 10^4$ intensity threshold, exclusion after 1 time, 2.5 s exclusion duration, a 3 ppm isolation window, and HCD, collision-induced dissociation (CID) and ultraviolet photodissociation (UVPD) fragmentation. HCD fragmentation was performed with a 1 $m/z$ isolation window, stepped collision energies of 20, 35, 50, 75 and 100%, and 15,000 resolution. CID fragmentation was performed with a 1 $m/z$ isolation window, 30% collision energy, 15,000 resolution, and 22 ms maximum injection time. UVPD fragmentation was performed with a 1.6 $m/z$ isolation window, molecular weight-dependent UVPD activation time, 12% activation time, 15,000 resolution, and 22 ms maximum injection time.

Comparative metabolomics for $t_0$ and $t_{16}$ samples were performed using Compound Discoverer: The raw spectra of all samples were imported into Compound Discoverer 3.3 (Thermo Scientific), where peak alignment, compound detection, and compound grouping was performed. Compound detection was performed with a 3 ppm mass tolerance, a 10,000 minimum peak intensity, 1.5 chromatographic signal/noise threshold, and isotope grouping for Br and Cl ions. All ions were considered. Compound grouping was performed with a 3 ppm mass tolerance, a 0.2 min retention time tolerance, preferred ions of $[M + H]^+$ and $[M − H]^-$ and area integration of the most common ion. Peak rating contributions were 3 for area contribution, 10 for coefficient of variation contribution, 5 for full width at half maximum (FWHM) to base contribution, 5 for jaggedness contribution, 5 for modality contribution, and 5 for zig-zag index contribution. Peak rating filters were 3 for peak rating threshold and 2 for number of files. Gap filling was performed with a mass tolerance of 5 ppm and a signal to noise threshold of 1.5. Systematic error removal using random forest (SERRF) QC correction was applied with a 50% minimum QC coverage, a 30% maximum QC area relative s.d., a 25% maximum corrected QC area relative s.d., 2 batches, and interpolated gap filling. Background compounds were marked with a maximum sample/blank ratio of 5, a maximum blank/sample ratio of 0 and the background compounds were then hidden. MS spectra were assigned as $t_{16}$ or $t_0$ and differential analysis was performed following spectra processing. Differential analysis was performed with area values transformed to log10. Peak rating contributions for differential analysis were 3 for area contribution, 10 for coefficient of variation contribution, 5 for FWHM to base contribution, 5 for jaggedness contribution, 5 for modality contribution, and 5 for zig-zag index contribution. Following statistical analysis, results were filtered to include only those with a calculated molecular weight and $m/z$ greater than 333.

### Reaction of azaserine with alkyne probes

One-hundred microlitre reaction mixtures containing 100 μM azaserine and 100 μM of the specified alkyne (10 mM stock solution in methanol) were prepared in water. The reaction mixtures were incubated for 16 h at room temperature. Samples were prepared using method 1. Detection of probe adducts was accomplished using the method described above for the Thermo Orbitrap IQ-X Tribrid mass spectrometer.

### Optimization of the reaction with azaserine and DBCO-acid

**Solvent screen.** One-hundred microlitre reaction mixtures containing 100 μM azaserine and 100 μM DBCO-acid (10 mM stock solution in methanol) in $H_2O$, 1:1 $H_2O$:methanol, methanol, 1:1 $H_2O$:ACN, or ACN were incubated overnight at room temperature. Samples were prepared using method 1. Detection of **3** was accomplished using the method described above for the Thermo Orbitrap IQ-X Tribrid mass spectrometer.

**Reaction time screen.** One-hundred microlitre reaction mixtures containing 100 μM azaserine and 500 μM DBCO-acid (10 mM stock solution in methanol) in $H_2O$ were incubated for the specified amount of time at room temperature. At the specified time point, samples were prepared using method 1. Detection of **3** was accomplished using the method described above for the Thermo Orbitrap IQ-X Tribrid mass spectrometer.

**DBCO-acid equivalents.** One-hundred microlitre reaction mixtures containing 100 μM azaserine and the specified equivalents of DBCO-acid (1 mM stock solution in methanol) in 10% methanol in $H_2O$ were incubated overnight at room temperature. Samples were prepared using method 1. Detection of **3** was accomplished using the method described above for the Thermo Orbitrap IQ-X Tribrid mass spectrometer.

### DBCO-acid substrate scope
Two-hundred and fifty microlitre reaction mixtures containing 100 μM diazo substrate (azaserine stock solution: 25 mM in $H_2O$; *t*-butyldiazoacetate, benzyldiazoacetate, ethyl diazoacetate stock solution: 25 mM in DMSO) and 500 μM DBCO-acid (50 mM stock solution in methanol) in $H_2O$ were incubated for 18 h at room temperature. Samples were prepared using method 1. Detection of DBCO-acid adducts was accomplished using the method described above for the Thermo Orbitrap IQ-X Tribrid mass spectrometer.

### *G. harbinensis* $t_0$ versus $t_{16}$ comparative metabolomics
*G. harbinensis* was grown on NZ amine plates (10 g l$^{-1}$ glucose, 20 g l$^{-1}$ soluble starch, 5 g l$^{-1}$ yeast extract, 5 g l$^{-1}$ NZ-amine, 1 g l$^{-1}$ CaCO$_3$, 15 g l$^{-1}$ agar) at 30 °C for 7–10 days and was then used to inoculate 5 ml of NZ amine medium. The NZ amine culture was incubated for 7–10 days at 30 °C with shaking at 200 rpm. Four-hundred microlitres of the liquid culture were used to inoculate 20 ml of molasses production medium (10 g l$^{-1}$ glucose, 5 g l$^{-1}$ Bacto peptone, 20 g l$^{-1}$ molasses, 1 g l$^{-1}$ CaCO$_3$) which was incubated at 30 °C with shaking at 200 rpm for 14 days. Two-hundred microlitres of culture were spun down for 5 min at maximum speed. One microlitre of 25 mM DBCO-acid in methanol was added to 49 μl of supernatant. The $t_0$ control was obtained by dilution of 10 μl of the reaction mixture into 150 μl of LC–MS grade $H_2O$ + 20 μl of LC–MS grade ACN + 20 μl of LC–MS methanol, filtration (0.2 μm, VWR, nylon), and storage at −80 °C prior to analysis. The $t_{16}$ sample was obtained from a 16 h incubation of the remaining reaction mixture at room temperature, followed by dilution of 10 μl of the reaction mixture into 150 μl of LC–MS grade $H_2O$ + 20 μl of LC–MS grade ACN + 20 μl of LC–MS grade methanol, and filtration (0.2 μm, VWR, nylon). Comparative metabolomics were performed using a Thermo Orbitrap IQ-X Tribrid mass spectrometer as described above.

### Compilation of diazo biosynthetic gene clusters
To compare the number of experimentally validated versus bioinformatically predicted biosynthetic gene clusters involved in diazo formation, we first performed a literature search to identify all experimentally verified diazo biosynthetic gene clusters[5,12,14–18,20–22,38,55–63] (20 clusters identified; see Source Data for Fig. 1). We then performed a literature search to compile prior genome mining efforts to identify biosynthetic gene clusters potentially involved in diazo formation. Janso et al.[55] identified homologues of putative diazo-forming enzymes by querying *lom29*, *lom32*, *lom33*, *lom34* and *lom35* against sequenced bacterial genomes (note: it is unclear whether gene or protein sequences were used for this search). Accession numbers and sequences for all protein sequences used as queries are provided in Supplementary Table 7. Protein sequences or accession numbers were provided if gene accession numbers could not be identified. Waldman performed a gene cluster architecture search using *creD*, *creE* and *creM* as queries against

the GenBank database using the Integrated Microbial Genomes-Joint Genome Institute (IMG/JGI) MultiGeneBlast program[56]. Kawai et al.[14] identified clusters containing *azpL* homologues by querying *azpL* against an unspecified genome database. Kawai et al.[16,18] identified homologues of the *ava* gene cluster using BiG-SCAPE. Van Cura et al.[40,41] identified homologues of the *aza* gene cluster by using the EFI-GNT to query AzaDHILNP against the Uniprot database[20]. Shikai et al.[22] identified homologues of the *azs* gene cluster using antiSMASH and BiG-SCAPE. Compiling predicted gene clusters from these studies and dereplicating based on strain name and query sequence yielded 382 bioinformatically predicted diazo biosynthetic gene clusters. Compilation of these results with the 87 biosynthetic gene clusters identified through our genome mining (see Source Data for Fig. 1) resulted in Fig. 1c. Because prior studies were performed using earlier versions of databases, the actual number of predicted diazo biosynthetic gene clusters is likely to be greater than 382.

### Genome mining for biosynthetic gene clusters involved in hydrazone N-oxidation
In short, homologues of the hydrazine-forming enzyme AzaE were identified via a BLAST search of the NCBI reference protein database (refseq_protein). The genes neighbouring these homologues were retrieved using prettyClusters[39] to identify putative biosynthetic gene clusters. Manual filtering was performed to select only gene clusters containing Pfams consistent with putative hydrazone biosynthetic enzymes and an additional oxidoreductase.

To begin this analysis, the protein sequence of AzaE (WP_091038156.1) was queried against the NCBI nucleotide collection database (November 2023) and the non-redundant protein sequence database in tblastn and blastp searches, respectively. Blastp results with an E value <10$^{-98}$ that were non-redundant were added to the tblastn search results. GenBank files were downloaded using the accession number for each result. The files were prepared according to the prettyClusters[39] workflow, detailed below. The GenBank documents were annotated with Prokka[64]. Amino acid sequences were extracted and added to a local database. AzaE was queried against this database in a blastp search and homologues with an E value <10$^{-31}$ were taken forward. Gene and neighbour metadata and neighbour sequences were generated with the gbToIMG function using default values except NeighborNumber was set to 15. InterProScan was used to generate additional metadata for all genes and neighbours. These data were merged with the previous metadata using the incorpIprScan function. The prepNeighbors function was run with NeighborNumber = 10 and trimShortClusters = false. AzaE homologues were dereplicated using the Enzyme Function Initiative-Enzyme Similarity Tool (EFI-EST)[40,41] tool and the repNodeTrim function. Manual curation identified gene neighbourhoods containing homologues of hydrazone-forming enzymes (neighbourhoods contained at least one representative each of Pfam02770, Pfam13434, Pfam01266, Pfam00501, Pfam13302, and Pfam03417; Pfam00550 was not considered as acyl carrier proteins are commonly not annotated). Additional curation identified gene neighbourhoods containing an additional oxidoreductase as indicated by Pfam annotations. The analyzeNeighbors function was run on the clusters containing the hydrazone biosynthetic enzymes with neighborThreshold = 2.5 and tgCutoff = 56. Results were visualized with Cytoscape[65]. This analysis identified 129 gene clusters. Of these, 2 were related to s56-p1 biosynthesis and 14 were related to triacsin biosynthesis, yielding 113 putative diazo biosynthetic gene clusters. Of these, 26 were previously identified as putative azaserine biosynthetic gene clusters and dereplication of these clusters yielded 87 unique clusters. Combining these results with the 382 biosynthetic gene clusters identified in previous studies resulted in Fig. 1c (see Source Data for Fig. 1). While the NCBI reference protein database contains proteins from a variety of source,s including plants, animals and fungi, hits were observed only from bacterial sources.

## Diazo-containing natural product discovery from *N. ninae*

*N. ninae* was grown on GYM agar (4 g l⁻¹ glucose, 4 g l⁻¹ yeast extract, 10 g l⁻¹ malt extract, 2 g l⁻¹ CaCO₃, 20 g l⁻¹ agar) plates for 7–10 days at 30 °C. Colonies were used to inoculate 5 ml of GYM medium (4 g l⁻¹ glucose, 4 g l⁻¹ yeast extract, 10 g l⁻¹ malt extract), and the culture was incubated for 7–10 days at 30 °C with shaking at 200 rpm. One-hundred microlitres of the culture were used to inoculate 5 ml of GYM medium and this was incubated for 7–10 days at 30 °C with shaking at 200 rpm. Two-hundred microlitres of spent culture medium were spun down at maximum speed for 5 min. One microlitre of 25 mM DBCO-acid in methanol was added to 49 µl of spent culture medium and incubated for 16 h at room temperature. Samples were prepared using method 1. Samples were analysed using a Thermo Orbitrap IQ-X Tribrid mass spectrometer as described above with minor modifications. The MS settings were: mass range 70–700 $m/z$. Comparative metabolomics for $t_0$ and $t_{16}$ samples were performed using Compound Discoverer as described in the general methods above.

## Detection of 2 by LC–MS

Compound **2** was detected using a Thermo Orbitrap IQ-X Tribrid mass spectrometer as described above with minor modifications. The MS settings were: mass range 70–700 $m/z$.

## Detection of 4 and 5 from *N. tenerifensis*

*N. tenerifensis* was grown on GYM agar plates for 7–10 days at 30 °C. Colonies were used to inoculate 5 ml of GYM medium and the culture was incubated for 7–10 days at 30 °C with shaking at 200 rpm. One-hundred microlitres of the culture were used to inoculate 5 ml of molasses production medium, and this culture was incubated for 7–10 days at 30 °C with shaking at 200 rpm. Two-hundred microlitres of spent culture medium were spun down at maximum speed for 5 min. One microlitre of 25 mM DBCO-acid in methanol was added to 49 µl of spent culture medium and incubated overnight at room temperature. Samples were prepared using method 1. Detection of **4** and **5** was accomplished using a Thermo Orbitrap IQ-X Tribrid mass spectrometer as described above with minor modifications. The MS settings were: mass range 70–700 $m/z$. MS/MS was accomplished using HCD fragmentation with 30% normalized collision energy or CID fragmentation with 30% normalized collision energy, 10 ms activation time, and 0.25 activation Q for **5** and **4**, respectively.

## Cloning *dob*-pDualP

Construction of the *dob*-pDualP vector was performed as previously reported for *aza*-pDualP, with minor modifications[20]. Cloning of *dob*-pDualP was performed by Terra Bioforge. *N. ninae* cells were lysed using proprietary methods to maintain genomic DNA integrity. Genomic DNA > 50 kb was isolated using proprietary gel free methods. The *dob* gene cluster was excised from *N. ninae* genomic DNA using Cas9 with guide RNAs targeting upstream and downstream of the gene cluster. A *Streptomyces* compatible dual-inducible vector (pDualP) containing the Potr inducible expression system[66] (oxytetracycline) and *PnitA*[67,68] inducible promoter (ε-caprolactam) flanking the insertion site was amplified using PCR with primers designed to create ~40 bp of overlap specific to the *dob* gene cluster. Modified Gibson Assembly of the *dob* gene cluster and PCR amplified linear pDualP was performed using a proprietary reaction mix to produce the *dob*-pDualP expression vector. The *dob*-pDualP vector was transformed into *E. coli* BacOpt2.0 (*E. coli* DH10b derivative) and clones were verified by whole plasmid sequencing.

## Conjugation of *dob*-pDualP into *S. coelicolor* M1152

*S. coelicolor* M1152 was grown on MSF agar (20 g l⁻¹ mannitol, 20 g l⁻¹ soy flour, 20 g l⁻¹ agar) for 7 days at 30 °C. A single colony was used to inoculate 5 ml of YEME medium (3 g l⁻¹ yeast extract, 3 g l⁻¹ malt extract, 5 g l⁻¹ peptone, 10 g l⁻¹ glucose, 340 g l⁻¹ sucrose), which were incubated at 30 °C with shaking at 200 rpm for 7 days. Five millilitres of LB + apramycin (50 µg ml⁻¹) + kanamycin (50 µg ml⁻¹) + chloramphenicol (20 µg ml⁻¹) medium were inoculated with *E. coli* ET12567/pUZ8002 *dob*-pDualP and grown at 37 °C with shaking at 200 rpm for 2 days. The culture was spun down at 3,000 rpm for 10 min. Five millilitres of LB medium were added and the pellet was resuspended. The resuspension was spun down at 3,000 rpm for 10 min and the supernatant was discarded. This wash step was then repeated before the pellet was resuspended in 100 µl of LB medium. The *S. coelicolor* culture was spun down at 4,000 rpm for 10 min on the same day. Two millilitres of sucrose (10% w/v in water) were added, the pellet was resuspended, and the resuspension was spun down at 4,000 rpm for 10 min. This was repeated once before the pellet was resuspended in 100 µl of LB. One microlitre of *S. coelicolor* suspension were combined with 100 µl of *E. coli* suspension and mixed. This was plated on MSF + 10 mM MgCl₂ agar and grown at 30 °C. After 18 h, the plate was flooded with 1 ml of sterile water containing 1 mg apramycin and 0.5 mg nalidixic acid before being incubated at 30 °C. Ex-conjugants were grown on MSF + apramycin (50 µg ml⁻¹) + nalidixic acid (30 µg ml⁻¹) agar and incorporation of *dob*-pDualP was confirmed by genome sequencing.

## Detection of 4 and 5 from *S. coelicolor dob*-pDualP

*S. coelicolor dob*-pDualP was grown on MSF + nalidixic acid (30 µg ml⁻¹) + apramycin (50 µg ml⁻¹) agar plates at 30 °C for 5–7 days and then used to inoculate 5 ml of YEME medium. Wild-type *S. coelicolor* was grown on MSF agar plates at 30 °C for 5–7 days and used to inoculate 5 ml of YEME medium. The YEME cultures were incubated at 30 °C with shaking at 200 rpm for 7 days. One-hundred microlitres of the YEME cultures were used to inoculate 5 ml of molasses production medium which was incubated at 30 °C with shaking at 200 rpm for 14 days. ε-caprolactam and oxytetracycline were added to the cultures for final concentrations of 0.1% w/v and 2.5 mM, respectively. After incubation of the induced cultures at 30 °C with shaking at 200 rpm for 7 days, 200 µl of spent culture medium were derivatized and analysed as described above with minor modifications. The MS settings were: mass range 400–480 $m/z$. MS/MS spectra were acquired with a 5 ppm isolation window, 15,000 resolution, 1,000% AGC target, dynamic maximum injection time, and HCD fragmentation with 30% normalized collision energy or CID fragmentation with 30% collision energy, 10 ms activation time, and 0.25 activation Q for **5** and **4**, respectively.

## Estimation of the concentration of 5 in derivatized culture supernatants

Compound **5** was diluted to 50 µM, 5 µM and 500 nM, in GYM medium or 10 µM, 1 µM, and 100 nM in molasses or ISP4 medium. Samples were prepared in biological triplicates. *N. ninae*, *N. tenerifensis*, and *S. coelicolor dob*-pDualP were cultured and their supernatants derivatized with DBCO-acid as described above. LC–MS samples for the derivatized supernatants and standard curves were prepared using method 1 and analysed using a Thermo Orbitrap IQ-X Tribrid mass spectrometer as described above for the detection of **5** from *N. tenerifensis*. The EIC of **5** was extracted ± 5 ppm and the combined peak area of the regioisomeric peaks was plotted in GraphPad. A simple linear regression was performed using the GraphPad software. Each replicate $y$ value was considered as an individual point and the range of the regression line was automatically calculated. The equation of the resulting regression line was then used to approximate the concentration of **5** in the derivatized supernatants.

## Cloning *dob*-pDualP Δ*dob3*

The deletion of Dob3 was performed by Terra Bioforge. In brief, *dob*-pDualP was digested in vitro with CRISPR nuclease complexed with two guide RNAs targeting both the 5′ and 3′ ends of the FDO gene. The digested DNA was then repaired in an isothermal assembly reaction

with the AmpR cassette from pUC19, which was amplified by PCR using the primers 5′-gaaatggagggatagccgtgtaagtttaaacggcactttttcgggg-3′ and 5′-aaattggacagcagccgcttgtttaaacgttaccaatgcttaatcagtgagg-3′. The assembly reaction to remove the FDO gene and replace it with the AmpR cassette was then transformed via electroporation into *E. coli* DH10B. The transformation culture was then plated on LB agar containing 25 µg ml⁻¹ apramycin and 100 µg ml⁻¹ ampicillin. Transformant colonies were picked, and plasmid DNA was extracted. Whole plasmid sequencing was performed by Plasmidsaurus using Oxford Nanopore Technology with custom analysis and annotation.

### Conjugation of *dob*-pDualP Δ*dob3* into *S. coelicolor* M1152

Five millilitres of LB + apramycin (25 µg ml⁻¹) + chloramphenicol (10 µg ml⁻¹) + kanamycin (25 µg ml⁻¹) medium were inoculated with *E. coli* ET12567/pUZ8002 *dob*-pDualP Δ*dob3* and grown at 37 °C with shaking at 200 rpm overnight. After the culture reached OD₆₀₀ > 0.4, it was spun down at 3,000 rpm for 10 min and the supernatant was discarded. The pellet was resuspended in 5 ml of LB medium. The resuspension was spun down at 3,000 rpm for 10 min and the supernatant was discarded. This wash step was repeated once before the pellet was resuspended in 500 µl of LB medium. Ten microlitres of *S. coelicolor* M1152 spore stock were added to 500 µl of 2× YT medium (16 g l⁻¹ casein digest peptone, 10 g l⁻¹ yeast extract, 5 g l⁻¹ NaCl) and incubated for 10 min at 50 °C. Samples were then cooled to room temperature. Five-hundred microlitres of *E. coli* ET12567/pUZ8002 *dob*-pDualP ΔDob3 in LB medium were added to 500 µl of *S. coelicolor* in 2× YT medium. Samples were spun down at 3,000 rpm for 10 min and 850 µl of supernatant were removed. The pellet was resuspended in the remaining supernatant and plated on MSF + 50 mM MgCl₂ agar. The plate was then incubated at 30 °C for 24 h before it was flooded with 1 ml of sterile water containing 1 mg of apramycin and 0.5 mg of nalidixic acid. Ex-conjugants were grown on MSF + apramycin + nalidixic acid agar and incorporation of *dob*-pDualP ΔDob3 was confirmed by genome sequencing.

### Genome sequencing verification of *S. coelicolor dob*-pDualP Δ*dob3*

*S. coelicolor dob*-pDualP and *S. coelicolor dob*-pDualP Δ*dob3* were grown on MSF + apramycin (50 µg ml⁻¹) agar for 7 days at 30 °C. Single colonies were used to inoculate 5 ml of YEME medium (3 g l⁻¹ yeast extract, 3 g l⁻¹ malt extract, 5 g l⁻¹ peptone, 10 g l⁻¹ glucose, 340 g l⁻¹ sucrose) which were incubated for 7 days at 30 °C with shaking at 200 rpm. Genomic DNA was extracted using the Lucigen MasterPure Gram Positive gDNA Purification Kit. Samples were incubated with Ready-Lyse lysozyme for 36 h and eluted with 10 µl of H₂O. Genomic DNA sequencing was performed by Plasmidsaurus using Oxford Nanopore Technology with custom analysis and annotation.

### Comparison of *S. coelicolor dob*-pDualP versus *S. coelicolor dob*-pDualP Δ*dob3* spent media

Comparison of *S. coelicolor dob*-pDualP and *S. coelicolor dob*-pDualP Δ*dob3* spent media was performed using the method described above for the detection of **4** and **5** from *S. coelicolor dob*-pDualP with the following modification: 100 µl of YEME starter culture were used to inoculate 5 ml of ISP4 medium (10 g l⁻¹ soluble starch, 1 g l⁻¹ MgSO₄•7H₂O, 1 g l⁻¹ NaCl, 2 g l⁻¹ (NH₄)₂SO₄, 2 g l⁻¹ CaCO₃ and 1 ml of trace salts solution (stock solution: 1 g l⁻¹ FeSO₄•7H₂O, 1 g l⁻¹ MnCl₂•4H₂O, 1 g l⁻¹ ZnSO₄•7H₂O)).

### Isolation of *N. ninae* genomic DNA

*N. ninae* was grown on GYM agar plates for 7–10 days at 30 °C. Single colonies were used to inoculate 5 ml of GYM medium and the culture was incubated for 7–10 days at 30 °C with shaking at 200 rpm. Genomic DNA was extracted using the Lucigen MasterPure Gram Positive gDNA Purification Kit with a 16 h lysozyme incubation. Samples were eluted with 10 µl of H₂O.

### Cloning DobA, DobB, DobQ, DobC, DobG, DobM, Dob2 and Dob3

Genes encoding each protein were amplified by PCR using the appropriate forward and reverse primers listed in Supplementary Table 1. PCR products were isolated using Zymoclean Gel DNA Recovery Kit and DNA Clean & Concentrator. PCR products were ligated into a NdeI/HindIII-digested pET28a plasmid using Gibson Assembly. Plasmids were transformed into *E. coli* Top10 and the sequence was confirmed by Sanger sequencing. Plasmids containing the *dobA*, *dobB*, *dobC*, *dobG*, *dobM* and *dob3* inserts were transformed into *E. coli* BL21(DE3) for protein expression. Plasmids carrying the *dobQ* (carrier protein) and *dob2* (PKS) inserts were transformed into *E. coli* BAP1 (ref. 69) for expression of phosphopantetheinylated holoenzymes.

### Site-directed mutagenesis

Dob3 point mutants were constructed by adapting the Agilent QuikChange protocol[70]. PCR reactions contained 18 µl of H₂O, 1 µl of pET28a-Dob3 (86.1 ng µl⁻¹), 2 µl of DMSO, 25 µl of Q5 High-Fidelity 2× Master Mix and 1 µl of appropriate primers (10 µM stock) listed in Supplementary Table 1. Two microlitres of DpnI were used to digest source plasmids following amplification. Plasmids were transformed into *E. coli* Top10 and the sequence was confirmed by Sanger sequencing. Plasmids containing the mutated inserts were transformed into *E. coli* BL21(DE3) for protein expression.

### Expression and purification of DobA, DobB, DobC, DobG, DobM, DobQ, Dob2, Dob3 and Dob3 variants

A single colony was used to inoculate 30 ml of LB medium + kanamycin (50 µg ml⁻¹) which were incubated overnight at 37 °C with shaking at 200 rpm. A 20 ml aliquot was used to inoculate 1 l of LB medium + kanamycin (50 µg ml⁻¹) culture which was incubated at 37 °C with shaking at 180 rpm until an OD₆₀₀ of at least 0.6 was reached. Cultures were cold-shocked on ice for 30 min before induction with 50 µg ml⁻¹ isopropyl β-D-1-thiogalactopyranoside (IPTG). For DobG and DobM, 75 mg l⁻¹ riboflavin were added after induction. Cultures were incubated at 18 °C with shaking at 180 rpm overnight. After 16 h, cells were centrifuged at 4,000 rpm at 4 °C for 20 min. Pellets were resuspended in 30 ml of 20 mM HEPES pH 8.0, 10 mM MgCl₂, and 500 mM NaCl. For DobG and DobM, 1 mM FAD was added to the lysis buffer. Cells were lysed using a cell disruptor (Avestin EmulsiFlex-C3) and lysates were spun down at 15,000*g* for 40 min at 4 °C. Spent culture medium was combined with 3 ml of Ni-NTA resin and incubated with gentle rotation at 4 °C for 30 min. The resin was washed with buffer containing 25 mM imidazole before eluting proteins in buffer containing 200 mM imidazole. Eluted proteins were concentrated to <2 ml using a MilliporeSigma Amicon Ultra-15 Centrifugal Filter Unit with an appropriate molecular weight cutoff. DobB, DobC, DobG, DobM, DobQ, Dob2, Dob3 and Dob3 mutants were desalted using buffer (20 mM HEPES pH 8.0, 10 mM MgCl₂, 50 mM NaCl, 10% glycerol) and Cytiva PD-10 columns pre-packed with Sephadex G-25 resin. DobA was desalted and further purified on a Bio-Rad BioLogic DuoFlow FPLC system with a Superdex 75 column using 20 mM HEPES pH 8.0, 10 mM MgCl₂, 50 mM NaCl, and 10% glycerol as the mobile phase. Desalted proteins were concentrated using a MilliporeSigma Amicon Ultra-15 Centrifugal Filter Unit with an appropriate molecular weight cutoff before storing concentrated proteins at −80 °C. To determine the molecular weight of Dob3, Bio-Rad Gel Filtration (1511901, batch 64656562) standards were analysed on a Bio-Rad BioLogic DuoFlow FPLC system with a Superdex 200 column using 20 mM HEPES pH 8.0, 10 mM MgCl₂, 50 mM NaCl, and 10% glycerol as the mobile phase. Dob3 was analysed using the same system with a Superdex 200 column using 20 mM HEPES pH 8.0, 10 mM MgCl₂, 50 mM NaCl, and 10% glycerol as the mobile phase to determine its molecular weight.

## In vitro activity assay of DobG

The in vitro activity of DobG was investigated as previously described for the homologue AzaG[20], with the modification that supernatants were additionally passed through a 0.2 µM filter (VWR, 13 mm, Nylon, 0.2 µm) prior to LC–MS analysis. MS/MS spectra were collected with 10 V collision energy.

## In vivo activity assay of DobE

Analysis of in vivo activity of DobE was conducted as previously described for AzaE[20].

## In vitro activity assay of DobB

The in vitro activity of DobB was investigated as previously described for AzaB[20] with the following modifications: After the reaction, samples were diluted 95:5 $H_2O$:ACN and filtered (3 kDa, Amicon, cellulose).

## Ppant ejection assay of DobA, DobB, DobC, DobM and DobQ

The Ppant ejection assay of DobA, DobB, DobC, DobM, and DobQ was conducted as previously described for their homologues AzaB, C, M, and Q[20], with the modification that 40 µM DobA (AzaA homologue) was also added to the reaction mixtures.

## In vitro activity assay of Dob3

Fifty-microlitre reaction mixtures containing 100 µM phenazine method sulfate (PMS), 20 µM Dob3, 1 mM **6-OMe** (50 mM stock solution in methanol), 2 mM NADPH, 0.1 mg PLE, and 500 µM DBCO-acid (25 mM stock solution in methanol) in exchange buffer (20 mM HEPES pH 8.0, 10 mM $MgCl_2$, 50 mM NaCl, 10% glycerol) were incubated at room temperature overnight. Samples were prepared using method 2. Detection of **4** and **5** was accomplished using a Thermo Orbitrap IQ-X Tribrid mass spectrometer as described above with minor modifications. The MS settings were: mass range 70–700 $m/z$. MS/MS spectra of **5** were obtained using HCD fragmentation with 30% normalized collision energy. MS/MS spectra of **4** were obtained using CID fragmentation with 30% collision energy, 10 ms activation time and 0.25 activation Q.

## DobA, DobB, DobC, DobM, DobQ, Dob2 and Dob3 cascade reaction

Fifty-microlitre reaction mixtures containing 1 mM **10**, 20 µM DobB, 200 µM succinyl-CoA, 50 µM DobQ, 20 µM DobC, 5 mM ATP, 20 µM DobM, 200 µM FAD, 40 µM DobA, 10 µM Dob2, 200 µM malonyl-CoA, 20 µM Dob3, 100 µM PMS, 2 mM NADPH, and 500 µM DBCO-acid (25 mM stock solution in methanol) in exchange buffer (20 mM HEPES pH 8.0, 10 mM $MgCl_2$, 50 mM NaCl, 10% glycerol) were incubated aerobically at room temperature for 16 h. Samples were prepared using method 2. Detection of **4** and **5** was accomplished using a Thermo Orbitrap IQ-X Tribrid mass spectrometer as described above with minor modifications. The MS settings were: mass range 70–700 $m/z$. MS/MS spectra of **5** were obtained using HCD fragmentation with 30% normalized collision energy. MS/MS spectra of **4** were obtained using CID fragmentation with 30% collision energy, 10 ms activation time and 0.25 activation Q.

## Optimization of Dob3 redox system

Fifty-microlitre reaction mixtures containing 1 mM **6-OMe**, 20 µM Dob3, 500 µM DBCO-acid, and a redox system (2 mM NADPH + 100 µM PMS, 1 mM ascorbate, 1 mM dithiothreitol, 100 µg $ml^{-1}$ spinach ferredoxin + 100 mU $ml^{-1}$ spinach ferredoxin reductase + 1 mM NADPH, or no redox system) in exchange buffer (20 mM HEPES pH 8.0, 10 mM $MgCl_2$, 50 mM NaCl, 10% glycerol) were incubated aerobically at room temperature overnight. Samples were prepared using method 2. Detection of **4-OMe** was accomplished using a Thermo Orbitrap IQ-X Tribrid mass spectrometer as described above with minor modifications. The MS settings were: mass range 70–700 $m/z$.

## Optimization of Dob3 electron mediator

Fifty-microlitre reaction mixtures containing 1 mM **6-OMe**, 2 mM NADPH, 20 µM Dob3, 500 µM DBCO-acid, and 100 µM electron mediator (phenazine methosulfate, phenazine ethosulfate, or methyl viologen) were incubated aerobically at room temperature overnight. Samples were prepared using method 2. Detection of **4-OMe** was accomplished using a Thermo Orbitrap IQ-X Tribrid mass spectrometer as described above with minor modifications. The MS settings were: mass range 70–700 $m/z$.

## ICP-MS analysis of Dob3 expressed in LB

Sixty microlitres of trace-metal free nitric acid was added to 200 µl of 1 mg $ml^{-1}$ Dob3 in water. This mixture was incubated for 3 h at 60 °C before precipitates were removed by centrifugation. Two-hundred microlitres of supernatant was diluted to a total volume of 3 ml with molecular biology grade water. Samples were analysed on an Agilent 7900 Inductive Coupled Mass Spectrometer.

## ICP-MS analysis of Dob3 expressed in M9 with metal supplementation

A 105 µl volume of trace-metal free nitric acid were added to 350 µl of 1 mg $ml^{-1}$ Dob3 in water. This mixture was incubated for 3 h at 60 °C before precipitates were removed by centrifugation. Four-hundred microlitres of supernatant were diluted to a total volume of 5 ml with molecular biology grade water. ICP-MS was performed at the Northwestern University Quantitative Bulk-Element Information Core on a computer-controlled (QTEGRA software) Thermo iCapQ ICP-MS (Thermo Fisher Scientific) operating in KED mode and equipped with a ESI SC-2DX PrepFAST autosampler. Nickel skimmer and sample cones were used from Thermo Scientific (part numbers 1311870 and 3600812). Internal standard was added inline using the prepFAST system and consisted of 1 ng $ml^{-1}$ of a mixed element solution containing Bi, In, [6]Li, Sc, Tb and Y (IV-ICP-MS-71D from Inorganic Ventures). Each sample was acquired using 1 survey run (10 sweeps) and 3 main (peak jumping) runs (40 sweeps). The isotopes selected for analysis were [55]Mn, [56,67]Fe, [59]Co, [60,62]Ni, [63,65]Cu, [64,66,68]Zn, [95]Mo, and [45]Sc, [89]Y and [115]In (chosen as internal standards for data interpolation and machine stability). Instrument performance was optimized daily through autotuning followed by verification via a performance report (passing manufacturer specifications).

## Substrate scope of Dob3

Fifty-microlitre reaction mixtures containing 1 mM of the specified hydrazone substrate, 2 mM NADPH, 100 µM PMS, 20 µM Dob3, and 500 µM DBCO-acid (25 mM stock solution in methanol) in exchange buffer (20 mM HEPES pH 8.0, 10 mM $MgCl_2$, 50 mM NaCl, 10% glycerol) were incubated overnight. Samples were prepared using method 2. Detection of corresponding pyrazoles was accomplished using a Thermo Orbitrap IQ-X Tribrid mass spectrometer as described above. HCD fragmentation with 30% normalized collision was used. Spectra were acquired in positive ion mode. A mass window of 5 ppm was used for EICs.

## One-pot chemoenzymatic production of diazoacetone and diazomethylbenzene

Fifty-microlitre reaction mixtures containing 100 µM PMS, 20 µM Dob3 and 1 mM aldehyde (methylglyoxal or benzylidenehydrazine, 50 mM stock solutions in exchange buffer or methanol, respectively), 2 mM NADPH, 2 mM hydrazine hydrate, and 500 µM DBCO-acid (25 mM stock solution in methanol) in exchange buffer (20 mM HEPES pH 8.0, 10 mM $MgCl_2$, 50 mM NaCl and 10% glycerol) were incubated overnight. Samples were prepared using method 2. Detection of **5** and **20** was accomplished using a Thermo Orbitrap IQ-X Tribrid mass spectrometer as described above with minor modifications. The MS settings were: mass range 70–700 $m/z$.

## Anaerobic and aerobic hydrazone incubation

Fifty-microlitre reaction mixtures containing 500 µM benzylidene-hydrazine (25 mM stock solution in methanol), 100 µM PMS, 2 mM NADPH, and 500 µM DBCO-acid (25 mM stock solution in methanol) in exchange buffer (20 mM HEPES pH 8.0, 10 mM MgCl$_2$, 50 mM NaCl, 10% glycerol) were incubated for 22 h at room temperature either aerobically or anaerobically. Samples were quenched aerobically or anaerobically through addition of 70 µl of buffer, 15 µl of LC–MS grade ACN, and 15 µl of LC–MS grade methanol and then filtered (0.2 µm, VWR, nylon). Anaerobic samples were moved to the autosampler approximately 5 min prior to injection. Detection of **20** was accomplished using a Thermo Orbitrap IQ-X Tribrid mass spectrometer as described above with minor modifications. The MS settings were: mass range 70–700 $m/z$.

## pH screen of hydrazone autoxidation

Fifty-microlitre reaction mixtures containing 500 µM benzylidene-hydrazine (25 mM stock solution in methanol), 100 µM PMS, 2 mM NADPH, and 500 µM DBCO-acid (25 mM stock solution in methanol) in exchange buffer at the specified pH values (20 mM HEPES, 10 mM MgCl$_2$, 50 mM NaCl, 10% glycerol) were incubated for 22 h at room temperature either aerobically or anaerobically. Samples were prepared using method 2. Detection of **20** was accomplished using a Thermo Orbitrap IQ-X Tribrid mass spectrometer as described above with minor modifications. The MS settings were: mass range 70–700 $m/z$.

## Reporting summary

Further information on research design is available in the Nature Portfolio Reporting Summary linked to this article.

## Data availability

Metabolomics datasets are available on MassIVE (MSV000100436). All other raw LC–MS and LC–MS/MS data are available upon request, owing to the large file size. Previously published crystal structures are available in the Protein Data Bank (https://www.rcsb.org/) under accession codes 3CHH and 5HYH. Protein expression vectors are available in Addgene. All other data are available in the manuscript or Supplementary Information. Source data are provided with this paper.

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

**Acknowledgements** We thank G. Kenney for assistance with genome mining and Terra Bioforge for construction of *dob*-pDualP and *dob*-pDualP ΔDob3. Metal analysis was performed at the Northwestern University Quantitative Bulk-Elemental Information Core (QBIC). D.V.C. acknowledges funding from the National Science Foundation Graduate Research Fellowship Program (grant numbers DGE2140743 and DGE1745303). We acknowledge financial support from the National Institutes of Health (grant number R01GM132564). E.P.B. is a Howard Hughes Medical Institute Investigator. This article is subject to HHMI's Open Access to Publications policy. HHMI lab heads have previously granted a nonexclusive CC BY 4.0 license to the public and a sublicensable license to HHMI in their research articles. Pursuant to those licenses, the author-accepted manuscript of this article can be made freely available under a CC BY 4.0 license immediately upon publication.

**Author contributions** D.V.C. and E.P.B. initiated the study. K.P. and D.V.C. designed and optimized the reactivity-based screen. K.P. and D.V.C. performed LC–MS metabolomics and identified **1** and **2**. K.P. and D.V.C. performed bioinformatics analyses and identified the gene cluster. K.P. and D.V.C. performed heterologous expression of the biosynthetic gene cluster. K.P. and D.V.C. performed biochemical characterization of DobA, DobB, DobC, DobE, DobF, DobG, DobM, DobQ, Dob2 and Dob3. K.P. performed the in vivo gene knockout. K.P. and D.V.C. designed substrate scope and chemoenzymatic experiments. K.P. performed substrate scope and chemoenzymatic experiments. K.J.Y.W. performed chemical syntheses of substrates and standards. All authors analysed and discussed the results and prepared the manuscript.

**Competing interests** The authors declare no competing interests.

**Additional information**
**Correspondence and requests for materials** should be addressed to Emily P. Balskus.

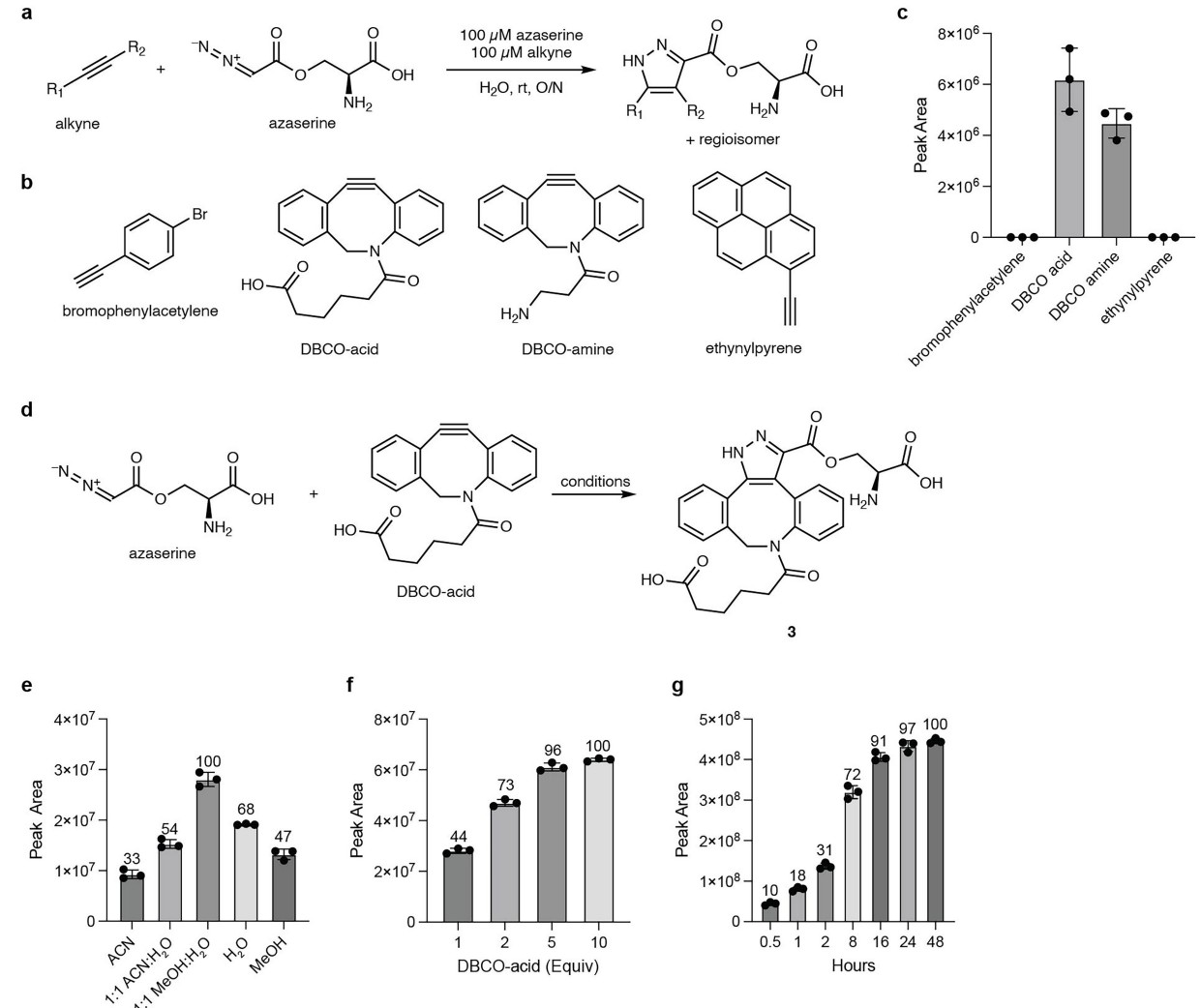

**Extended Data Fig. 1 | Azaserine undergoes cycloadditions with strained cyclooctynes, but not terminal alkynes. a**) Reaction scheme of the cycloaddition between azaserine and alkynes to yield pyrazole regioisomers. **b**) Bromophenylacetylene, DBCO-acid, DBCO-amine, and ethynylpyrene were incubated with azaserine. **c**) Product peak areas resulting from the cycloadditions. Error bars indicate mean ± standard deviation. Peak areas are calculated as the sum of the two regioisomer peaks. Experiments were run in biological triplicates. **d**) Model reaction between azaserine and DBCO-acid. **e**) Solvent optimization of the model reaction. **f**) Optimization of DBCO-acid equivalents for the model reaction. **g**) Time course of the model reaction. For all panels, values are a percentage of the maximum peak area observed. Peak areas are calculated as the sum of the two regioisomers. Error bars indicate mean ± standard deviation. Experiments were run in biological triplicates.

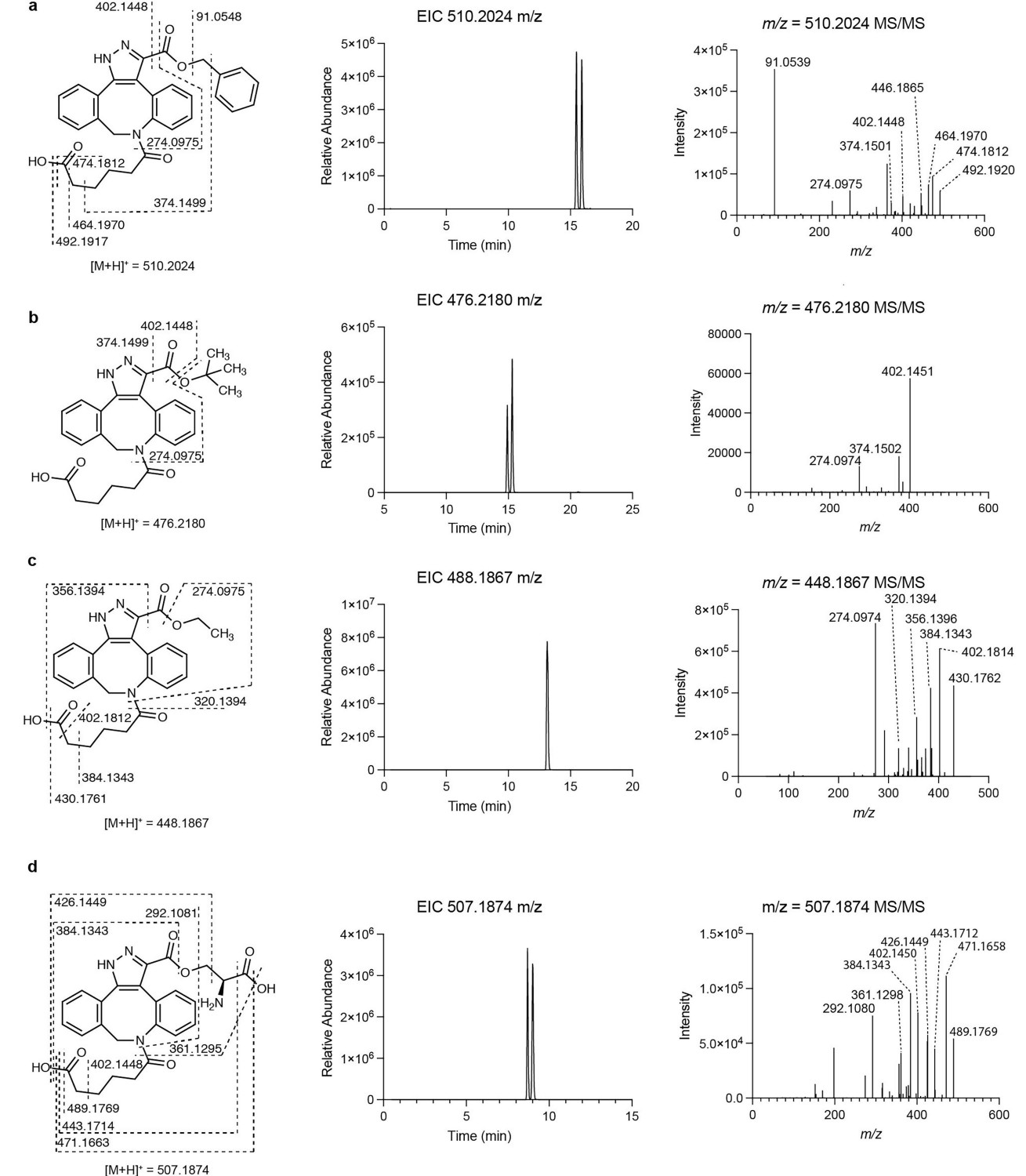

**Extended Data Fig. 2 | DBCO-acid substrate scope. a)** EIC and targeted MS/MS of the reaction of DBCO-acid with benzyldiazoacetate ($m/z$ = 510.2024 ± 5 ppm). **b)** EIC and targeted MS/MS of the reaction of DBCO-acid with *tert*-butyl diazoacetate ($m/z$ = 476.2180 ± 5 ppm) **c)** EIC and targeted MS/MS of the reaction of DBCO-acid with ethyl diazoacetate ($m/z$ = 448.1867 ± 5 ppm). We hypothesize only one peak is observed in the EIC due to the two regioisomers having identical retention times or the formation of one regioisomer being strongly favored. **d)** EIC and targeted MS/MS of **3** ($m/z$ = 507.1874 ± 5 ppm). Experiments were performed in biological triplicates and representative results are shown.

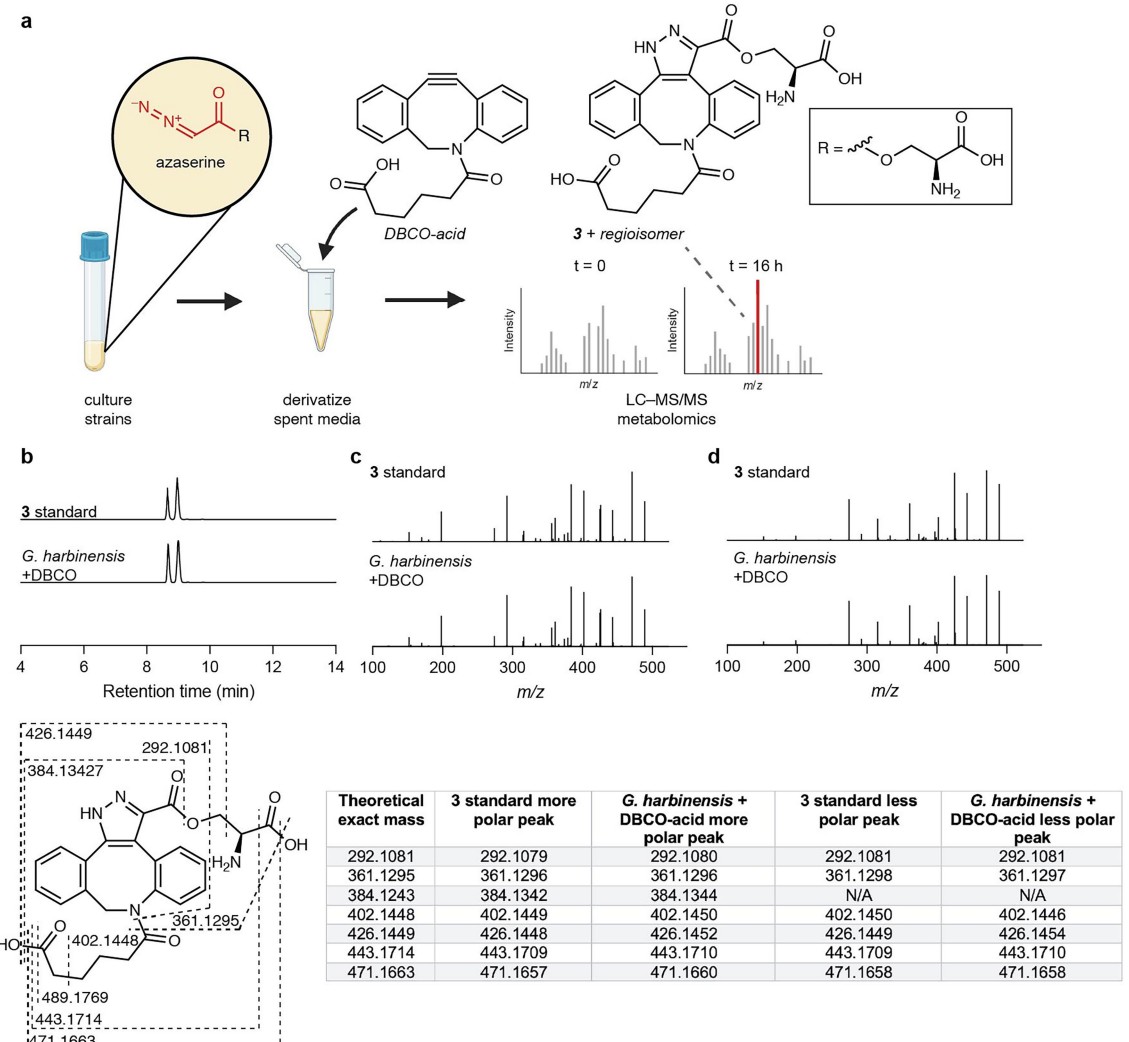

| Theoretical exact mass | 3 standard more polar peak | G. harbinensis + DBCO-acid more polar peak | 3 standard less polar peak | G. harbinensis + DBCO-acid less polar peak |
|---|---|---|---|---|
| 292.1081 | 292.1079 | 292.1080 | 292.1081 | 292.1081 |
| 361.1295 | 361.1296 | 361.1296 | 361.1298 | 361.1297 |
| 384.1243 | 384.1342 | 384.1344 | N/A | N/A |
| 402.1448 | 402.1449 | 402.1450 | 402.1450 | 402.1446 |
| 426.1449 | 426.1448 | 426.1452 | 426.1449 | 426.1454 |
| 443.1714 | 443.1709 | 443.1710 | 443.1709 | 443.1710 |
| 471.1663 | 471.1657 | 471.1660 | 471.1658 | 471.1658 |

**Extended Data Fig. 3 | DBCO-acid reacts selectively with diazo-containing natural products in complex biological matrices. a**) Reaction of DBCO-acid with azaserine in a complex metabolite matrix. DBCO-acid was incubated for 16 h with *G. harbinensis* spent medium. Adapted from a figure created in BioRender. Pfeifer, K. (2025) https://BioRender.com/vzba60o. **b**) EICs of the synthetic standard of **3** (*m/z* = 507.1874 ± 5 ppm) and **3** produced by *G. harbinensis* spent media derivatized with DBCO-acid. **c**) MS/MS fragmentation of the more polar (earlier eluting) regioisomer of the synthetic standard of **3** compared to *G. harbinensis* spent medium treated with DBCO-acid. **d**) MS/MS fragmentation of the less polar (later eluting) regioisomer of the synthetic standard of **3** compared to *G. harbinensis* spent medium treated with DBCO-acid. Experiments were performed in biological triplicates and representative results are shown.

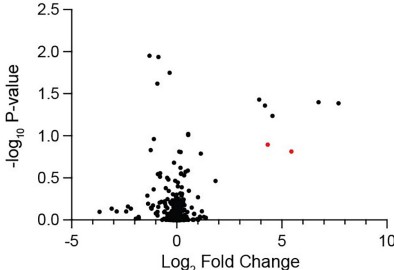

**Extended Data Fig. 4 | Volcano plot of comparative metabolomics for
*G. harbinensis* spent media.** Regioisomers of **3** are highlighted in red. P-values
are calculated by a one-way ANOVA model with Tukey as post-hoc test.

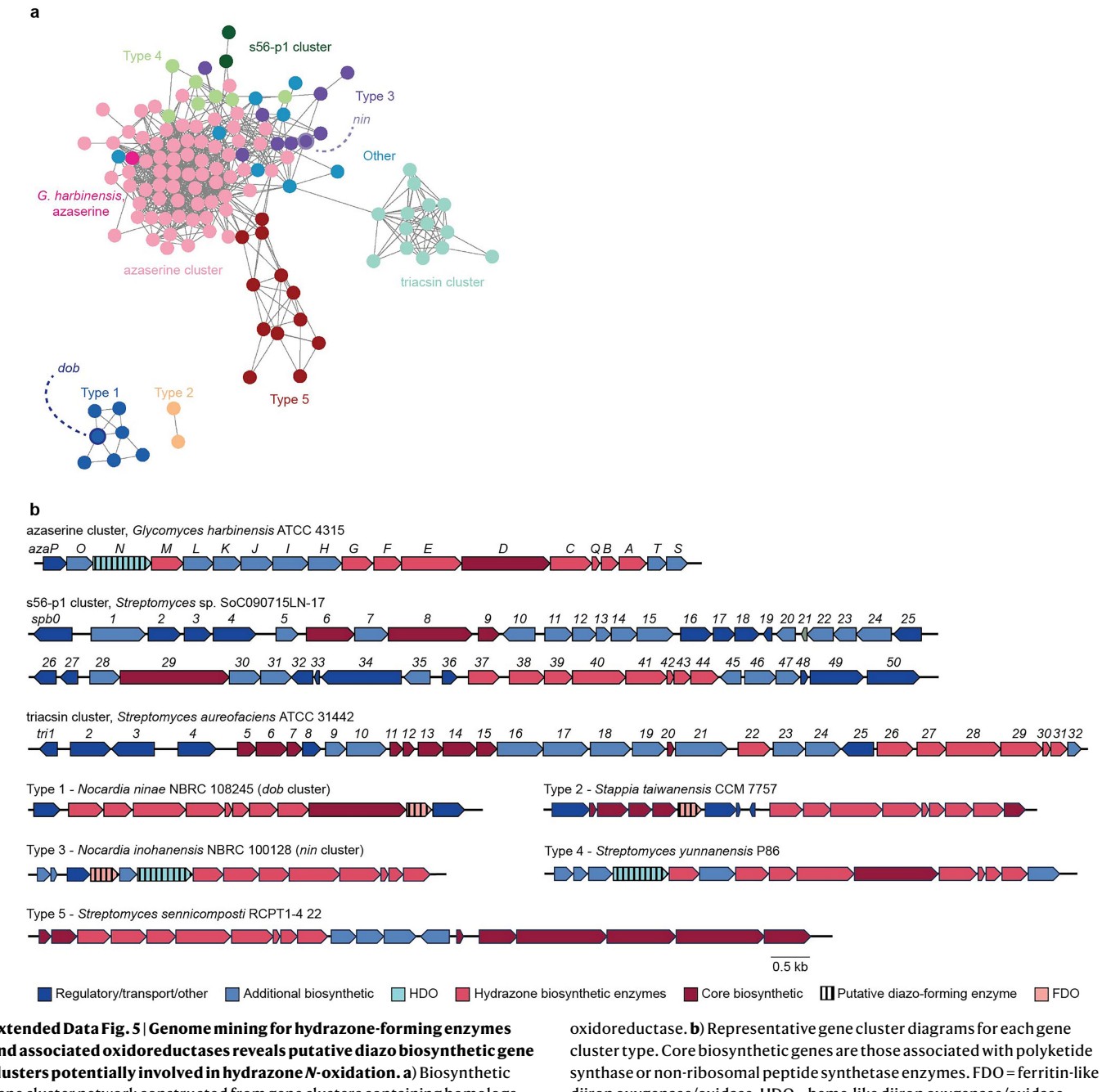

**Extended Data Fig. 5 | Genome mining for hydrazone-forming enzymes and associated oxidoreductases reveals putative diazo biosynthetic gene clusters potentially involved in hydrazone *N*-oxidation. a**) Biosynthetic gene cluster network constructed from gene clusters containing homologs of known hydrazone-forming enzymes[20–22,37] and at least one additional oxidoreductase. **b**) Representative gene cluster diagrams for each gene cluster type. Core biosynthetic genes are those associated with polyketide synthase or non-ribosomal peptide synthetase enzymes. FDO = ferritin-like diiron oxygenase/oxidase, HDO = heme-like diiron oxygenase/oxidase.

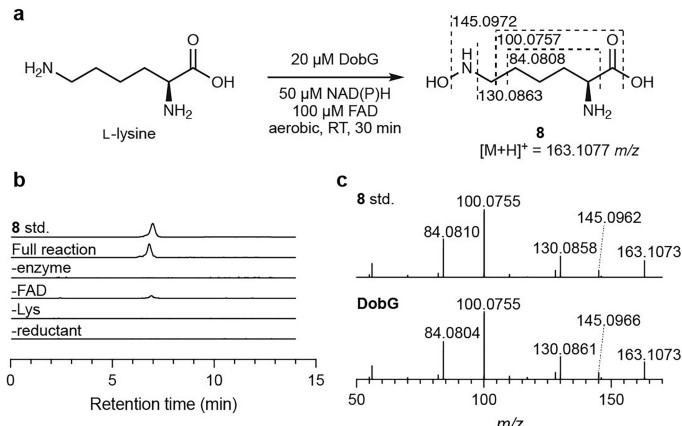

**a**

H₂N ... L-lysine

20 µM DobG
50 µM NAD(P)H
100 µM FAD
aerobic, RT, 30 min

145.0972
100.0757
84.0808
130.0863

**8**
[M+H]⁺ = 163.1077 *m/z*

**b**

8 std.
Full reaction
-enzyme
-FAD
-Lys
-reductant

0   5   10   15
Retention time (min)

**c**

8 std.

84.0810    100.0755    145.0962
           130.0858    163.1073

**DobG**

84.0804    100.0755    145.0966
           130.0861    163.1073

50    100    150
*m/z*

**Extended Data Fig. 6 | In vitro activity assays of DobG confirm its role as an L-lysine *N*-monooxygenase. a**) In vitro reaction of DobG. Calculated masses of predicted MS/MS fragments are shown. **b**) EICs (*m/z* = 163.1077 ± 10 ppm) of the DobG reaction compared to a synthetic standard of **8** and no enzyme, no FAD, no NADH/NADPH, and no L-lysine controls. **c**) MS/MS fragmentation of the 163.1077 ion from the DobG reaction matches the fragmentation pattern of the standard. Experiments were performed in biological triplicates and representative results are shown.

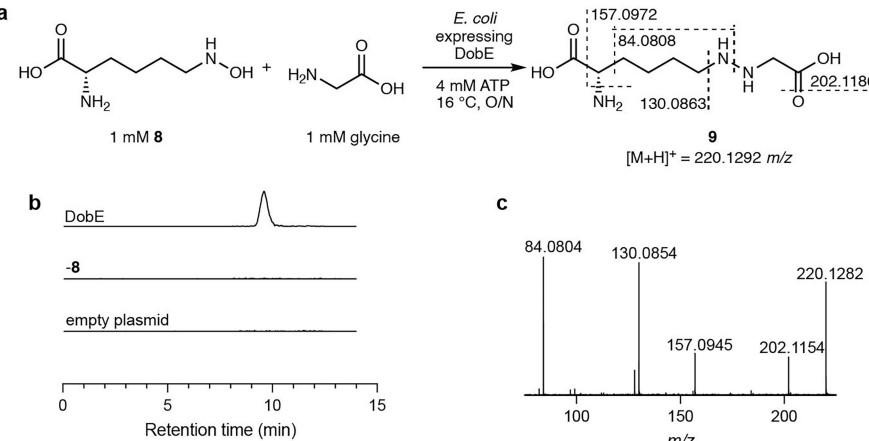

**Extended Data Fig. 7 | In vivo activity assays demonstrate that DobE catalyzes N–N bond formation between glycine and 8. a**) In vivo reaction of DobE. Calculated masses of predicted fragments are shown. **b**) EIC (*m/z* = 220.1292 ± 10 ppm) of DobE cultures compared to an empty pET28a control. **c**) MS/MS of the 220.1292 ion from the DobE reaction. Fragmentation patterns matched previously reported spectra for **9**[35]. Assays were performed in biological triplicates and representative results are shown.

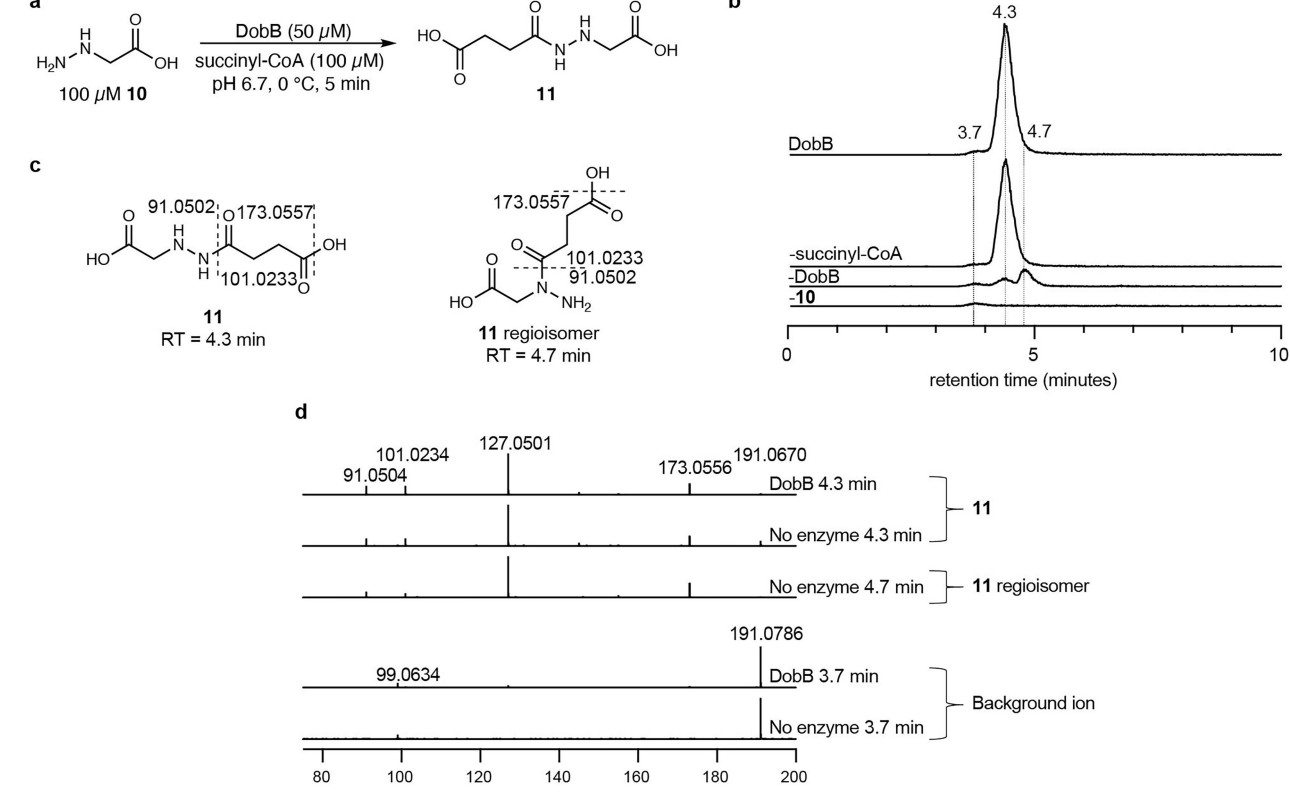

**Extended Data Fig. 8 | In vitro activity assays demonstrate that DobB catalyzes succinylation of 10. a**) In vitro reaction of DobB. **b**) EIC ($m/z$ = 191.0662 ± 10 ppm) of the DobB reaction mixture compared to no enzyme, no succinyl-CoA, and no **10** controls. Omission of succinyl-CoA did not abolish production of **11**, likely due to copurification of DobB with succinyl-CoA as previously reported for close homologs[20,37]. **c**) Putative structures of **11** and its regioisomer are shown with expected fragment masses. **d**) MS/MS fragments of the 191.0662 mass from the DobB reaction are consistent with the expected fragmentation of **11**, but do not distinguish between regioisomers. The MS/MS fragmentation patterns of the 4.3 min and 4.7 min peaks in the no enzyme control are identical to the fragmentation pattern of the AzaB reaction product. MS/MS fragmentation of the background ion peak at 3.7 min is distinct from **11**. Experiments were performed in biological triplicates and representative results are shown.

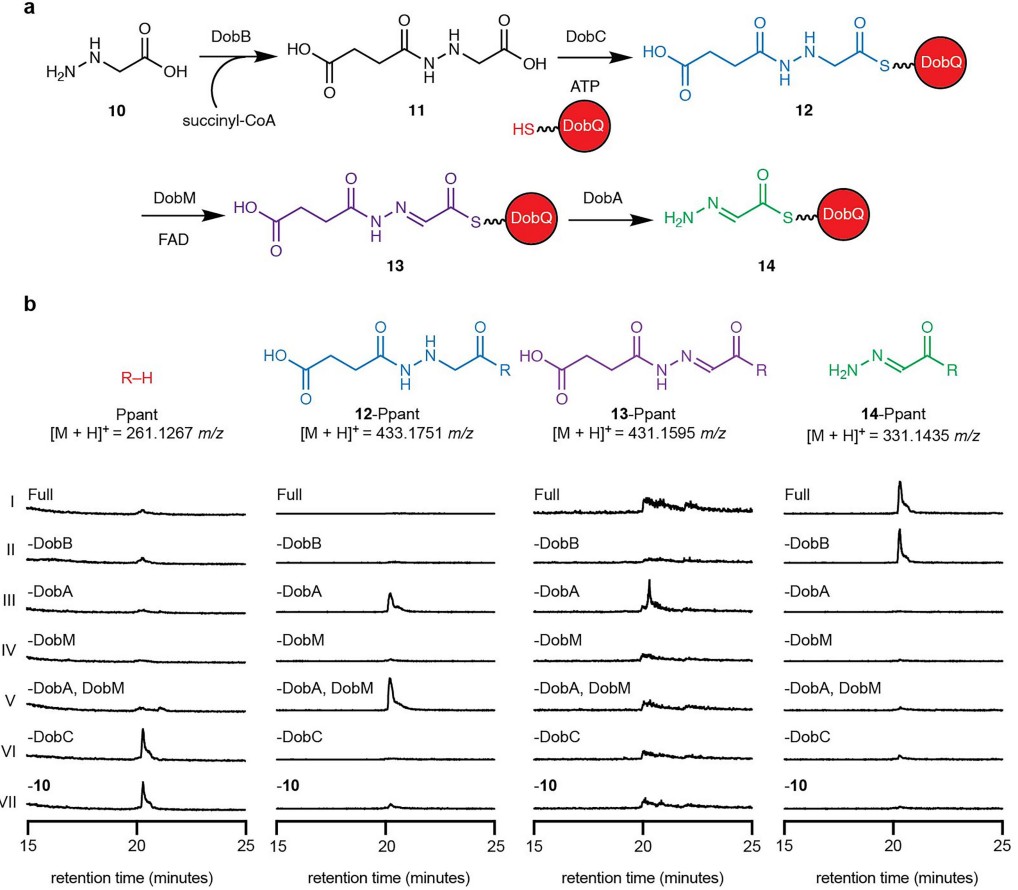

**Extended Data Fig. 9 | In vitro reactions of DobABCMQ suggest DobM may catalyze 13 oxidation prior to hydrolysis by DobA. a**) DobABCMQ cascade reaction scheme. **b**) EICs of (**I**) Full DobABCMQ reaction. (**II**) Omission of DobB did not abolish production of **14**-Ppant, likely due to non-enzymatic succinylation which was previously observed in the in vitro AzaB assay[20] (Extended Data Fig. 8). (**III**) Omission of DobA resulted in accumulation of a mass consistent with **13**-Ppant. (**IV**) Interestingly, omission of DobM resulted in no observed products.

We hypothesized that DobA may additionally act in a proof-reading capacity to hydrolyze stalled/aberrant DobQ thioester intermediates. (**V**) Omission of DobM and DobA led to accumulation of **12**-Ppant, potentially supporting this hypothesis. (**VI**) Omission of DobC or (**VII**) substrate resulted in accumulation of free Ppant. **10**-Ppant ($m/z$ = 333.1591) was not observed in any of the reactions. EICs are ± 10 ppm. Experiments were performed in biological triplicates and representative results are shown.

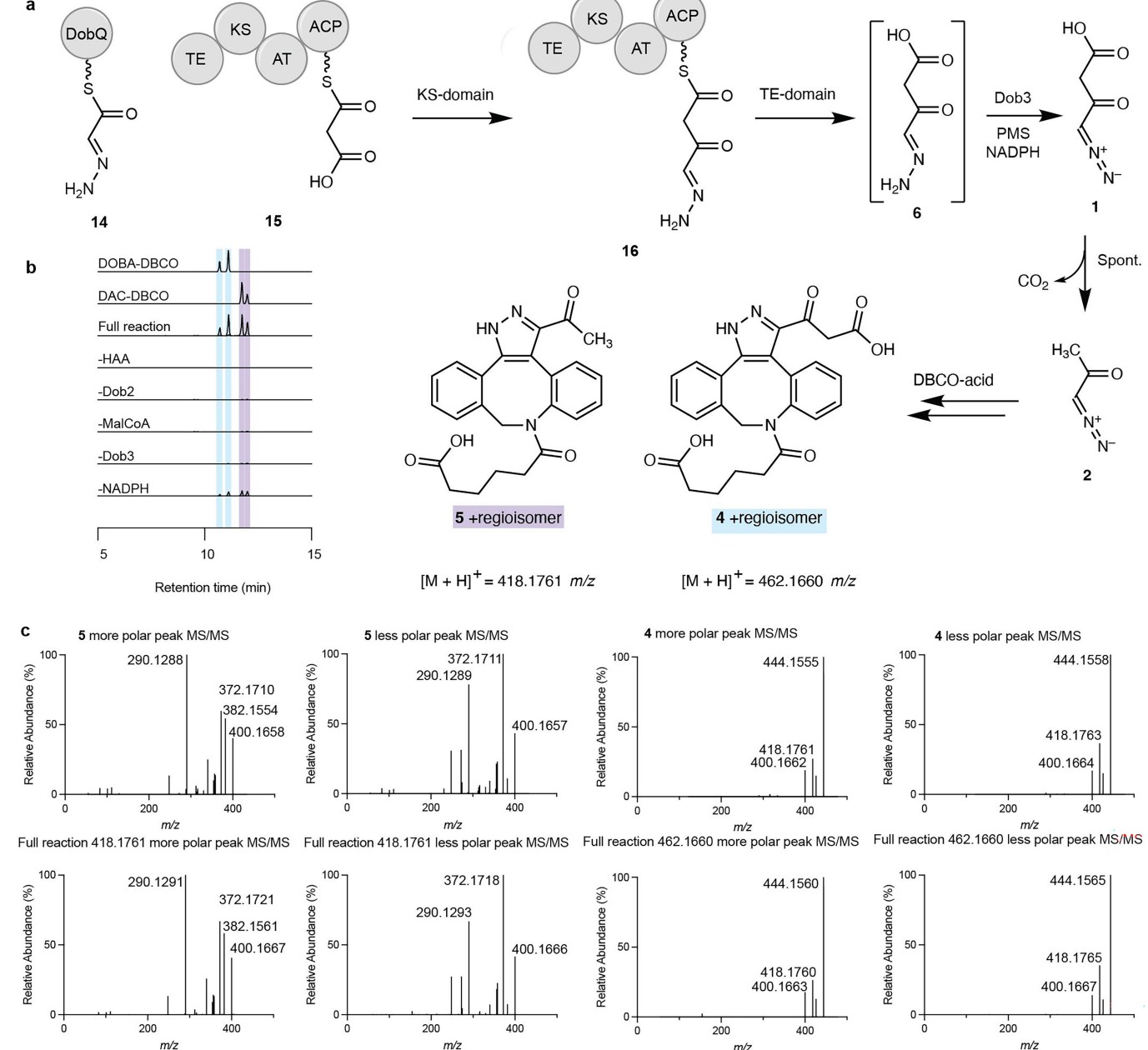

**Extended Data Fig. 10 | The enzymatic cascade reaction of DobABCMQ23 converts 10 to 1 and 2 in vitro. a)** DobABCMQ23 reaction scheme. **14** is generated enzymatically from **10** by the hydrazone biosynthetic enzymes DobABCMQ. Reaction mixtures were incubated at pH 8.0. **b)** EICs for **4** ($m/z$ = 462.1660 ± 5 ppm) and **5** ($m/z$ = 418.1761 ± 5 ppm) in 1 mM **10**, 20 µM DobB, 200 µM succinyl-CoA, 50 µM DobQ, 20 µM DobC, 5 mM ATP, 20 µM DobM, 200 µM FAD, 40 µM DobA, 10 µM Dob2, 200 µM malonyl-CoA, 20 µM Dob3, 2 mM NADPH, and 500 µM DBCO-acid as well as controls. EICs for 462.1660 and 418.1761 are superimposed. Full reaction and control data is normalized to the full reaction maximum separately for **4** and **5**. **c)** MS/MS fragmentation of **4** and **5** vs the products observed in the assay. Experiments were performed in biological triplicates and representative results are shown.

# Reporting Summary

## Statistics

For all statistical analyses, confirm that the following items are present in the figure legend, table legend, main text, or Methods section.

| n/a | Confirmed | |
|---|---|---|
| ☐ | ☒ | The exact sample size (*n*) for each experimental group/condition, given as a discrete number and unit of measurement |
| ☐ | ☒ | A statement on whether measurements were taken from distinct samples or whether the same sample was measured repeatedly |
| ☐ | ☒ | The statistical test(s) used AND whether they are one- or two-sided<br>*Only common tests should be described solely by name; describe more complex techniques in the Methods section.* |
| ☒ | ☐ | A description of all covariates tested |
| ☒ | ☐ | A description of any assumptions or corrections, such as tests of normality and adjustment for multiple comparisons |
| ☐ | ☒ | A full description of the statistical parameters including central tendency (e.g. means) or other basic estimates (e.g. regression coefficient) AND variation (e.g. standard deviation) or associated estimates of uncertainty (e.g. confidence intervals) |
| ☐ | ☒ | For null hypothesis testing, the test statistic (e.g. *F*, *t*, *r*) with confidence intervals, effect sizes, degrees of freedom and *P* value noted<br>*Give P values as exact values whenever suitable.* |
| ☒ | ☐ | For Bayesian analysis, information on the choice of priors and Markov chain Monte Carlo settings |
| ☒ | ☐ | For hierarchical and complex designs, identification of the appropriate level for tests and full reporting of outcomes |
| ☒ | ☐ | Estimates of effect sizes (e.g. Cohen's *d*, Pearson's *r*), indicating how they were calculated |

*Our web collection on statistics for biologists contains articles on many of the points above.*

## Software and code

Policy information about availability of computer code

| | |
|---|---|
| Data collection | Data collection was performed using the following commercially available software: Thermo Xcalibur 4.5 was used for analysis using the Thermo Orbitrap IQ-X. Agilent MassHunter Workstation Data Acquisition Version 10.1 was used for analysis performed using the QTOF. Agilent ICP-MS MassHunter 4.5 version C.01.05 was used for ICP-MS analysis as well as ThermoFisher QTEGRA version 2.10 (2.10.4345.64) software. Agilent Carey UV Workstation 1.3.4 was used for UV-vis analysis. Cytiva UNICORN 7.7 was used for fast protein liquid chromatography and size exclusion chromatography. Azure biosystems version 1.8.0.1230 was used for gel imaging. Bruker TopSpin 4.4.0 was used to acquire NMR spectra. |
| Data analysis | Data analysis was performed using the following commercially available software: Compound Discoverer 3.3.1.111 was used for comparative metabolomics analysis. Prism 10.1.1 was used for data visualization. FreeStyle 1.8 SP2 version 1.8.63.0 was used for extracted ion chromatograph and MS/MS spectra analysis. Microsoft Excel Version 16.80 was used for calculations. PyMOL Molecular Graphics System version 3.0 was used for protein structure analysis. Agilent MassHunter Workstation Qualitative Analysis version 10.0 was used to analyze extracted ion chromatograms and MS/MS spectra. Geneious 2023.2.1 was used for bioinformatic analysis. JalView 2.11.4.1 was used for multiple sequence alignment visualization. Enzyme Function Initiative Genome Neighborhood Tool (EFI-GNT; 2019 release) was used for genomic neighborhood analysis. Cytoscape 3.10.1 was used for genome neighborhood network visualization. MeRestNova 15.0.0-34764 was used to analyze NMR spectra. RStudio version 2023.12.0+369 was used to run prettyClusters analysis. Data analysis was performed using prettyClusters tools, available on GitHub (https://github.com/g-e-kenney/prettyClusters) |

For manuscripts utilizing custom algorithms or software that are central to the research but not yet described in published literature, software must be made available to editors and reviewers. We strongly encourage code deposition in a community repository (e.g. GitHub). See the Nature Portfolio guidelines for submitting code & software for further information.

## Data

Policy information about availability of data

All manuscripts must include a data availability statement. This statement should provide the following information, where applicable:
- Accession codes, unique identifiers, or web links for publicly available datasets
- A description of any restrictions on data availability
- For clinical datasets or third party data, please ensure that the statement adheres to our policy

Genome mining was performed using the NCBI reference protein database. Raw LC–MS data and LC–MS/MS data are available upon request due to the large file size. Previously published crystal structures are available in the Protein DataBank (https://www.rcsb.org/) under accession codes 3CHH and 5HYH. All other data are available in the manuscript or Supplementary Information. Source data are provided with this paper.

## Research involving human participants, their data, or biological material

Policy information about studies with human participants or human data. See also policy information about sex, gender (identity/presentation), and sexual orientation and race, ethnicity and racism.

| | |
|---|---|
| Reporting on sex and gender | Humans were not studied and this information was therefore not collected. |
| Reporting on race, ethnicity, or other socially relevant groupings | Humans were not studied and this information was therefore not collected. |
| Population characteristics | Humans were not studied and this information was therefore not collected. |
| Recruitment | Humans were not studied and this information was therefore not collected. |
| Ethics oversight | Humans were not studied and this information was therefore not collected. |

Note that full information on the approval of the study protocol must also be provided in the manuscript.

# Field-specific reporting

Please select the one below that is the best fit for your research. If you are not sure, read the appropriate sections before making your selection.

☒ Life sciences        ☐ Behavioural & social sciences        ☐ Ecological, evolutionary & environmental sciences

For a reference copy of the document with all sections, see nature.com/documents/nr-reporting-summary-flat.pdf

# Life sciences study design

All studies must disclose on these points even when the disclosure is negative.

| | |
|---|---|
| Sample size | *Describe how sample size was determined, detailing any statistical methods used to predetermine sample size OR if no sample-size calculation was performed, describe how sample sizes were chosen and provide a rationale for why these sample sizes are sufficient.* |
| Data exclusions | No data was excluded from analysis. |
| Replication | Experiments were performed in biological triplicates to ensure reproducibility, except for ICP-MS experiments which were performed in biological duplicates due to material limitations. All attempts at replication were successful. |
| Randomization | Randomization was not required as there were no experimental groups. |
| Blinding | Blinding was not relevant as no group allocation was performed. |

# Reporting for specific materials, systems and methods

We require information from authors about some types of materials, experimental systems and methods used in many studies. Here, indicate whether each material, system or method listed is relevant to your study. If you are not sure if a list item applies to your research, read the appropriate section before selecting a response.

## Materials & experimental systems

| n/a | Involved in the study |
|-----|----------------------|
| ☒ | ☐ Antibodies |
| ☒ | ☐ Eukaryotic cell lines |
| ☒ | ☐ Palaeontology and archaeology |
| ☐ | ☒ Animals and other organisms |
| ☒ | ☐ Clinical data |
| ☒ | ☐ Dual use research of concern |
| ☒ | ☐ Plants |

## Methods

| n/a | Involved in the study |
|-----|----------------------|
| ☒ | ☐ ChIP-seq |
| ☒ | ☐ Flow cytometry |
| ☒ | ☐ MRI-based neuroimaging |

## Animals and other research organisms

Policy information about studies involving animals; ARRIVE guidelines recommended for reporting animal research, and Sex and Gender in Research

| | |
|---|---|
| Laboratory animals | The study did not involve laboratory animals. |
| Wild animals | The study did not involve wild animals. |
| Reporting on sex | As the study did not involve animals, sex was not reported on. |
| Field-collected samples | The study did not include samples collected from the field. |
| Ethics oversight | No ethical guidance was required as the work was performed with bacterial strains. |

Note that full information on the approval of the study protocol must also be provided in the manuscript.

## Plants

| | |
|---|---|
| Seed stocks | Seed stocks were not used. |
| Novel plant genotypes | No novel plant genotypes were produced. |
| Authentication | No plants were used and therefore no authentication procedures were used. |

