## [Peer Review File · Nature]

Chemical capture of diazo metabolites reveals biosynthetic hydrazone oxidation

Corresponding Author: Dr Emily Balskus

Version 0:

Reviewer comments:

Referee #1

(Remarks to the Author)

A. Summary of the key results

In this article, Pfeifer and van Cura et al. present a simple, reactivity-based screening approach using mass spectrometry and comparative metabolomics to discover diazo-containing natural products. They validated the established workflow, which employs derivatization with dibenzocyclooctyne (DBCO), using a model metabolite (azaserine) before applying it to discover two novel diazo-containing metabolites from *N. ninae*. These two new natural products, 4-diazo-3-oxo-butanoic acid (DOBA) and diazoacetone (DAC), were identified as being biosynthesized by the *dob* gene cluster. This cluster was also identified in another species, *N. tenerifensis*, which produced mass features identical to the two new compounds. To further demonstrate that this BGC encodes all the enzymes necessary for the biosynthesis of the two diazo natural products, the gene cluster was cloned into a plasmid, and corresponding enzymes were heterologously expressed in *S. coelicolor*. This verified the production of the two diazo compounds using the established DBCO derivatization assay. Next, the authors elucidated the role of the *dob* gene cluster in the biosynthesis of the two newly identified diazo compounds. They reconstituted the activity of each putative hydrazone biosynthetic enzyme, followed by the enzymes responsible for DOBA/DAC production and the diazo-forming enzyme. Based on these *in vitro* activity studies, the authors proposed a biosynthetic pathway for DOBA and confirmed that diazo formation is catalyzed by a ferritin-like diiron oxygenase (FDO). This enzyme catalyzes the unique reaction of 2-electron oxidation of the hydrazone functional group to the diazo group. Interestingly, small-substrate-scope screening revealed that the enzyme accepts structurally diverse hydrazones, highlighting its promiscuous nature.

B. Originality and significance: if not novel, please include reference

The manuscript is highly original and significant, and very impressive in the scale of the effort. It represents a significant advancement in our ability to identify unstable diazo-containing natural products in the spent culture medium of producer organisms, greatly expanding our understanding of how these diazo groups are synthesized. The established reactivity-based screening approach is a solid methodology for discovering more of these compounds in the future. This will not only increase the number of N-N bond-containing natural products known to date, but also lead to the discovery of novel enzymes for their synthesis and biocatalytic application.

Overall, we very much enjoyed reading this manuscript, and would support publication of the manuscript in Nature. In addition to our suggestions for revisions related to the experiments and data reported (outlined in Section F of this review), we have added a series of editorial recommendations in section G of this review file.

C. Data & methodology: validity of approach, quality of data, quality of presentation

The work has been carried out with state of the art methodology and the paper is well laid out. All important raw data (including NMR spectra and LC-MS chromatograms, MS/MS fragmentation, calibration curves, plasmid maps, primer sequences) are included in the SI.

D. Appropriate use of statistics and treatment of uncertainties

All experiments were performed in biological triplicates.

E. Conclusions: robustness, validity, reliability

Most of the conclusions are well drawn from the data but please also see our specific comments below to address some inconsistencies.

F. Suggested improvements: experiments, data for possible revision

Line 49 ff. and data in Figure 1C: The dataset from which Figure 1C is derived is not reported, making it unclear how the 440 biosynthetic gene clusters were identified and how the 13 clusters with known products were classified. Moreover, the figure itself shows 15 known products (11+2+1+1), while the text states that only 13 BGCs have known products. In addition, the percentage given—3.2%—does not match either value: 13 out of 440 corresponds to 2.95%, and 15 corresponds to 3.41%. Additionally, the figure caption states that previous studies used enzymes from only four diazo-containing natural products, yet the main text (Line 53) references six compounds (cremeomycin, alazopeptin, avenalumatic acid, spinamycin, azaserine, and lomaiviticin).

Conceptually, the graph is also somewhat confusing: classifying diazo BGCs by bacterial origin (e.g., host-associated, soil, aquatic) provides limited biological insight unless these categories are better defined. For example, the term “host-associated” is vague, pathogens are also host-associated, so the distinction between categories like “pathogen” and “host-associated” should be explicitly defined.

The corresponding paragraph (line 49 ff.) suggests that the gap between the 440 predicted biosynthetic gene clusters (BGCs) and the 13 known diazo-containing natural products is primarily due to the chemical instability of diazo compounds. While instability is a valid factor, this reasoning assumes a one-to-one correspondence between each BGC and a distinct, isolable diazo metabolite, which may not be the case. Some BGCs may encode redundant or homologous pathways, produce only transient diazo intermediates, or result in final products that no longer contain a diazo group (e.g., avenalumatic acid). I recommend revising the text to reflect the complexity of genome-to-metabolite predictions and to avoid overstating the link between BGC counts and chemical diversity.

All these discrepancies should be resolved.

Line 139-141: the statement “We identified biosynthetic gene clusters of interest by searching for homologs of hydrazone biosynthetic enzymes (Supplementary Figure 3) encoded alongside at least one additional predicted oxidoreductase.” appears to be inaccurate if the search refers specifically to AzaE, as indicated in Extended Data Figure 6. As far as we know, AzaE is involved in hydrazine formation, not hydrazone biosynthesis (ref 31). Same for line 208, the query used was a hydrazine-forming enzyme and probably hydrazone-forming enzymes are also contained in the cluster, but the ones used for mining were hydrazine-forming ones (AzaE) and oxidoreductases, if not mistaken. The authors should clarify this and clearly indicate in the supporting information, such as an additional table, which enzymes were used as queries. Similarly, they should add the gene cluster diagrams for the relevant cluster types used as queries to enable comparison to Extended Figure 6.

Line 154: The authors mention potential hits from *S. yunnanensis* and *N. ninae*; however, the data in Figure 2B is only for *N. ninae*. The authors should clarify what the analysis of the spent medium of *S. yunnanensis* revealed and report the results, if applicable. If only *N. ninae* delivered a hit, this should also be clearly stated, as no mass features corresponding to the DBCO-acid adduct regioisomers could be observed in *S. yunnanensis*.

Lines 369 ff: Please provide a more specific description of which Dob3 variants were generated and which amino acid positions were mutated. Also, add a brief explanation of how the Dob3 variants were generated to the Methods section. Additionally, this section may require clarification. If we understand correctly, the authors expressed Dob3 in minimal media supplemented with different metals to identify Dob3's active cofactor. Why did the authors not also perform expression in minimal medium supplemented with iron or iron and manganese? For the sake of completeness, the expression of Dob3 in a minimal medium supplemented only with iron should be included, as this would help confirm the conclusion that Dob3 contains a diiron site.

Figure 4B: The Dob3 reactions were supplemented with PMS as the electron mediator. The authors should explain why PMS was chosen and whether other electron mediators were tested beforehand. Additionally, the authors should provide a brief description of the potential native electron mediator of Dob3 and whether it could be a redox partner protein encoded in the *dob* BGC.

Chemical compound synthesis and NMR spectra: In general, it might be helpful for ease of understanding if all compounds shown throughout the manuscript and SI could be numbered. This would particularly help clarify which NMR spectra in the SI refer to which synthesis, as outlined in the Methods section. The Methods section should properly reference the NMR spectra shown in the SI (with reference to figure number) where the authors refer to the NMR data. The NMR spectra in the SI should be in the same order as the compounds in the Methods section. Also, please add the missing NMR spectra of N6-hydroxy-L-lysine and HAA to the SI.

Furthermore, the NMR spectra of ethyl (2-diazoacetyl)glycinate are shown in the SI; however, it is unclear whether they derive from a chemical synthesis, a commercial source, an enzymatic synthesis, or isolation from a producer organism. Please clarify.

Also, please indicate more explicitly the origin of the starting material used for the chemical synthesis of the reference compounds described in the Methods section (e.g., whether they derive from chemical synthesis, a commercial source, enzymatic synthesis, or isolation from a producer organism).

G: Clarity and context: lucidity of abstract/summary, appropriateness of abstract, introduction and conclusions / References

Overall, the manuscript is well-written and scientifically sound. To improve readability further and correct some minor

editorial errors, we have listed them below for the authors' review.

Lines 19-21: We suggest replacing the second mention of "Dob3" with the possessive pronoun "its" thereby avoiding redundancy while maintaining grammatical coherence. The paragraph will read like this:

"Biosynthetic investigations revealed a distinct enzymatic logic for diazo formation involving hydrazone oxidation catalyzed by the metalloenzyme Dob3, and its biochemical characterization of Dob3 suggests promising future applications in biocatalysis".

Line 26: The standard line representation of diazo group is $R_1R_2C=N^+=N^-$ that clearly shows charge separation and bonding context. The authors should change this in line 26 and use this notation consistently throughout the manuscript.

Lines 37–47: Here the authors describe three enzymatic strategies for diazo formation, two of which have been biochemically characterized, and a third that remains hypothetical. The reference to Figure 1B appears immediately after the first strategy is introduced (line 40), even though the figure includes all three. It would be more appropriate to place this citation after the third strategy is described. Similarly, a reference to the Supplementary Figure 1 is made in line 42 following the mention of only the second strategy. This initially suggests that the figure focuses on that specific pathway, but it actually includes all three, creating some confusion. Overall, the paragraph could be revised to clarify which figure illustrates which strategies and to avoid repetition between the main and supplementary figures.

Line 39 and 40: In the sentence, "In the first, ATP-dependent ligases catalyze diazotization of primary amines using enzymatically produced nitrite (Figure 1B).^{14,25,29}", one of the cited references (ref. 25) refers to AzpL, which has been shown to catalyze diazotization in vivo, but differs mechanistically from the others. AzpL is a membrane-bound protein, not classified as a ligase, and no ATP dependence has been demonstrated. Therefore, the sentence should be revised to accurately reflect the different mechanisms of this group of enzymes and the inclusion of AzpL in the associated figure should also be re-evaluated for consistency. Please also revise Supplementary Figure 1 accordingly (please also see our later comment). Additionally, please ensure that all enzymes cited in this context are correctly classified as ligases and that their ATP dependence is supported by the literature.

Consider also including Pzm18 and NapB4, other putative diazo-forming enzymes that have not been mentioned during the paper.

Pzm18 reference:

Liu, W.; Ma, L.; Zhang, L.; Chen, Y.; Zhang, Q.; Zhang, H.; Zhang, W.; Zhang, C.; Zhang, W. Two New Phenylhydrazone Derivatives from the Pearl River Estuary Sediment-Derived *Streptomyces* Sp. SCSIO 40020. *Mar Drugs* 2022, 20 (7), 449, DOI: 10.3390/md20070449

NapB4 references:

Waldman, A. J.; Pechersky, Y.; Wang, P.; Wang, J. X.; Balskus, E. P. The Cremeomycin Biosynthetic Gene Cluster Encodes a Pathway for Diazo Formation. *ChemBioChem*. 2015, 16 (15), 2172–2175, DOI: 10.1002/cbic.201500407

Winter, J. M.; Jansma, A. L.; Handel, T. M.; Moore, B. S. Formation of the Pyridazine Natural Product Azamerone by Rearrangement of an Aryl Diazoketone. *Angew. Chem., Int. Ed. Engl.* 2009, 48 (4), 767–770.

Winter, J. M.; Moffitt, M. C.; Zazopoulos, E.; McAlpine, J. B.; Dorrestein, P. C.; Moore, B. S. Molecular Basis for Chloronium-Mediated Meroterpene Cyclization: Cloning, Sequencing, and Heterologous Expression of the Napyradiomycin Biosynthetic Gene Cluster. *J. Biol. Chem.* 2007, 282 (22), 16362–16368

Angeli, C.; Atienza-Sanz, S.; Schröder, S.; Hein, A.; Li, Y.; Argyrou, A.; Osipyan, A.; Terholsen, H.; Schmidt, S. Recent Developments and Challenges in the Enzymatic Formation of Nitrogen–Nitrogen Bonds. *ACS Catal.* 2025, 15 (1), 310–342, DOI: 10.1021/acscatal.4c05268

Figure 1B: In Figure 1B, under the "Diazotization" section, the compounds shown for each enzyme pathway are inconsistent with respect to what stage of the biosynthetic route is depicted. The figure legend (and Line 40) states that this section highlights diazo-containing products formed by the enzymes cited in references 14 and 25–29. For AzpL (ref 25, alazopeptin pathway), the correct intermediate 6-diazo-5-oxo-norleucine (DON) is shown, which is indeed the first compound to display the diazo group in the pathway. However, for AvaA6, the final natural product avenalumic acid is shown instead of the actual diazotized intermediate 3-diazoavenalumic acid (3-DAA). This is misleading, as avenalumic acid is not a diazo-containing compound—its diazo group is lost via N_2 elimination catalyzed by AvaA7. A similar issue is present for CmaA6, where p-coumaric acid is shown, even though the enzyme generates the intermediate 3-diazocoumaric acid (3-DCA). Again, the final product lacks a diazo group. In the case of SpiA7 (spinamycin pathway), the final product of the route (spinamycin) does retain the N-N bond but not as diazo but as a hydrazide moiety. We recommend the authors standardize what is depicted in this section of the figure: either consistently show (i) the first diazotized intermediate formed by each enzyme, or (ii) the final natural product of the route (iii) the name of the diazotases. This would greatly improve clarity and prevent confusion.

Line 42: missing punctuation after (Supplementary Figure 1). Phrase should read like this "(Supplementary Figure 1).³⁰ A third strategy...".

Line 50-53: The sentence has some inaccuracies and could be improved like this: "Compilation of previous genome mining efforts and our own searches has have identified 440 biosynthetic gene clusters encoding putative potentially responsible for the production of diazo-containing metabolites in diverse bacteria, including human, plant, and animal pathogens." The use of "identified" with "compilation" is not correct as compilations cannot perform actions. Also, biosynthetic gene clusters do not "encode metabolites"; rather, they encode enzymes or pathways. I recommend rephrasing for clarity and

accuracy.

Line 93: a comma is missing after “e.g.”. For correct punctuation, it should read: “e.g., UV absorption, ionization efficiency...”. Please revise accordingly.

Line 103: include reference to papers related to azaserine.

Line 110: the time course experiment was performed with 8- and 24-hour time points (75% and 97%), but the 16-hour time point is missing from Extended Figure 2D. Since the authors selected the 16-hour time point for future experiments, it is important to include this time point in the graph.

Line 109: the reference to “Extended Data Figure 2A–D” is unnecessarily detailed if the intention is to refer to the full figure. As done correctly for other extended data figures in the manuscript, it is sufficient to cite it simply as “Extended Data Figure 2.” Please revise for consistency.

Extended Data Figure 4: the azaserine structure includes an undefined “R” group. Please clarify what “R” represents in the figure or maybe the full structure should be depicted, as this experiment shows specifically the reaction between DBCO-acid with azaserine.

Line 134-136: “To prioritize organisms for screening, we mined bacterial genomes in the National Center for Biotechnology Information database for biosynthetic gene clusters encoding putative diazo-containing natural products.” This phrase is conceptually inaccurate. Biosynthetic gene clusters encode enzymes, not metabolites. We recommend rephrasing to reflect that these clusters are predicted to be involved in the biosynthesis of diazo-containing natural products, rather than encoding the compounds themselves.

Supplementary Figure 6: in line 143 and 144 the authors wrote “identified five conserved biosynthetic gene cluster architectures in addition to the known azaserine, triacsin, and s56-p1 gene clusters” but this figure only shows these 5 new architectures. It would be helpful to include the cluster diagrams of azaserine, triacsin, and S56-p1 to facilitate comparison.

Lines 153 and 618: There is a discordance between the subject “comparative metabolomics” and the verb in:

- Line 153 “and LC–MS-based comparative metabolomics was performed in...”

- Line 618: “Comparative metabolomics was performed using a...”

Plural form of the verb should be used as done in line 546: “Comparative metabolomics for t0 and t16 samples were performed...”. If the authors prefer to use the singular form of the verb, it could be phrased as comparative metabolomics analysis/experiment.

Line 157: the referral to “Supplementary Data” appears incomplete or ambiguous. It likely refers to Supplementary Figure 4, but this should be explicitly clarified in the text to avoid confusion.

Lines 174, 309 and 391: the Latin phrase *in situ* appears in italics in Line 174 but in regular font in Lines 309 and 391. Please standardize its formatting throughout the manuscript, using italics in accordance with common scientific style conventions.

Line 195: There are several inconsistencies in the use of species abbreviations throughout the manuscript.

· *Nocardia tenerifensis* appears first in Line 195 already abbreviated as *N. tenerifensis*, but is written out in full in Lines 197 and 228. To follow standard conventions, the full name should be used at first mention (Line 195), and abbreviated thereafter.

· Similarly, *Nocardia ninae* is introduced in full in both the abstract (Line 198) and the main text (Line 155), but later appears inconsistently abbreviated—e.g., abbreviated in Lines 192 and 195, but written in full in Line 187 within the same paragraph in the caption of Figure 2.

· *Streptomyces coelicolor* is correctly written out at first mention (Line 198), but appears again in full in the caption of Figure 3 (Lines 346 and 347) where the abbreviation *S. coelicolor* should be used. Please revise these instances for consistency with standard taxonomic abbreviation practices.

Line 257: Here, the authors refer to putative hydrazone biosynthetic enzymes previously described in the triacsin and azaserine pathways. It is unclear why this reference is made here, and should be further clarified.

Line 330: “C–C bond-formation” the hyphen is incorrectly placed. “Bond formation” is a noun phrase and should not be hyphenated in this context. Please revise accordingly.

Line 437: In the sentence “Previously characterized diazo-containing metabolites have potent biological activity and may play important biological roles for producing microbes,” the use of “for” could be confusing as it sounds like the metabolites are producing the microbes.

I would suggest writing “in the” or “for the microbes that produce them”.

Line 440: the use of diazo here is inaccurate, could be improved to “containing putative biosynthetic gene clusters involved in diazo-containing natural products”.

Line 441: the microbes are not exactly “infections of plants”, it could be improved by changing it to “infecting plants”.

Line 456: the expression “en route to” does not sound properly chosen in the context. We would suggest changing to “in the synthesis of”.

Line 492: The description of plasmid purification contains a redundancy or typographical error (“purification of recombinant DNA purification of recombinant plasmids”) that should be corrected for clarity.

Lines 493-495: In the sentence describing the use of restriction enzymes, the authors write that “digests were performed.” This phrasing could be imprecise as in molecular biology, “digest” refers to the product of a digestion reaction, whereas “digestion” refers to the process itself. Since the authors are describing the procedure they carried out, the correct term is “digestion”.

Line 535: “Comparative Metabolomics” is capitalized, whereas in the rest of the manuscript, this term appears in lowercase.

Lines 540 and 541: The term *m/z* is correctly italicized throughout most of the manuscript, but appears in regular font in Lines 540 and 541. Please revise these instances to ensure consistent formatting.

Line 603: In the sentence describing mass spectrometric detection, the verb agreement is incorrect. “Detection” is a singular noun and should be followed by “was accomplished,” not “were accomplished”. The object of detection being singular or plural does not determine the conjugation of the verb as detection is a singular noun that is always accompanied by “was”. There is a correct example of use of detection followed by a plural noun in line 572. Also multiple correct examples of the use of detection followed by a singular noun in lines 581, 588 and 596. Please revise the sentence accordingly to ensure proper subject–verb agreement.

Lines 613, 650, 663, 769 and 917: The term “max speed” is too colloquial for a scientific manuscript. Please replace it with the formal alternative “maximum speed”. Alternatively, if you prefer to use the abbreviated form “max speed,” it should be written in full at first mention and abbreviated consistently thereafter.

Line 615: the phrase “Gene metadata and neighbor sequences and metadata” includes a redundant repetition of the word “metadata.” Please revise for clarity, for example: “Gene and neighbor sequence metadata were generated...” or a similar rephrasing to avoid redundancy.

Lines 695 and 696: promoter names should be italicized following standard genetic nomenclature. For example, *PnitA* should be written in italics as “*PnitA*” as seen in literature. Additionally, according to reference 95, “Potr” is not a promoter but a synthetic inducible expression system, which includes multiple components, including the promoter *otrBp*. The manuscript refers to Potr as a promoter, which is confusing. Please clarify whether the full Potr system or only the *otrBp* promoter was cloned into the pDualP plasmid, and revise the terminology accordingly for accuracy and clarity.

Line 697: the word “*dob*” in “*dob* gene cluster” should be italicized to follow standard formatting for gene or gene cluster names, as is correctly done in Line 698. Please revise for consistency.

Line 712: the sentence “Transformant colonies were picked and plasmid DNA extracted” is missing the auxiliary verb “was” before “extracted.” For correct parallel structure, it should read: “Transformant colonies were picked and plasmid DNA was extracted.” Please revise accordingly.

Line 713: “Whole Plasmid Sequencing” is capitalized, while in Line 701 it appears in lowercase. Please revise for consistency.

Line 790: “Gibson assembly” should be capitalized as “Gibson Assembly” to reflect the proper name of the method. This is already done correctly in Lines 235, 496, and 698. Please revise for consistency.

Line 793-794: the phrase “Plasmids encoding the *dobQ* (carrier protein) and *dob2* (PKS) inserts were transformed...” Plasmids cannot encode “gene inserts”, they encode “proteins” and carry/contain “gene inserts”. We recommend revising this to either: “Plasmids carrying the *dobQ* and *dob2* inserts were transformed...” or “Plasmids encoding the *DobQ* and *Dob2* proteins were transformed...” for clarity and grammatical accuracy.

Lines 801 and 839: the phrase “cold-shocked on ice” appears once with a hyphen and once without. As this is a compound verb, the hyphenated form “cold-shocked” is correct and should be used consistently throughout the manuscript. Please revise for consistency.

Line 808: the use of the verb “nutate” is unusual and may be unclear to many readers. We recommend replacing it with a more standard and widely understood alternative such as “incubated with gentle rotation” for clarity.

Line 830 and 862: the phrase “biological triplicate” should be corrected to “biological triplicates” to reflect proper plural agreement when referring to three independent biological samples. It is also present in the caption of several figures in the Supplementary Information. Please revise accordingly.

Line 851: a space is missing in “3kDa.” For clarity and correct formatting of units, it should be written as “3 kDa.” Please

revise accordingly.

Line 916: “room temp” is used as an abbreviation, while the rest of the manuscript consistently uses the full term “room temperature.” Please revise for consistency by using “room temperature” throughout.

Line 930: the phrase “HCD fragmentation with 30% normalized collision” is missing the verb, making the sentence incomplete. Please revise it to a more grammatically correct form, such as: “HCD fragmentation with 30% normalized collision energy was used.”

Lines 934 and 943: The term “one-pot” appears inconsistently formatted in the manuscript—hyphenated in Lines 405 and 408, but not in Lines 934 and 943. As “one-pot” is a compound adjective, it should be hyphenated consistently throughout. Please revise accordingly.

Line 1057 and 1071: in Line 1007, “L-selectride” is formatted correctly, but in Lines 1057 and 1071 the formatting is inconsistent.

Throughout the whole text:

in Line 1056, “dichloromethane” is written out in full, whereas the abbreviation “DCM” is used elsewhere in the manuscript. For consistency, we recommend using “dichloromethane (DCM)” at first mention and the abbreviation “DCM” thereafter.

Throughout the manuscript, there are inconsistencies in subject–verb agreement when referring to volumes greater than 1 $\mu\text{L}/\text{mL}$. In scientific writing, plural forms should be used when referencing quantities greater than one (e.g., “10 μL were added” rather than “10 μL was added”). While some instances are correctly written (e.g., Lines 569, 578, 585, and 613), several others require revision to ensure grammatical consistency and accuracy. I noted such cases in Lines 571, 579, 587, 594, 601, 610, 612, 648, 652, 661, 663, 665, 678, 682, 721, 728, 736, 738, 741, 743, 745, 754, 763, 771, 798, 910, 918, 955, and 968. A careful review and correction of these instances is recommended.

The abbreviation EICs is introduced in Line 113 for “extracted ion chromatograms,” but both the full term and the abbreviation are used alternately throughout the manuscript (e.g., “EICs” in Lines 193, 200, 342; “extracted ion chromatograms” in Lines 196, 341, 345, 520...). For consistency and clarity, we recommend using the full term at first mention, followed by the abbreviation in parentheses, and using only EICs thereafter. Please revise accordingly.

The manuscript inconsistently alternates between writing out “minutes” (e.g., Lines 823, 824, 851...) and “hours” (e.g., Lines 994, 1009, 1037...) and using their abbreviations “min” (e.g., Lines 613, 650, 663...) and “h” (e.g., Lines 110, 127, 729, 909...).

For consistency and clarity, we recommend choosing one format—typically abbreviations are preferred in scientific writing—and applying it uniformly throughout the manuscript.

The term *in vitro* is italicized in Line 705 but appears in regular font in other parts of the manuscript (e.g., Lines 65, 260, 302, 345, 820, 848, 882). As *in vitro* is a Latin expression, it should be consistently italicized throughout the manuscript in accordance with scientific style conventions (if I am not mistaken, check Nature journal rules). Same happens with “*in vivo*” and “*in vitro*” mentioned in the Supplementary Information. Please revise accordingly.

The abbreviations IPTG (e.g., Lines 802 and 839) and OD (used multiple times) are not defined upon first use. For clarity and completeness, please provide the full name at first mention—e.g., isopropyl β -D-1-thiogalactopyranoside (IPTG) and optical density (OD)—and use the abbreviation consistently thereafter. This will help ensure accessibility for a broader readership.

The manuscript uses both abbreviations (e.g., ACN, MeOH) and full names (acetonitrile, methanol) inconsistently across different sections—for instance, “ACN” appears in Lines 618, 652, and 665, while “acetonitrile” is used in Lines 517, 526, and 824. Similarly, both “MeOH” (e.g., Lines 572, 578) and “methanol” (e.g., Lines 1009, 1011, 1075) are used. We recommend standardizing the use of solvent names throughout the manuscript, using the full name at first mention with the abbreviation in parentheses (e.g., “acetonitrile (ACN)”), and then using the abbreviation consistently thereafter.

The term “UV-vis” appears with inconsistent capitalization across the manuscript; both “UV-vis” (e.g., 376, 378, 401) and “UV-Vis” (e.g., 920) are used. Please standardize the formatting throughout.

Supporting information

Supplementary Figure 1 and 2: the references to the papers do not match the content of the figures. Please review and correct the reference numbers in both figures to ensure they align with the proper citations in the manuscript.

Supplementary Figure 1B: there are two issues in this figure that require correction. First, the enzyme name “SpiED” is written with a comma between “E” and “D”, which incorrectly implies two separate enzymes. However, SpiED is a single fusion protein encoding homologues of CreE and CreD. Please remove the comma.

Second, the reaction catalyzed by SpiA7 is inaccurately depicted. Both the substrate and product structures shown are incorrect. Specifically, the molecule includes a hydroxyl group that should not be present, and the product should contain a diazonium group, not a neutral diazo. We strongly recommend redrawing this reaction using the correct structures as established in reference 28.

Supplementary Figure 1D: there is a typographical error in the name of the substrate for CmaA6. It is incorrectly written as “3-aminocoumaic acid,” missing the “r” in “coumaric.” The correct name is 3-aminocoumaric acid.

Supplementary Table 3: the plasmid name “pdualp” appears in lowercase in the last two rows referring to *S. coelicolor* strains. To maintain consistency with nomenclature used throughout the manuscript and figures, this should be corrected to “pDualP.” Please revise accordingly.

Supplementary table 4 and 5: the tables list accession numbers for various strains, but the source database (e.g., NCBI or another repository) is not specified. For clarity and easy accession, please indicate the origin of these accession numbers either in the table legend or a dedicated column. This clarification will help readers locate the sequences reliably.

Supplementary Table 5: there are formatting mistakes in strain names that should be corrected. In the second row, *Actinomadura* sp. J1-007 should be corrected to *Actinomadura* sp. J1-007. In the last row, *Halostreptopolyspora alba* YIM 96095 should be formatted as *Halostreptopolyspora alba* YIM 96095.

Supplementary Figure 18: there is a space missing between “homodimer.” and “A”).

References missing: please consider including some of these references to improve the completeness of the paper:

Creomeomycin:

Waldman, A. J.; Pechersky, Y.; Wang, P.; Wang, J. X.; Balskus, E. P. The Creomeomycin Biosynthetic Gene Cluster Encodes a Pathway for Diazo Formation. *ChemBioChem*. 2015, 16 (15), 2172–2175.

Lomaiviticin:

Kersten, R. D.; Lane, A. L.; Nett, M.; Richter, T. K. S.; Duggan, B. M.; Dorrestein, P. C.; Moore, B. S. Bioactivity-Guided Genome Mining Reveals the Lomaiviticin Biosynthetic Gene Cluster in *Salinispora Tropicana*. *ChemBioChem*. 2013, 14 (8), 955–962.

Discovery of the nitrous acid biosynthetic pathway (worth mentioning in Line 40):

Sugai, Y.; Katsuyama, Y.; Ohnishi, Y. A Nitrous Acid Biosynthetic Pathway for Diazo Group Formation in Bacteria. *Nat. Chem. Biol.* 2016, 12 (2), 73–75.

Referee #2

(Remarks to the Author)

I co-reviewed this manuscript with one of the reviewers who provided the listed reports.

Referee #3

(Remarks to the Author)

The MS focusses on the biosynthesis of unusual diazo functionalised natural products. The diazo group is quite well known as a naturally occurring motif, and biosynthetic pathways are known, for example to the canonical compound azaserine. The initial parts of the paper try to make the point that diazo compounds are rather unstable, and therefore that special methods are needed to capture and study them. This is not a very convincing argument, because diazo compounds are common synthetic reagents - it is possible to buy diazoacetone as a commercial material, so it is unlikely that the low titres of such compounds are due to the instability of the diazo group per se. It is more likely that the compounds are simply produced in low titre. It would have been useful throughout to estimate the titres of the detected compounds - this would not be hard to do given that the authors have materials in hand. However, having said this, low titre diazo compounds, especially with the low molecular weights of diazoacetone and diazoacetate will be challenging to isolate, and so the use of a specific molecular 'bait' to capture compounds from complex milieu is a sensible idea. This follows a well-trodden path where molecular hooks have been used in many other cases to capture biosynthetic intermediates. Here, those hooks are the well-known strained alkynes used in Huisgen dipolar cycloaddition click reactions. Thus, I don't think that the capture and analysis methodology is particularly novel. However, these tools do allow the characterisation of a new pathway to biological diazo compounds, and it is here that the work has its key interest.

The genome mining and coupling to metabolomic analysis is very well done and the logic and data is clear. The combined use of heterologous expression in *S. coelicolor* and then targeted in vitro assays convincingly illustrates the new chemistry, and the use of Dob3 as a potential catalyst for the production of more 'synthetic' diazo compounds is a useful addition to the work. The story is logical, but I found it quite wordy - there is a lot of justification for experiments that could be cut - just explain what was done and what was observed.

Overall I feel that the key interest here is the new pathway to diazo compounds and characterization of the biosynthetic steps and associated genes and BGCs. The reactivity based screening is less novel. Nevertheless this paper contains significant interesting material for many working in the field of biosynthesis, and I feel it will be read and well cited.

Other points:

1. I (and Google) could not find reference 51...

2. ¹H and ¹³C NMR spectra for all synthetic compounds should be assigned - are these known compounds? if so then citation of relevant literature should be given. If new, then the NMR spectra should be shown.

3. NMR data for hydroxy lysine and HAA should be given.

Version 1:

Reviewer comments:

Referee #1

(Remarks to the Author)

We have read both present and previous versions of the manuscripts and the authors' responses to the reviewers. The authors have done an excellent job in carefully revising the manuscript according to all Reviewers' comments. Much effort has been devoted to address our comments and other reviewers' (Reviewer 3). The authors' responses are reasonable, and with the additional data and the changes to the main manuscript the quality of the manuscript has significantly improved. We only identified some last very minor editorial issues that should be addressed when preparing the final version of the manuscript. These are outlined below:

Line 195, main manuscript: *Nocardia tenerifensis* is always abbreviated, first appearance in line 208 should not be.

Line 426 in Supplementary info: 3k Da should be 3 kDa, the space should be added before k, not after.

Throughout the whole text: there are inconsistencies in subject–verb agreement when referring to volumes greater than 1 $\mu\text{L}/\text{mL}$. In scientific writing, plural forms should be used when referencing quantities greater than one (e.g., “10 μL were added” rather than “10 μL was added”). While some instances are correctly written several others require revision to ensure grammatical consistency and accuracy. A careful review and correction of these instances is recommended. Some examples are mentioned below:

Line 210 in supplementary info: 200 μL of spent culture medium was spun down at maximum speed for 5 min.

Line 258: 2 mL of 10% glucose was added

Supplementary Figure 1: The references are incorrect, and the numbering is not done correctly. Please revise accordingly.

Referee #2

(Remarks to the Author)

I co-reviewed this manuscript with one of the reviewers who provided the listed reports.

Referee #3

(Remarks to the Author)

The MS has been significantly improved and the authors have addressed the large majority of the comments from the first round of review. My view now is 'publish after minor revisions':

1. The title should perhaps be changed to specify alpha-diazoacetyl metabolites because these are the only ones to have been captured and analysed here - the authors themselves mention that the capture process works best for alpha-diazo-carbonyls (line 120 ish), but no other types of diazo natural products have been demonstrated yet as targets.
2. line 41 - remove the sentence "A potential exception....protein expression" - this does not seem to add any useful information.
3. line 55 - 469 BGCs were found in diverse bacteria.... It would be useful to explain if only bacterial genomes were examined, or if genomes from other kingdoms were examined, but hits were only found in bacteria.
4. Fig 1C - Insert 'bacterial' between 'Predicted' and 'biosynthetic'.
5. line 95/97 replace at least one of the 'leverages' with 'using'...
6. Line 269 - the data in Supplementary Figs 14-18 is essential to the story and it would be better to include this information in the actual paper itself - it is very inconvenient for readers to have to go searching through the supplementary info for key data.
7. I don't understand the bar graphs in Fig 4D.... the vertical scale is 'Fold change vs no enzyme' which makes sense, but the second bar is No Dob3 - which is no enzyme and so by definition is 1.... so it makes no sense to include this second bar ? I guess the authors are trying to emphasise the differences for the different substrates - so it would make more sense to combine the bars from all 6 cases into a single chart.
8. line 449 replace 'diazo-' with 'alpha-diazoacetyl-'
9. line 445 add "and comparison to standard materials" at the end of the sentence

Referees' comments:

Referee #1 (Remarks to the Author):

A. Summary of the key results

In this article, Pfeifer and van Cura et al. present a simple, reactivity-based screening approach using mass spectrometry and comparative metabolomics to discover diazo-containing natural products. They validated the established workflow, which employs derivatization with dibenzocyclooctyne (DBCO), using a model metabolite (azaserine) before applying it to discover two novel diazo-containing metabolites from *N. ninae*. These two new natural products, 4-diazo-3-oxo-butanoic acid (DOBA) and diazoacetone (DAC), were identified as being biosynthesized by the *dob* gene cluster. This cluster was also identified in another species, *N. tenerifensis*, which produced mass features identical to the two new compounds. To further demonstrate that this BGC encodes all the enzymes necessary for the biosynthesis of the two diazo natural products, the gene cluster was cloned into a plasmid, and corresponding enzymes were heterologously expressed in *S. coelicolor*. This verified the production of the two diazo compounds using the established DBCO derivatization assay. Next, the authors elucidated the role of the *dob* gene cluster in the biosynthesis of the two newly identified diazo compounds. They reconstituted the activity of each putative hydrazone biosynthetic enzyme, followed by the enzymes responsible for DOBA/DAC production and the diazo-forming enzyme. Based on these *in vitro* activity studies, the authors proposed a biosynthetic pathway for DOBA and confirmed that diazo formation is catalyzed by a ferritin-like diiron oxygenase (FDO). This enzyme catalyzes the unique reaction of 2-electron oxidation of the hydrazone functional group to the diazo group. Interestingly, small-substrate-scope screening revealed that the enzyme accepts structurally diverse hydrazones, highlighting its promiscuous nature.

B. Originality and significance: if not novel, please include reference

The manuscript is highly original and significant, and very impressive in the scale of the effort. It represents a significant advancement in our ability to identify unstable diazo-containing natural products in the spent culture medium of producer organisms, greatly expanding our understanding of how these diazo groups are synthesized. The established reactivity-based screening approach is a solid methodology for discovering more of these compounds in the future. This will not only increase the number of N-N bond-containing natural products known to date, but also lead to the discovery of novel enzymes for their synthesis and biocatalytic application.

Overall, we very much enjoyed reading this manuscript, and would support publication of the manuscript in *Nature*. In addition to our suggestions for revisions related to the

experiments and data reported (outlined in Section F of this review), we have added a series of editorial recommendations in section G of this review file.

We thank the Reviewers for their recognition of the scale, novelty, and importance of the work, and for their support of publication of this manuscript in *Nature*. We appreciate the Reviewers' detailed feedback which has substantially improved the revised manuscript. We have addressed all concerns as described below.

C. Data & methodology: validity of approach, quality of data, quality of presentation

The work has been carried out with state of the art methodology and the paper is well laid out. All important raw data (including NMR spectra and LC-MS chromatograms, MS/MS fragmentation, calibration curves, plasmid maps, primer sequences) are included in the SI.

D. Appropriate use of statistics and treatment of uncertainties

All experiments were performed in biological triplicates.

E. Conclusions: robustness, validity, reliability

Most of the conclusions are well drawn from the data but please also see our specific comments below to address some inconsistencies.

We thank the Reviewers for these comments and have revised the manuscript to address them as outlined below.

F. Suggested improvements: experiments, data for possible revision

Line 49 ff. and data in Figure 1C: The dataset from which Figure 1C is derived is not reported, making it unclear how the 440 biosynthetic gene clusters were identified and how the 13 clusters with known products were classified.

We thank the reviewers for pointing out this lack of clarity. We arrived at 440 biosynthetic gene clusters by combining our own genome mining results, a compilation of previous genome mining efforts, and gene clusters previously experimentally demonstrated to be involved in diazo biosynthesis. While previous genome mining approaches queried putative diazo-forming machinery or gene clusters homologous to known pathways, our genome mining strategy uniquely targeted pathways employing a hydrazone intermediate that could be oxidized to the final diazo-containing natural product.

To clarify the origin of the biosynthetic gene clusters used to produce Figure 1C, we have now included a Source Data file for Figure 1 containing the bacterial strains, the genome accession numbers, the bioinformatic strategy used to identify the gene cluster, and the

environmental source of each strain. Additionally, to improve clarity, we have updated the Methods section in lines 144-167 of the SI to describe the previous literature searches, lines 170-174 of the SI to give an overview of our genome mining approach, and lines 199-202 of the SI to clearly describe how we arrived at the numbers used in the text and Figure 1C.

Moreover, the figure itself shows 15 known products (11+2+1+1), while the text states that only 13 BGCs have known products. In addition, the percentage given—3.2%—does not match either value: 13 out of 440 corresponds to 2.95%, and 15 corresponds to 3.41%.

We thank the reviewers for pointing out this discrepancy. While preparing the manuscript, the list of compiled clusters was updated several times, however, the main text was not updated accordingly, resulting in the observed discrepancy in numbers. Further, the 15 known natural products shown in the figure resulted from the 13 BGCs we initially identified in our literature search plus the *dob* clusters in *N. ninae* and *N. tenerifensis* that we identified in this work. For clarity, we have now removed the *dob* clusters from *N. ninae* and *N. tenerifensis* from Figure 1C.

Reviewer comments regarding additional references for diazo biosynthesis, prompted us to expand our literature search to ensure completeness. From this search, we discovered an additional 29 gene clusters involved in diazo biosynthesis, resulting in a total of 469 gene clusters: 382 gene clusters identified in previous studies and 87 gene clusters identified by our genome mining. Of these gene clusters, 20 (4.3%) are experimentally linked to diazo-containing natural products or diazo-containing intermediates. We have now updated lines 53-58 of the main text and Figure 1C to reflect the corrected values.

Additionally, the figure caption states that previous studies used enzymes from only four diazo-containing natural products, yet the main text (Line 53) references six compounds (cremeomycin, alazopeptin, avenalunic acid, spinamycin, azaserine, and lomaiviticin).

We thank the Reviewers for identifying this error. The text was intended to refer to four different types of diazo-forming enzymes—those resembling CreM, AzpL, AlpH, and the azaserine oxidoreductase—as outlined in Figure 1B, though this was unclear as previously written. For clarity, we have revised the text in the Figure 1C caption to: "Our genome mining, combined with genome mining from previous studies, revealed 469 biosynthetic gene clusters from diverse sources that are potentially involved in the biosynthesis of diazo-containing natural products."

Further, the six references were an incomplete subset of studies that we used to compile our list of previously predicted gene clusters involved in diazo biosynthesis. We have now

added the complete set of references used to compile our list to line 53 of the text as well as within the Source Data file for Figure 1.

Conceptually, the graph is also somewhat confusing: classifying diazo BGCs by bacterial origin (e.g., host-associated, soil, aquatic) provides limited biological insight unless these categories are better defined. For example, the term “host-associated” is vague, pathogens are also host-associated, so the distinction between categories like “pathogen” and “host-associated” should be explicitly defined.

We have now updated Figure 1C to more broadly define microbes as environmental or host-associated. We have defined "host-associated" to mean microbes isolated from animal or plant sources, and we have included this definition in the figure caption (lines 87-89). Our intention in categorizing microbes in this way is to draw attention to the potential of microbial diazo-containing natural products to affect host (i.e. animal or plant) biology.

The corresponding paragraph (line 49 ff.) suggests that the gap between the 440 predicted biosynthetic gene clusters (BGCs) and the 13 known diazo-containing natural products is primarily due to the chemical instability of diazo compounds. While instability is a valid factor, this reasoning assumes a one-to-one correspondence between each BGC and a distinct, isolable diazo metabolite, which may not be the case. Some BGCs may encode redundant or homologous pathways, produce only transient diazo intermediates, or result in final products that no longer contain a diazo group (e.g., avenalamic acid). I recommend revising the text to reflect the complexity of genome-to-metabolite predictions and to avoid overstating the link between BGC counts and chemical diversity.

We thank the Reviewers for raising this important point. To describe the complexity of genome-to-metabolite predictions more clearly, we have updated the text in lines 56-64 to include additional factors like poor expression, low production titers, and redundant biosynthetic pathways as possible explanations for the gap between predicted BGCs and known diazo-containing natural products.

All these discrepancies should be resolved.

We thank the Reviewers for their detailed feedback. We feel that the changes outlined above have addressed all points highlighted by the above Reviewer comments.

Line 139-141: the statement "We identified biosynthetic gene clusters of interest by searching for homologs of hydrazone biosynthetic enzymes (Supplementary Figure 3)

encoded alongside at least one additional predicted oxidoreductase.” appears to be inaccurate if the search refers specifically to AzaE, as indicated in Extended Data Figure 6. As far as we know, AzaE is involved in hydrazine formation, not hydrazone biosynthesis (ref 31).

We thank the Reviewers for pointing out the inaccurate language in Extended Data Figure 6. To clarify, our approach for identifying biosynthetic gene clusters was performed in multiple steps and included searches for enzymes beyond AzaE. First, we gathered BGCs that could potentially encode pathways for hydrazone biosynthesis (as well as pathways for the biosynthesis of other N–N bond-containing functional groups) by searching for homologs of the key N–N bond-forming enzyme in azaserine biosynthesis, AzaE. The results of this search included pathways putatively producing a variety of N–N bond-containing natural products. Next, we used the prettyClusters tool to identify Pfams of the proteins encoded by neighboring genes. Finally, we manually curated the hits to only include biosynthetic gene clusters containing Pfams consistent with hydrazone biosynthetic genes (*azaA*, *azaB*, *azaC*, *azaE*, *azaF*, *azaG*, and *azaM* homologs), as well as Pfams encoding for an additional oxidoreductase. A detailed description of this workflow can now be found in the Methods section (Supplementary Information, lines 169-202).

The original figure caption in Extended Data Figure 6 described only the first step of this workflow. We have now updated the figure caption in lines 840-842 to read "Genome mining for hydrazone-forming enzymes and associated oxidoreductases reveals putative diazo biosynthetic gene clusters potentially encoding hydrazone *N*-oxidation."

Same for line 208, the query used was a hydrazine-forming enzyme and probably hydrazone-forming enzymes are also contained in the cluster, but the ones used for mining were hydrazine-forming ones (AzaE) and oxidoreductases, if not mistaken. The authors should clarify this and clearly indicate in the supporting information, such as an additional table, which enzymes were used as queries.

To clarify, line 208 describes our multistep bioinformatic workflow (outlined above), rather than a single search for AzaE homologs. The Pfams used in this analysis are listed in the Methods section in lines 191-192 of the SI. For ease of reference, we have also now added a table (Supplementary Table 4) to the SI which contains the Pfam identifiers and the NCBI accession IDs.

Similarly, they should add the gene cluster diagrams for the relevant cluster types used as queries to enable comparison to Extended Figure 6.

We have added the azaserine, s56-p1, and triacsin BGC diagrams to Extended Data Figure 6 to enable comparison of the identified BGCs.

Line 154: The authors mention potential hits from *S. yunnanensis* and *N. ninae*; however, the data in Figure 2B is only for *N. ninae*. The authors should clarify what the analysis of the spent medium of *S. yunnanensis* revealed and report the results, if applicable. If only *N. ninae* delivered a hit, this should also be clearly stated, as no mass features corresponding to the DBCO-acid adduct regioisomers could be observed in *S. yunnanensis*.

Reactivity-guided metabolomic analysis of *S. yunnanensis* revealed a hit at $m/z = 452.2548 \pm 5$ ppm. The low abundance of this metabolite led to one of the regioisomers being below the threshold of detection for our comparative metabolomics analysis. Manual inspection of differentially produced mass features led us to observe the two putative regioisomer peaks, consistent with derivatization of a metabolite with DBCO-acid. The relatively low abundance of these peaks led to poor MS/MS spectra and challenges with initial attempts at characterization. Overall, the low abundance of this hit compared to the hits identified from *N. ninae* led us to prioritize characterization of the metabolites from *N. ninae*. Details of the *S. yunnanensis* comparative metabolomics were previously omitted due to our focus on the more abundant and well-characterized hit from *N. ninae*.

To better describe the initial hit from *S. yunnanensis*, we have now included a figure showing these data (Supplementary Figure 4) which we reference in line 167 of the main text. An explanation as to why this hit was deprioritized is included in the caption for Supplementary Figure 4.

Lines 369 ff: Please provide a more specific description of which Dob3 variants were generated and which amino acid positions were mutated. Also, add a brief explanation of how the Dob3 variants were generated to the Methods section.

We generated and attempted to express E101A, E137A, H140A, E198A, H225A, E229A, and H232A variants of Dob3, each of which resulted in insoluble protein. We have now added this information to lines 393-394 of the main text. We have also added the procedure for generating these variants to the Methods section in lines 377-383 of the SI.

Additionally, this section may require clarification. If we understand correctly, the authors expressed Dob3 in minimal media supplemented with different metals to identify Dob3's active cofactor. Why did the authors not also perform expression in minimal medium supplemented with iron or iron and manganese? For the sake of completeness,

the expression of Dob3 in a minimal medium supplemented only with iron should be included, as this would help confirm the conclusion that Dob3 contains a diiron site.

We thank the Reviewers for raising this point and agree that the original main text was unclear. We identified iron as the putative active cofactor of Dob3 by performing inductively-coupled plasma mass spectrometry (ICP-MS) on Dob3 expressed in *E. coli* grown in complex LB medium, which revealed 1.4 equiv of Fe per Dob3 monomer (Figure 4B). To probe the possibility that a metal other than Fe could be the true active cofactor, we attempted to obtain apo-Dob3 which we intended to reconstitute with various metals, including Fe. However, expression of Dob3 in minimal medium lacking metal supplementation resulted in insoluble protein. We also attempted Dob3 expression in M9 medium supplemented with only Mn based on a previous report that obtained apo metalloenzyme under these conditions (<https://www.pnas.org/doi/10.1073/pnas.2210908119>); however, this attempt also resulted in insoluble protein. Based on these results, we suspect that iron binding is required for proper Dob3 folding which is potentially supported by the insolubility of the Dob3 variants we attempted to produce.

We have now performed ICP-MS analysis on Dob3 expressed in *E. coli* grown in M9 supplemented with only Fe or Fe/Mn. This analysis revealed Fe was the most abundant metal in both conditions with 2.5 and 3.2 equiv of Fe per Dob3 monomer, respectively. We have included these data in Supplementary Figure 26 and referenced this figure in lines 395-397 of the main text. We have also updated the main text in lines 384-386 to more clearly explain that the ICP-MS data in Figure 4B comes from Dob3 expressed in cultures grown in LB medium, and the various minimal medium expression attempts were aimed at obtaining apo-Dob3 but were unsuccessful. Finally, we have included the procedure for this analysis in the Methods section in lines 480-494.

Figure 4B: The Dob3 reactions were supplemented with PMS as the electron mediator. The authors should explain why PMS was chosen and whether other electron mediators were tested beforehand.

We initially selected PMS based on the reported activity of other dimetal enzymes using the PMS/NAD(P)H redox system (e.g. CmlI, SznF). We also tested other reductants (ascorbate, DTT, and spinach ferredoxin/ferredoxin reductase) and other electron mediators (PES and methyl viologen). The best reductant was NADPH producing 650% more product area than the next best reductant (ascorbate), and 11,900% more product area than the worst reductant (DTT). The best electron mediator was PMS which was similar to the next best mediator (PES) but produced 6,800% more product area than the

worst mediator (methyl viologen). Thus, we selected PMS/NADPH as the redox system for Dob3 activity assays.

We have now added a supplementary figure with this data (Supplementary Figure 21). We have also added a brief discussion of these results to lines 325-328 of the main text.

Additionally, the authors should provide a brief description of the potential native electron mediator of Dob3 and whether it could be a redox partner protein encoded in the *dob* BGC.

We analyzed the genes within the *dob* BGC, and those within 10 kb on either side, and identified no obvious redox partners. Thus, based on this analysis we conclude that the native redox system for Dob3 is likely encoded by genes located outside of the gene cluster. Identification of this system may require further study beyond the scope of this work. We have now added a brief description of the potential native redox partner of Dob3 to the main text in lines 328-329.

Chemical compound synthesis and NMR spectra: In general, it might be helpful for ease of understanding if all compounds shown throughout the manuscript and SI could be numbered.

We thank the Reviewers for this suggestion, and we have now numbered all compounds in the text and figures.

This would particularly help clarify which NMR spectra in the SI refer to which synthesis, as outlined in the Methods section.

We have now added compound numbers to the Methods section and NMR spectra.

The Methods section should properly reference the NMR spectra shown in the SI (with reference to figure number) where the authors refer to the NMR data.

We have now referenced the NMR spectra in the Methods section.

The NMR spectra in the SI should be in the same order as the compounds in the Methods section.

We have reordered the spectra in the SI such that they are in the same order as the compounds in the Methods section.

Also, please add the missing NMR spectra of N6-hydroxy-L-lysine and HAA to the SI.

We have added the spectra for N6-hydroxy-L-lysine (**8**) and HAA (**10**) to the SI (Supplementary Figures 40, 41, 42, and 43).

Furthermore, the NMR spectra of ethyl (2-diazoacetyl)glycinate are shown in the SI; however, it is unclear whether they derive from a chemical synthesis, a commercial source, an enzymatic synthesis, or isolation from a producer organism. Please clarify.

We mistakenly included the NMR spectra of ethyl (2-diazoacetyl)glycinate after removing the figure demonstrating the use of this compound during manuscript preparation. As we do not include any experimental results in the main text, Extended Data, or SI which involve ethyl (2-diazoacetyl)glycinate, we have removed the NMR spectra from the SI.

Also, please indicate more explicitly the origin of the starting material used for the chemical synthesis of the reference compounds described in the Methods section (e.g., whether they derive from chemical synthesis, a commercial source, enzymatic synthesis, or isolation from a producer organism).

All starting materials used for chemical synthesis originate from commercial sources or chemical synthesis. We have now clarified this in the Methods section in lines 23-25 of the SI.

G: Clarity and context: lucidity of abstract/summary, appropriateness of abstract, introduction and conclusions / References

Overall, the manuscript is well-written and scientifically sound. To improve readability further and correct some minor editorial errors, we have listed them below for the authors' review.

Lines 19-21: We suggest replacing the second mention of "Dob3" with the possessive pronoun "its" thereby avoiding redundancy while maintaining grammatical coherence.

The paragraph will read like this:

“Biosynthetic investigations revealed a distinct enzymatic logic for diazo formation involving hydrazone oxidation catalyzed by the metalloenzyme Dob3, and its biochemical characterization of ~~Dob3~~ suggests promising future applications in biocatalysis”.

We have edited the manuscript to make this change.

Line 26: The standard line representation of diazo group is $R_1R_2C=N^+=N^-$ that clearly shows charge separation and bonding context. The authors should change this in line 26 and use this notation consistently throughout the manuscript.

We have edited the manuscript to make this change.

Lines 37–47: Here the authors describe three enzymatic strategies for diazo formation, two of which have been biochemically characterized, and a third that remains hypothetical. The reference to Figure 1B appears immediately after the first strategy is introduced (line 40), even though the figure includes all three. It would be more appropriate to place this citation after the third strategy is described. Similarly, a reference to the Supplementary Figure 1 is made in line 42 following the mention of only the second strategy. This initially suggests that the figure focuses on that specific pathway, but it actually includes all three, creating some confusion. Overall, the paragraph could be revised to clarify which figure illustrates which strategies and to avoid repetition between the main and supplementary figures.

We have edited the manuscript to make this change.

Line 39 and 40: In the sentence, “In the first, ATP-dependent ligases catalyze diazotization of primary amines using enzymatically produced nitrite (Figure 1B).^{14, 25-29}”, one of the cited references (ref. 25) refers to AzpL, which has been shown to catalyze diazotization in vivo, but differs mechanistically from the others. AzpL is a membrane-bound protein, not classified as a ligase, and no ATP dependence has been demonstrated. Therefore, the sentence should be revised to accurately reflect the different mechanisms of this group of enzymes and the inclusion of AzpL in the associated figure should also be re-evaluated for consistency.

We thank the Reviewers for identifying this inconsistency. We have now changed the figure to reflect that while AzpL catalyzes a similar overall transformation to diazotases (namely diazotization of primary amines with enzymatically produced nitrite), it is phylogenetically distinct and not necessarily ATP-dependent. Further, we have updated the text to read, “In the first strategy, diazotization of primary amines is accomplished with enzymatically produced nitrite, typically catalyzed by ATP-dependent ligases,” which more accurately reflects the conserved diazo-forming strategy of the diazotases and AzpL. We have additionally updated lines 41-42 of the main text to include “A potential exception is AzpL which has not been experimentally demonstrated to utilize ATP due to challenges with recombinant protein expression.”

Please also revise Supplementary Figure 1 accordingly (please also see our later comment). Additionally, please ensure that all enzymes cited in this context are correctly classified as ligases and that their ATP dependence is supported by the literature.

We have edited the manuscript to make this change.

Consider also including Pzm18 and NapB4, other putative diazo-forming enzymes that have not been mentioned during the paper.

Pzm18 reference:

Liu, W.; Ma, L.; Zhang, L.; Chen, Y.; Zhang, Q.; Zhang, H.; Zhang, W.; Zhang, C.; Zhang, W. Two New Phenylhydrazone Derivatives from the Pearl River Estuary Sediment-Derived *Streptomyces* Sp. SCSIO 40020. *Mar Drugs* 2022, 20 (7), 449, DOI: 10.3390/md20070449

NapB4 references:

Waldman, A. J.; Pechersky, Y.; Wang, P.; Wang, J. X.; Balskus, E. P. The Cremeomycin Biosynthetic Gene Cluster Encodes a Pathway for Diazo Formation. *ChemBioChem*. 2015, 16 (15), 2172– 2175, DOI: 10.1002/cbic.201500407

Winter, J. M.; Jansma, A. L.; Handel, T. M.; Moore, B. S. Formation of the Pyridazine Natural Product Azamerone by Rearrangement of an Aryl Diazoketone. *Angew. Chem., Int. Ed. Engl.* 2009, 48 (4), 767– 770.

Winter, J. M.; Moffitt, M. C.; Zazopoulos, E.; McAlpine, J. B.; Dorrestein, P. C.; Moore, B. S. Molecular Basis for Chloronium-Mediated Meroterpene Cyclization: Cloning, Sequencing, and Heterologous Expression of the Napyradiomycin Biosynthetic GeneCluster. *J. Biol. Chem.* 2007, 282 (22), 16362– 16368

Angeli, C.; Atienza-Sanz, S.; Schröder, S.; Hein, A.; Li, Y.; Argyrou, A.; Osipyan, A.; Terholsen, H.; Schmidt, S. Recent Developments and Challenges in the Enzymatic Formation of Nitrogen–Nitrogen Bonds. *ACS Catal.* 2025, 15 (1), 310– 342, DOI: 10.1021/acscatal.4c05268

We thank the Reviewers for pointing out these additional references. We have now added the clusters identified in the Pzm18 and NapB4 references to our genome mining data (see Source Data for Figure 1) and have added the primary references listed above to line 53 of the text.

Figure 1B: In Figure 1B, under the “Diazotization” section, the compounds shown for each enzyme pathway are inconsistent with respect to what stage of the biosynthetic route is depicted. The figure legend (and Line 40) states that this section highlights diazo-containing products formed by the enzymes cited in references 14 and 25–29. For AzpL (ref 25, alazopeptin pathway), the correct intermediate 6-diazo-5-oxo-norleucine (DON) is shown, which is indeed the first compound to display the diazo group in the pathway. However, for AvaA6, the final natural product avenalumatic acid is shown instead of the actual diazotized intermediate 3-diazoavenalumatic acid (3-DAA). This is misleading, as avenalumatic acid is not a diazo-containing compound—its diazo group is lost via N_2 elimination catalyzed by AvaA7. A similar issue is present for CmaA6, where p-coumaric acid is shown, even though the enzyme generates the intermediate 3-diazocoumaric acid (3-DCA). Again, the final product lacks a diazo group. In the case of SpiA7 (spinamycin pathway), the final product of the route (spinamycin) does retain the N-N bond but not as diazo but as a hydrazide moiety. We recommend the authors standardize what is depicted in this section of the figure: either consistently show (i) the first diazotized intermediate formed by each enzyme, or (ii) the final natural product of the route (iii) the name of the diazotases. This would greatly improve clarity and prevent confusion.

We thank the Reviewers for identifying this inconsistency, we have now revised the schemes in Supplementary Figure 1 to show the first diazotized intermediate for each pathway. For azaserine (panel H), we have also included the steps for hydrazine and hydrazone formation to illustrate the iterative oxidation logic employed in this pathway.

Line 42: missing punctuation after (Supplementary Figure 1). Phrase should read like this “(Supplementary Figure 1).³⁰ A third strategy...”.

We have edited the manuscript to make this change.

Line 50-53: The sentence has some inaccuracies and could be improved like this: “Compilation of previous genome mining efforts and our own searches has have identified 440 biosynthetic gene clusters encoding putative potentially responsible for the production of diazo-containing metabolites in diverse bacteria, including human, plant, and animal pathogens.” The use of “identified” with “compilation” is not correct as compilations cannot perform actions. Also, biosynthetic gene clusters do not “encode metabolites”; rather, they encode enzymes or pathways. I recommend rephrasing for clarity and accuracy.

We have edited the manuscript to address these inaccuracies. The sentence now reads: “We performed a literature search to identify bioinformatically predicted and

experimentally verified biosynthetic gene clusters involved in diazo formation. By combining these results^{13,25,27–31,33–45} with our own bioinformatic search (described below), we identified 469 biosynthetic gene clusters potentially responsible for producing diazo-containing metabolites in diverse bacteria, including human-, plant-, and animal-associated strains (Source Data for Figure 1)."

Line 93: a comma is missing after "e.g.". For correct punctuation, it should read: "e.g., UV absorption, ionization efficiency...". Please revise accordingly.

We have edited the manuscript to make this change.

Line 103: include reference to papers related to azaserine.

We have edited the manuscript to make this change.

Line 110: the time course experiment was performed with 8- and 24-hour time points (75% and 97%), but the 16-hour time point is missing from Extended Figure 2D. Since the authors selected the 16-hour time point for future experiments, it is important to include this time point in the graph.

We thank the Reviewers for this suggestion and agree that the 16 h time point is important to include. We have now added the data to Extended Data Figure 2.

Line 109: the reference to "Extended Data Figure 2A–D" is unnecessarily detailed if the intention is to refer to the full figure. As done correctly for other extended data figures in the manuscript, it is sufficient to cite it simply as "Extended Data Figure 2." Please revise for consistency.

We have edited the manuscript to make this change.

Extended Data Figure 4: the azaserine structure includes an undefined "R" group. Please clarify what "R" represents in the figure or maybe the full structure should be depicted, as this experiment shows specifically the reaction between DBCO-acid with azaserine.

We have edited the figure so that the "R" group is now defined.

Line 134-136: "To prioritize organisms for screening, we mined bacterial genomes in the National Center for Biotechnology Information database for biosynthetic gene clusters encoding putative diazo-containing natural products." This phrase is conceptually

inaccurate. Biosynthetic gene clusters encode enzymes, not metabolites. We recommend rephrasing to reflect that these clusters are predicted to be involved in the biosynthesis of diazo-containing natural products, rather than encoding the compounds themselves.

We have edited the manuscript to make this change.

Supplementary Figure 6: in line 143 and 144 the authors wrote “identified five conserved biosynthetic gene cluster architectures in addition to the known azaserine, triacsin, and s56-p1 gene clusters” but this figure only shows these 5 new architectures. It would be helpful to include the cluster diagrams of azaserine, triacsin, and S56-p1 to facilitate comparison.

We have edited Extended Data Figure 6 to make this change.

Lines 153 and 618: There is a discordance between the subject “comparative metabolomics” and the verb in:

- Line 153 “and LC–MS-based comparative metabolomics was performed in...”

- Line 618: “Comparative metabolomics was performed using a...”

Plural form of the verb should be used as done in line 546: “Comparative metabolomics for t0 and t16 samples were performed...”. If the authors prefer to use the singular form of the verb, it could be phrased as comparative metabolomics analysis/experiment.

We have edited the manuscript to use the plural verb form in both instances.

Line 157: the referral to “Supplementary Data” appears incomplete or ambiguous. It likely refers to Supplementary Figure 4, but this should be explicitly clarified in the text to avoid confusion.

We have edited the manuscript to refer to Figure 2B.

Lines 174, 309 and 391: the Latin phrase *in situ* appears in italics in Line 174 but in regular font in Lines 309 and 391. Please standardize its formatting throughout the manuscript, using italics in accordance with common scientific style conventions.

In accordance with Nature’s style guide, we have standardized this formatting to regular font.

Line 195: There are several inconsistencies in the use of species abbreviations throughout the manuscript.

- *Nocardia tenerifensis* appears first in Line 195 already abbreviated as *N. tenerifensis*, but is written out in full in Lines 197 and 228. To follow standard conventions, the full name should be used at first mention (Line 195), and abbreviated thereafter.
- Similarly, *Nocardia ninae* is introduced in full in both the abstract (Line 198) and the main text (Line 155), but later appears inconsistently abbreviated—e.g., abbreviated in Lines 192 and 195, but written in full in Line 187 within the same paragraph in the caption of Figure 2.
- *Streptomyces coelicolor* is correctly written out at first mention (Line 198), but appears again in full in the caption of Figure 3 (Lines 346 and 347) where the abbreviation *S. coelicolor* should be used. Please revise these instances for consistency with standard taxonomic abbreviation practices.

We have edited the manuscript to make these changes and use the full strain name at first use and the abbreviation thereafter.

Line 257: Here, the authors refer to putative hydrazone biosynthetic enzymes previously described in the triacsin and azaserine pathways. It is unclear why this reference is made here, and should be further clarified.

In line 257, we explain the hypothesized functions of genes in the *dob* BGC. Based on homology of the *dob* genes (*dobG*, *E*, *F*, *B*, *C*, *Q*, *M*, and *A*) to genes from the azaserine and triacsin pathways (*azaG*, *E*, *F*, *B*, *C*, *Q*, *M*, and *A*; and *tri26*, *28*, *27*, *31*, *29*, *30*, *22*, and *14*), we hypothesized these genes would carry out similar functions to convert L-lysine and glycine to a carrier protein-bound α -hydrazonoacetyl thioester. We cited the previous reports to highlight this homology and we also illustrated the transformations from the azaserine and triacsin pathways in Supplementary Figure 3.

We have now changed the text in lines 259-262 to read: "Based on the homology of *DobG*, *E*, *F*, *B*, *C*, *Q*, *M*, and *A* to hydrazone-forming enzymes previously reported in the triacsin⁵¹ and azaserine^{31–33} biosynthetic pathways, we anticipated these enzymes would similarly function to produce a carrier protein-bound α -hydrazonoacetyl thioester (7) intermediate from L-lysine and glycine (**Supplementary Figure 3**)."

Line 330: "C–C bond-formation" the hyphen is incorrectly placed. "Bond formation" is a noun phrase and should not be hyphenated in this context. Please revise accordingly.

We have edited the manuscript to make this suggested change.

Line 437: In the sentence "Previously characterized diazo-containing metabolites have potent biological activity and may play important biological roles for producing

microbes,” the use of “for” could be confusing as it sounds like the metabolites are producing the microbes.

I would suggest writing “in the” or “for the microbes that produce them”.

We have edited the manuscript to this suggested change.

Line 440: the use of diazo here is inaccurate, could be improved to “containing putative biosynthetic gene clusters involved in diazo-containing natural products”.

We have edited the manuscript to make this suggested change.

Line 441: the microbes are not exactly “infections of plants”, it could be improved by changing it to “infecting plants”.

We have edited the manuscript to make this suggested change.

Line 456: the expression “en route to” does not sound properly chosen in the context. We would suggest changing to “in the synthesis of”.

We have edited the manuscript to make this suggested change.

Line 492: The description of plasmid purification contains a redundancy or typographical error (“purification of recombinant DNA purification of recombinant plasmids”) that should be corrected for clarity.

We have edited the manuscript to make this suggested change.

Lines 493-495: In the sentence describing the use of restriction enzymes, the authors write that “digests were performed.” This phrasing could be imprecise as in molecular biology, “digest” refers to the product of a digestion reaction, whereas “digestion” refers to the process itself. Since the authors are describing the procedure they carried out, the correct term is “digestion”.

We have edited the manuscript to make this suggested change.

Line 535: “Comparative Metabolomics” is capitalized, whereas in the rest of the manuscript, this term appears in lowercase.

We have edited the manuscript to make this suggested change.

Lines 540 and 541: The term *m/z* is correctly italicized throughout most of the manuscript, but appears in regular font in Lines 540 and 541. Please revise these instances to ensure consistent formatting.

We have edited the manuscript to make these suggested changes.

Line 603: In the sentence describing mass spectrometric detection, the verb agreement is incorrect. “Detection” is a singular noun and should be followed by “was accomplished,” not “were accomplished”. The object of detection being singular or plural does not determine the conjugation of the verb as detection is a singular noun that is always accompanied by “was”. There is a correct example of use of detection followed by a plural noun in line 572. Also multiple correct examples of the use of detection followed by a singular noun in lines 581, 588 and 596. Please revise the sentence accordingly to ensure proper subject–verb agreement.

We have edited the manuscript to make this suggested change.

Lines 613, 650, 663, 769 and 917: The term “max speed” is too colloquial for a scientific manuscript. Please replace it with the formal alternative “maximum speed”. Alternatively, if you prefer to use the abbreviated form “max speed,” it should be written in full at first mention and abbreviated consistently thereafter.

We have edited the manuscript to make these suggested changes.

Line 615: the phrase “Gene metadata and neighbor sequences and metadata” includes a redundant repetition of the word “metadata.” Please revise for clarity, for example: “Gene and neighbor sequence metadata were generated...” or a similar rephrasing to avoid redundancy.

We have edited the manuscript to make this suggested change.

Lines 695 and 696: promoter names should be italicized following standard genetic nomenclature. For example, PnitA should be written in italics as “*PnitA*” as seen in literature. Additionally, according to reference 95, “Potr” is not a promoter but a synthetic inducible expression system, which includes multiple components, including the promoter otrBp. The manuscript refers to Potr as a promoter, which is confusing. Please clarify whether the full Potr system or only the otrBp promoter was cloned into the pDualP plasmid, and revise the terminology accordingly for accuracy and clarity.

We thank the Reviewers for noting this inaccuracy. The full Potr system is used in the BAC. We have edited the Methods Section in lines 240-243 of the SI to reflect this.

Line 697: the word “dob” in “dob gene cluster” should be italicized to follow standard formatting for gene or gene cluster names, as is correctly done in Line 698. Please revise for consistency.

We have edited the manuscript to make this suggested change.

Line 712: the sentence “Transformant colonies were picked and plasmid DNA extracted” is missing the auxiliary verb “was” before “extracted.” For correct parallel structure, it should read: “Transformant colonies were picked and plasmid DNA was extracted.” Please revise accordingly.

We have edited the manuscript to make this suggested change.

Line 713: “Whole Plasmid Sequencing” is capitalized, while in Line 701 it appears in lowercase. Please revise for consistency.

We have edited the manuscript to make this suggested change.

Line 790: “Gibson assembly” should be capitalized as “Gibson Assembly” to reflect the proper name of the method. This is already done correctly in Lines 235, 496, and 698. Please revise for consistency.

We have edited the manuscript to make these suggested changes.

Line 793-794: the phrase “Plasmids encoding the dobQ (carrier protein) and dob2 (PKS) inserts were transformed... Plasmids cannot encode “gene inserts”, they encode “proteins” and carry/contain “gene inserts”. We recommend revising this to either: “Plasmids carrying the dobQ and dob2 inserts were transformed...” or “Plasmids encoding the DobQ and Dob2 proteins were transformed...” for clarity and grammatical accuracy.

We have edited the manuscript to make these suggested changes.

Lines 801 and 839: the phrase “cold-shocked on ice” appears once with a hyphen and once without. As this is a compound verb, the hyphenated form “cold-shocked” is correct and should be used consistently throughout the manuscript. Please revise for consistency.

We have edited the manuscript to make these suggested changes.

Line 808: the use of the verb “nutate” is unusual and may be unclear to many readers. We recommend replacing it with a more standard and widely understood alternative such as “incubated with gentle rotation” for clarity.

We have edited the manuscript to make this suggested change

Line 830 and 862: the phrase “biological triplicate” should be corrected to “biological triplicates” to reflect proper plural agreement when referring to three independent biological samples. It is also present in the caption of several figures in the Supplementary Information. Please revise accordingly.

We have edited the manuscript to make these suggested changes.

Line 851: a space is missing in “3kDa.” For clarity and correct formatting of units, it should be written as “3 kDa.” Please revise accordingly.

We have edited the manuscript to make this suggested change.

Line 916: “room temp” is used as an abbreviation, while the rest of the manuscript consistently uses the full term “room temperature.” Please revise for consistency by using “room temperature” throughout.

We have edited the manuscript to make these suggested changes.

Line 930: the phrase “HCD fragmentation with 30% normalized collision” is missing the verb, making the sentence incomplete. Please revise it to a more grammatically correct form, such as: “HCD fragmentation with 30% normalized collision energy was used.”

We have edited the manuscript to make this suggested change.

Lines 934 and 943: The term “one-pot” appears inconsistently formatted in the manuscript—hyphenated in Lines 405 and 408, but not in Lines 934 and 943. As “one-pot” is a compound adjective, it should be hyphenated consistently throughout. Please revise accordingly.

We have edited the manuscript to make these suggested changes.

Line 1057 and 1071: in Line 1007, “L-selectride” is formatted correctly, but in Lines 1057 and 1071 the formatting is inconsistent.

We have edited the manuscript to make these suggested changes.

Throughout the whole text:

in Line 1056, “dichloromethane” is written out in full, whereas the abbreviation “DCM” is used elsewhere in the manuscript. For consistency, we recommend using “dichloromethane (DCM)” at first mention and the abbreviation “DCM” thereafter.

We have edited the manuscript to make these suggested changes.

Throughout the manuscript, there are inconsistencies in subject–verb agreement when referring to volumes greater than 1 $\mu\text{L}/\text{mL}$. In scientific writing, plural forms should be used when referencing quantities greater than one (e.g., “10 μL were added” rather than “10 μL was added”). While some instances are correctly written (e.g., Lines 569, 578, 585, and 613), several others require revision to ensure grammatical consistency and accuracy. I noted such cases in Lines 571, 579, 587, 594, 601, 610, 612, 648, 652, 661, 663, 665, 678, 682, 721, 728, 736, 738, 741, 743, 745, 754, 763, 771, 798, 910, 918, 955, and 968. A careful review and correction of these instances is recommended.

We have edited the manuscript to make these suggested changes.

The abbreviation EICs is introduced in Line 113 for “extracted ion chromatograms,” but both the full term and the abbreviation are used alternately throughout the manuscript (e.g., “EICs” in Lines 193, 200, 342; “extracted ion chromatograms” in Lines 196, 341, 345, 520...). For consistency and clarity, we recommend using the full term at first mention, followed by the abbreviation in parentheses, and using only EICs thereafter. Please revise accordingly.

We have edited the manuscript to make these suggested changes.

The manuscript inconsistently alternates between writing out “minutes” (e.g., Lines 823, 824, 851...) and “hours” (e.g., Lines 994, 1009, 1037...) and using their abbreviations “min” (e.g., Lines 613, 650, 663...) and “h” (e.g., Lines 110, 127, 729, 909...).

For consistency and clarity, we recommend choosing one format—typically abbreviations are preferred in scientific writing—and applying it uniformly throughout the manuscript.

We have edited the manuscript to make these suggested changes.

The term *in vitro* is italicized in Line 705 but appears in regular font in other parts of the manuscript (e.g., Lines 65, 260, 302, 345, 820, 848, 882). As *in vitro* is a Latin expression, it should be consistently italicized throughout the manuscript in accordance with scientific style conventions (if I am not mistaken, check Nature journal rules). Same happens with “*in vivo*” and “*in vitro*” mentioned in the Supplementary Information. Please revise accordingly.

In accordance with Nature policies, we have standardized all Latin phrases to appear in normal font.

The abbreviations IPTG (e.g., Lines 802 and 839) and OD (used multiple times) are not defined upon first use. For clarity and completeness, please provide the full name at first mention—e.g., isopropyl β -D-1-thiogalactopyranoside (IPTG) and optical density (OD)—and use the abbreviation consistently thereafter. This will help ensure accessibility for a broader readership.

We have edited the manuscript to make these suggested changes.

The manuscript uses both abbreviations (e.g., ACN, MeOH) and full names (acetonitrile, methanol) inconsistently across different sections—for instance, “ACN” appears in Lines 618, 652, and 665, while “acetonitrile” is used in Lines 517, 526, and 824. Similarly, both “MeOH” (e.g., Lines 572, 578) and “methanol” (e.g., Lines 1009, 1011, 1075) are used. We recommend standardizing the use of solvent names throughout the manuscript, using the full name at first mention with the abbreviation in parentheses (e.g., “acetonitrile (ACN)”), and then using the abbreviation consistently thereafter.

We have edited the manuscript to make these suggested changes.

The term “UV-vis” appears with inconsistent capitalization across the manuscript; both “UV-vis” (e.g., 376, 378, 401) and “UV-Vis” (e.g., 920) are used. Please standardize the formatting throughout.

We have edited the manuscript to make these suggested changes.

Supporting information

Supplementary Figure 1 and 2: the references to the papers do not match the content of the figures. Please review and correct the reference numbers in both figures to ensure they align with the proper citations in the manuscript.

We have edited the manuscript to make these suggested changes.

Supplementary Figure 1B: there are two issues in this figure that require correction. First, the enzyme name “SpiED” is written with a comma between “E” and “D”, which incorrectly implies two separate enzymes. However, SpiED is a single fusion protein encoding homologues of CreE and CreD. Please remove the comma.

We have edited the manuscript to make this suggested change.

Second, the reaction catalyzed by SpiA7 is inaccurately depicted. Both the substrate and product structures shown are incorrect. Specifically, the molecule includes a hydroxyl group that should not be present, and the product should contain a diazonium group, not a neutral diazo. We strongly recommend redrawing this reaction using the correct structures as established in reference 28.

We have edited the manuscript to make this suggested change.

Supplementary Figure 1D: there is a typographical error in the name of the substrate for CmaA6. It is incorrectly written as “3-aminocoumaic acid,” missing the “r” in “coumaric.” The correct name is 3-aminocoumaric acid.

We have edited the manuscript to make this suggested change.

Supplementary Table 3: the plasmid name “pdualp” appears in lowercase in the last two rows referring to *S. coelicolor* strains. To maintain consistency with nomenclature used throughout the manuscript and figures, this should be corrected to “pDualP.” Please revise accordingly.

We have edited the manuscript to make this suggested change.

Supplementary table 4 and 5: the tables list accession numbers for various strains, but the source database (e.g., NCBI or another repository) is not specified. For clarity and easy accession, please indicate the origin of these accession numbers either in the table legend or a dedicated column. This clarification will help readers locate the sequences reliably.

We have edited the manuscript to make this suggested change.

Supplementary Table 5: there are formatting mistakes in strain names that should be corrected. In the second row, *Actinomadura* sp. J1-007 should be corrected to *Actinomadura* sp. J1-007. In the last row, *Halostreptopolyspora alba* YIM 96095 should be formatted as *Halostreptopolyspora alba* YIM 96095.

We have edited the manuscript to make this suggested change.

Supplementary Figure 18: there is a space missing between “homodimer.” and “A”).

We have edited the manuscript to make this suggested change.

References missing: please consider including some of these references to improve the completeness of the paper:

Creameomycin:

Waldman, A. J.; Pechersky, Y.; Wang, P.; Wang, J. X.; Balskus, E. P. The Creameomycin Biosynthetic Gene Cluster Encodes a Pathway for Diazo Formation. *ChemBioChem*. 2015, 16 (15), 2172– 2175.

Lomaiviticin:

Kersten, R. D.; Lane, A. L.; Nett, M.; Richter, T. K. S.; Duggan, B. M.; Dorrestein, P. C.; Moore, B. S. Bioactivity-Guided Genome Mining Reveals the Lomaiviticin Biosynthetic Gene Cluster in *Salinispora tropica*. *ChemBioChem*. 2013, 14 (8), 955– 962.

Discovery of the nitrous acid biosynthetic pathway (worth mentioning in Line 40):

Sugai, Y.; Katsuyama, Y.; Ohnishi, Y. A Nitrous Acid Biosynthetic Pathway for Diazo Group Formation in Bacteria. *Nat. Chem. Biol.* 2016, 12 (2), 73– 75.

We thank the Reviewers for these suggestions to improve completeness of our manuscript. We have now included the additional creameomycin and nitrous acid biosynthetic pathway references in line 41 of the main text and in the caption of Supplementary Figure 1. We have also included the lomaiviticin reference in line 101 of the main text.

Referee #2 (Remarks to the Author):

I co-reviewed this manuscript with Φ one of the Reviewers who provided the listed reports.

We thank Reviewers 1 and 2 for their detailed and thorough feedback which have substantially improved the revised manuscript.

Referee #3 (Remarks to the Author):

The MS focusses on the biosynthesis of unusual diazo functionalised natural products. The diazo group is quite well known as a naturally occurring motif, and biosynthetic pathways are known, for example to the canonical compound azaserine. The initial parts of the paper try to make the point that diazo compounds are rather unstable, and therefore that special methods are needed to capture and study them. This is not a very convincing argument, because diazo compounds are common synthetic reagents - it is possible to buy diazoacetone as a commercial material, so it is unlikely that the low titres of such compounds are due to the instability of the diazo group per se. It is more likely that the compounds are simply produced in low titre.

We agree with the Reviewer that in the context of synthesis, the diazo group of DAC, DOBA, and some other diazo compounds (e.g., ethyl diazoacetate) are relatively stable in neat solutions or organic solvents at reduced temperature and with careful handling (e.g. to reduce exposure to light, heat, and moisture). However, we anticipate that DAC and DOBA will be relatively unstable under culture conditions, as they are good electrophiles being produced in aqueous solvent containing many nucleophilic metabolites, biomolecules, and media components at 30 °C.

In addition to improved stability, the pyrazoles that result from the reaction between diazo-containing metabolites and DBCO-acid have greatly improved ionization efficiency compared to the underivatized diazo compounds, which results in greatly enhanced LC-MS sensitivity. For example, derivatization of DAC with DBCO-acid improved the limit of detection (LOD) by approximately two orders of magnitude. The LOD was between 1 and 10 μ M for DAC, and between 10 and 100 nm for DAC-DBCO. Capture of DAC and DOBA by DBCO-acid was critical in our case because the underivatized metabolites were undetectable in cultures—potentially due to low titers and/or instability. We have now included these data in the SI (Supplementary Figure 6) and added a brief discussion of this point to the main text in lines 191-194. We have also included language in lines 194-197 and 241-244 to draw attention to the utility of our workflow for detecting natural product with low titers.

In response to this point and a comment from Reviewer 1 and 2 (see above), we have now added a brief discussion in lines 60-64 of the main text to offer alternative explanations for the relatively small number of reported diazo-containing natural products. These include low production titers, silent expression, poor detection via analytical methods, and/or gene clusters producing redundant metabolites.

It would have been useful throughout to estimate the titres of the detected compounds - this would not be hard to do given that the authors have materials in hand.

We thank the Reviewer for this suggestion. We note that it is challenging to accurately measure DOBA titers in culture because of rapid decarboxylation of both the culture samples and the synthetic standard. We have therefore only attempted to quantify the titers of DAC. The synthetic standard of DAC-DBCO is stable and easy to quantify, however, we anticipate that DAC in cultures is both volatile and relatively unstable. It is worth noting that quantification is likely further complicated by competing rates of DAC degradation and DAC capture by DBCO-acid such that our calculated titers are likely an underestimate of the total production of DAC. In addition, the reaction between DAC and DBCO-acid does not give 100% yield, leading to further underestimation of DAC levels in cultures. Nevertheless, we have now constructed a standard curve using synthetic DAC-DBCO, and subjected *N. ninae* spent medium to DBCO-acid derivatization to obtain an estimate of approximately 3 μM of DAC produced in 5 mL of culture after 7 days of growth. We also estimated that *N. tenerifensis* produces approximately 1 μM of DAC in 5 mL of culture after 7 days of growth. Further, the heterologous host *S. coelicolor* *dob*-pDualP produces approximately 2 μM of DAC after 14 days of growth, induction, and another day of growth.

We have now included these data in Supplementary Figures 7, 10, and 13 and discussed them the main text in lines 193-194, 241-244, and 252-255.

However, having said this, low titre diazo compounds, especially with the low molecular weights of diazoacetone and diazoacetoacetate will be challenging to isolate, and so the use of a specific molecular 'bait' to capture compounds from complex milieu is a sensible idea.

We thank the Reviewer for acknowledging the utility of our workflow for detecting challenging analytes like DAC and DOBA. As noted in several responses above, our strategy for capturing diazo-containing metabolites with DBCO-acid is also beneficial for discovery of low titer diazo natural products.

This follows a well-trodden path where molecular hooks have been used in many other cases to capture biosynthetic intermediates. Here, those hooks are the well-known strained alkynes used in Huisgen dipolar cycloaddition click reactions. Thus, I don't think that the capture and analysis methodology is particularly novel. However, these tools do allow the characterisation of a new pathway to biological diazo compounds, and it is here that the work has its key interest.

We agree with the Reviewer that reactivity-based screening and 1,3-dipolar cycloaddition click reactions are conceptually well precedented. However, this is the first example that leverages these approaches for diazo natural product discovery. As noted above, diazo-containing metabolites are especially challenging to detect due to instability and low titers. Our new application of these well-known approaches now addresses these longstanding issues, and therefore greatly facilitates the study of this interesting class of microbial natural products. Further, it is uncommon to combine this natural product discovery strategy with studies of biosynthesis. As noted by the Reviewer, this approach allowed us to investigate a new pathway for biological diazo formation, ultimately leading to the characterization of a novel diazo-forming enzyme with promising potential biocatalytic applications.

The genome mining and coupling to metabolomic analysis is very well done and the logic and data is clear. The combined use of heterologous expression in *S. coelicolor* and then targeted in vitro assays convincingly illustrates the new chemistry, and the use of Dob3 as a potential catalyst for the production of more 'synthetic' diazo compounds is a useful addition to the work.

We thank the Reviewer for recognizing the strengths of the manuscript, particularly the in vivo and in vitro evidence that we provide to support new enzymatic chemistry, and the demonstration of Dob3 as a potential biocatalyst for producing diazo compounds.

The story is logical, but I found it quite wordy - there is a lot of justification for experiments that could be cut - just explain what was done and what was observed.

We thank the Reviewer for this suggestion. During this revision we have focused on balancing these types of cuts with the addition of text to address reviewer comments. We anticipate making additional cuts to the main text during further revisions at the suggestion of the Editor.

Overall I feel that the key interest here is the new pathway to diazo compounds and characterization of the biosynthetic steps and associated genes and BGCs. The reactivity based screening is less novel. Never-the-less this paper contains significant

interesting material for many working in the field of biosynthesis, and I feel it will be read and well cited.

We thank the Reviewer for recognizing the novelty, significance, and broad appeal of the work, particularly regarding the new pathway to diazo compounds and characterization of the biosynthetic steps. As noted above, while the approach of reactivity-based screening is well-precedented, this is the first application of this approach for diazo-containing natural products, which addresses longstanding challenges of instability and low titers.

Other points:

1. I (and Google) could not find reference 51...

Reference 51 refers to the prettyClusters toolkit which is unpublished. We have now included the URL for the Github repository to aid in accessing this tool.

2. ¹H and ¹³C NMR spectra for all synthetic compounds should be assigned - are these known compounds? if so then citation of relevant literature should be given. If new, then the NMR spectra should be shown.

We have now provided NMR spectra for all synthesized compounds. We have also included peak assignments where possible (Methods Section, lines 633-636, 666-669, and 681-685 of the SI). Per *Nature's* policy on compound characterization, we have only assigned spectra for which we have 2D NMR evidence for peak assignments. We note that for the products of DBCO-acid cycloadditions it has previously been reported to be very challenging to separate regioisomers, and thus it is typical to report these compounds as mixtures of regioisomers (PMID: 35797662, 30931444, 30198046, 31368231, 24307493, 30180567, 24338798).

3. NMR data for hydroxy lysine and HAA should be given.

As noted above, we have now included NMR spectra for all synthesized compounds including N6-hydroxy-L-lysine (**8**) (Supplementary Figures 40 and 41) and HAA (**10**) (Supplementary Figures 42 and 43).

Response to reviewers - Second Revision

Referees' comments:

Referee #1 (Remarks to the Author):

We have read both present and previous versions of the manuscripts and the authors' responses to the reviewers. The authors have done an excellent job in carefully revising the manuscript according to all Reviewers' comments. Much effort has been devoted to address our comments and other reviewers' (Reviewer 3). The authors' responses are reasonable, and with the additional data and the changes to the main manuscript the quality of the manuscript has significantly improved.

We only identified some last very minor editorial issues that should be addressed when preparing the final version of the manuscript. These are outlined below:

Line 195, main manuscript: *Nocardia tenerifensis* is always abbreviated, first appearance in line 208 should not be.

We have edited the manuscript to make this change.

Line 426 in Supplementary info: 3k Da should be 3 kDa, the space should be added before k, not after.

We have edited the manuscript to make this change.

Throughout the whole text: there are inconsistencies in subject–verb agreement when referring to volumes greater than 1 $\mu\text{L}/\text{mL}$. In scientific writing, plural forms should be used when referencing quantities greater than one (e.g., “10 μL were added” rather than “10 μL was added”). While some instances are correctly written several others require revision to ensure grammatical consistency and accuracy. A careful review and correction of these instances is recommended. Some examples are mentioned below:

We have carefully reviewed the supplementary information and all instances of subject-verb disagreement regarding volumes should now be addressed.

Line 210 in supplementary info: 200 μL of spent culture medium was spun down at maximum speed for 5 min.

We have edited the manuscript to make this change.

Line 258: 2 mL of 10% glucose was added

We have edited the manuscript to make this change.

Supplementary Figure 1: The references are incorrect, and the numbering is not done correctly. Please revise accordingly.

We have revised the manuscript to correct the numbering within the figure and the references.

Referee #2 (Remarks to the Author):

I co-reviewed this manuscript with one of the reviewers who provided the listed reports.

Referee #3 (Remarks to the Author):

The MS has been significantly improved and the authors have addressed the large majority of the comments from the first round of review. My view now is 'publish after minor revisions':

1. The title should perhaps be changed to specify alpha-diazoacetyl metabolites because these are the only ones to have been captured and analysed here - the authors themselves mention that the capture process works best for alpha-diazo-carbonyls (line 120 ish), but no other types of diazo natural products have been demonstrated yet as targets.

We thank the Reviewer for this suggestion. While we understand the logic for this change, we note that while alpha-diazoacetyl metabolite are the focus of this work, our method could, in principle, be applied to a variety of diazo-containing metabolites. Further, we are unable to make the suggested change to the title and still conform to the character limit.

2. line 41 - remove the sentence "A potential exception....protein expression" - this does not seem to add any useful information.

We have edited the manuscript to make this change.

3. line 55 - 469 BGCs were found in diverse bacteria.... It would be useful to explain if only bacterial genomes were examined, or if genomes from other kingdoms were examined, but hits were only found in bacteria.

Some, but not all, of the previous bioinformatic searches that we compiled queried databases containing genomes from diverse sources including plants, animals, fungi, etc. However, hits were only observed from microbial genomes.

Our bioinformatic search used the NCBI reference protein database which contains proteins from a variety of sources including microbes, plants, animals, humans, fungi, etc. We only observed hits from microbes.

We have now included this information in the description of our genome-mining approach in the Methods Section.

4. Fig 1C - Insert 'bacterial' between 'Predicted' and 'biosynthetic'.

We have edited the manuscript to make this change.

5. line 95/97 replace at least one of the 'leverages' with 'using'...

We have edited the manuscript to make this change.

6. Line 269 - the data in Supplementary Figs 14-18 is essential to the story and it would be better to include this information in the actual paper itself - it is very inconvenient for readers to have to go searching through the supplementary info for key data.

We have now moved the activity assay data for DobG, DobE, DobB, DobABCMQ, and DobABCMQ23 to the Extended Data.

7. I don't understand the bar graphs in Fig 4D.... the vertical scale is 'Fold change vs no enzyme' which makes sense, but the second bar is No Dob3 - which is no enzyme and so by definition is 1.... so it makes no sense to include this second bar ? I guess the authors are trying to emphasise the differences for the different substrates - so it would make more sense to combine the bars from all 6 cases into a single chart.

We thank the Reviewer for this suggestion. We have now removed the "No Dob3" bars from each graph. We have also revised the figure such that all substrates from the Dob3 enzymatic reactions are combined in a single chart. For clarity, we have left the bar graphs for each chemoenzymatic reaction adjacent to their respective reaction scheme.

8. line 449 replace 'diazo-' with 'alpha-diazoacetyl-'

After shortening the manuscript at the Editor's request, this change is no longer applicable.

9. line 445 add "and comparison to standard materials" at the end of the sentence

After shortening the manuscript at the Editor's request, this change is no longer applicable.